

# Inverse-model estimates of the ocean's coupled phosphorus, silicon, and iron cycles

Benoît Pasquier[1] and Mark Holzer[1,2]

[1]Department of Applied Mathematics, School of Mathematics and Statistics, University of New South Wales, Sydney, NSW 2052, Australia
[2]Department of Applied Physics and Applied Mathematics, Columbia University, New York, NY., USA

*Correspondence to:* Benoît Pasquier (b.pasquier@unsw.edu.au)

**Abstract.** The ocean's nutrient cycles are important for the carbon balance of the climate system and for shaping the ocean's distribution of dissolved elements. Dissolved iron (dFe) is a key limiting micronutrient, but iron scavenging is observationally poorly constrained leading to large uncertainties in the external sources of iron and hence in the state of the marine iron cycle.

Here we build a model of the ocean's coupled phosphorus, silicon, and iron cycles embedded in a data-assimilated steady-state global ocean circulation. The model includes the redissolution of scavenged iron, parameterization of subgrid topography, and small, large, and diatom phytoplankton functional classes. Phytoplankton concentrations are implicitly represented in the parameterization of biological nutrient utilization through an equilibrium logistic model. Our coupled nutrient model thus carries only three nutrient tracers whose three-dimensional steady-state distributions can be found efficiently using a Newton solver. The very efficient numerics allow us to use the model in inverse mode to objectively constrain many biogeochemical parameters by minimizing the mismatch between modelled and observed nutrient and phytoplankton concentrations. We consider a family of possible solutions corresponding to a wide range of external iron source strengths. Iron source and sink parameters cannot jointly be optimized because of local compensation between regeneration, recycling, and scavenging. All optimized solutions have a similar mismatch with the observed nutrient concentrations and very similar large-scale dFe distributions. However, the relative contributions of aeolian, sedimentary, and hydrothermal iron to the total dFe concentration differ widely depending on the sources.

Both the magnitude and pattern of carbon and opal export are well constrained with global values of $(10.3 \pm 0.4)\,\mathrm{Pg\,C\,yr^{-1}}$ and $(171. \pm 3.)\,\mathrm{Tmol\,Si\,yr^{-1}}$. We diagnose the carbon and opal export supported by aeolian, sedimentary, and hydrothermal iron. The geographic patterns of the export supported by each iron type are well constrained across the family of solutions. Sedimentary-iron supported export is important in shelf and large-scale upwelling regions, while hydrothermal iron contributes to export mostly in the Southern Ocean. The globally integrated export supported by a given iron type varies systematically with the fractional contribution of its source to the total iron source. Aeolian iron is most efficient in supporting export in the sense that its fractional contribution to export exceeds its fractional contribution to the total source by as much as $\sim$30% for carbon and $\sim$20% for opal export. Conversely, sedimentary and hydrothermal iron are less efficient with a fractional export that is less than their fractional sources. For the same fractional contribution to the total source, hydrothermal iron is less efficient than sedimentary iron for supporting carbon export but about equally efficient for supporting opal export.





# 1 Introduction

The ocean's nutrient cycles control the primary productivity of the global marine ecosystem and the ocean's biological carbon pump, which are crucial components of the global carbon cycle that regulate atmospheric $CO_2$ concentrations. The nutrient cycling of the ocean is governed by the interplay of the ocean's advective-diffusive circulation, biological utilization, biogenic particle transport, and the external sources and sinks of nutrients. The cycles of macro and micronutrients are coupled through colimitation on biological uptake, and through the scavenging of micronutrients such as iron by sinking organic matter.

We focus on dissolved iron (dFe) as a key micronutrient because of its well-documented fundamental role in primary production (e.g., Boyd and Ellwood, 2010). Indeed, dFe was suggested to limit oceanic phytoplankton growth as early as the 1930's (e.g., Gran et al., 1931; Hart, 1934). Since then, numerous studies have reported that iron deficiency limits productivity over vast regions of the ocean, particularly high-nutrient low-chlorophyll (HNLC) regions like the Southern Ocean (e.g., de Baar et al., 1995; Lundry et al., 1997; Martin and Fitzwater, 1988; Boyd et al., 2007; Boyd and Ellwood, 2010). Martin (1990) went as far as to suggest that perturbations in the iron cycle played a crucial role in past climate fluctuations. More recently, iron-enrichment field experiments (e.g., Boyd et al., 2007) and model simulations (e.g., Nickelsen and Oschlies, 2015) have demonstrated the importance of iron for the global biological pump.

With a changing climate, we expect not only changes in the ocean circulation, but also changes in the winds, hydrological cycle, and land use, and hence in the aeolian iron supply. To understand how such changes impact the global ocean nutrient cycles it is necessary to model the coupling between the nutrients mechanistically. While global biogeochemistry models have been extensively used for this purpose (e.g., Tagliabue et al., 2014; Nickelsen and Oschlies, 2015), here it is our goal to constrain a model of the coupled nutrient cycles by optimizing the biogeochemical parameters against available observations. To that end, we build a model of intermediate complexity that focuses on the ocean's phosphorus, silicon, and iron cycles.

We model phosphate ($PO_4$) because it is essential to the metabolism of all living organisms (e.g., Smith, 1984; Howarth, 1988), which allows all biological production to be keyed to phosphate utilization (e.g., Kwon and Primeau, 2008; Primeau et al., 2013; Holzer and Primeau, 2013). Silicic acid ($Si(OH)_4$) was considered because of the importance of diatoms in in marine ecosystems, particularly in the Southern Ocean (e.g., Nelson et al., 1995; Buesseler, 1998; Moore et al., 2004; Brzezinski et al., 2011b), and because the pronounced silicon trapping of the Southern Ocean (e.g., Holzer et al., 2014) might be sensitive to iron availability.

The intercomparison of iron models by Tagliabue et al. (2016) showed that current models of the iron cycle contain significant uncertainties. Despite the fact that the models have iron source strengths that range over nearly two orders of magnitude, all models can be tuned to roughly the same mean dFe concentration with an inter-model variance of only 27%. This is due to essentially unconstrained scavenging rates so that models are free to employ different scavenging strengths to balance the sources at roughly comparable dFe concentrations. All the models of the intercomparison are prognostic forward models that are computationally too costly to explore the biogeochemical parameter space systematically, or to compute the sensitivity with respect to multiple parameters (Kwon and Primeau, 2006).



Recently, Frants et al. (2016) designed a simple inverse model of the global iron cycle embedded in the steady data-assimilated circulation of Primeau et al. (2013). A matrix representation of the associated transport operator and biogeochemical processes afforded numerically highly efficient solutions. This made it possible not only for biogeochemical parameters to be systematically optimized, but also for novel diagnostics to be computed such as a mean iron age and rigorous source

attribution of dFe (Holzer et al., 2016). Consistent with the findings of the iron-model intercomparison, Frants et al. (2016) showed that current dFe observations cannot constrain the iron sources because of local compensation between sources and sinks. Frants et al. (2016) therefore explored a family of solutions corresponding to a range of aeolian source strengths, all of which are consistent with the currently available dFe observations.

Here, we build on the iron model of Frants et al. (2016), but do not prescribe the phosphate cycle. Instead, we explicitly

couple the iron, phosphorus, and silicon cycles through their mutual colimitations so that the macronutrients can respond to changes in dFe. We furthermore refine the modelling of the sedimentary iron source, the representation of iron scavenging that now includes explicit representation of redissolution, and we model three phytoplankton functional classes, whose concentrations are derived from a steady-state logistic equation (Dunne et al., 2005).

The model's biogeochemical parameters are optimized by minimizing the quadratic mismatch of the nutrient and phyto-

plankton concentrations with the available observations. Following Frants et al. (2016), we consider a family of dFe solutions corresponding to a range of sources, expanded here to a greater range of sediment and hydrothermal sources. The spread in key metrics (e.g., the global carbon export) across our family of solutions is used as a measure of the metric's uncertainty.

We use the model to establish the geographic patterns of nutrient limitation and colimitation, and to quantify the export of each nutrient by each phytoplankton functional class. By partitioning dFe into contributions from the aeolian, sedimentary, and

hydrothermal sources, we quantify the role and efficiency of each dFe type in supporting carbon and opal export.

## 2    Biogeochemical model

We distinguish three phytoplankton functional groups: non-diatom small and large phytoplankton as well as diatoms, with a nominal separation between small and large at a cell diameter of $2\,\mu\text{m}$. We denote the molar $PO_4$ uptake rate per unit volume of each class by $U_c$, where the subscript $c \in \{\text{lrg}, \text{sml}, \text{dia}\}$ identifies functional class. The uptake rates $U_c$ are only non-zero

in the model's upper 73.4 m (2 layers), the model's euphotic zone.

We consider the three nutrients $PO_4$, $Si(OH)_4$, and dFe and denote their concentrations by $\chi^i$, with $i \in \{P, Si, Fe\}$. We write the steady-state tracer equations for these concentrations by keying all biological production to the uptake $U_c$ of phosphate as





follows:

$$\mathcal{T}\chi_{\mathrm{P}} = \sum_c (\mathcal{S}_c^{\mathrm{P}} - 1)U_c - \gamma_g(\chi_{\mathrm{P}} - \overline{\chi}_{\mathrm{P}}^{\mathrm{obs}}), \tag{1}$$

$$\mathcal{T}\chi_{\mathrm{Si}} = (\mathcal{S}^{\mathrm{Si}} - 1)R^{\mathrm{Si:P}}U_{\mathrm{dia}} - \gamma_g(\chi_{\mathrm{Si}} - \overline{\chi}_{\mathrm{Si}}^{\mathrm{obs}}), \tag{2}$$

$$\mathcal{T}\chi_{\mathrm{Fe}} = \sum_c (\mathcal{S}_c^{\mathrm{Fe}} - 1)R_c^{\mathrm{Fe:P}}U_c \tag{3}$$

$$+ (\mathcal{S}^{\mathrm{s,POP}} - 1)J_{\mathrm{POP}} + (\mathcal{S}^{\mathrm{s,bSi}} - 1)J_{\mathrm{bSi}} - J_{\mathrm{dst}}$$

$$+ s_{\mathrm{A}} + s_{\mathrm{S}} + s_{\mathrm{H}} \quad .$$

In (1)–(3), $\mathcal{T}$ is the advection-eddy-diffusion operator, the operators $\mathcal{S}_c^i$ model the biogenic transport and remineralization of nutrient $i$ taken up by functional class $c$, and the operators $\mathcal{S}^{\mathrm{s,POP}}$ and $\mathcal{S}^{\mathrm{s,bSi}}$ model the particle transport of scavenged iron and its partial redissolution at depth as the scavenging particles remineralize or dissolve (details in section 2.2). The iron scavenging rates per unit volume are $J_{\mathrm{POP}}$ for scavenging by particulate organic phosphorus (POP), $J_{\mathrm{bSi}}$ for scavenging by opal particles, and $J_{\mathrm{dst}}$ for scavenging by mineral dust (details in section 2.4.2). The terms $s_{\mathrm{A}}$, $s_{\mathrm{S}}$, and $s_{\mathrm{H}}$ are the aeolian, sediment, and hydrothermal iron sources (details in section 2.4.1). The factors $R^{\mathrm{Si:P}}$ and $R_c^{\mathrm{Fe:P}}$ are the stoichiometric uptake ratios that allow us to key all production to phosphorus. These ratios are functions of the nutrient concentrations as described in section 2.3.3.

The terms proportional to $\gamma_g$ in (1)–(2) fix the global mean phosphate and silicic acid concentrations through weak relaxation to their observed global means $\overline{\chi}_{\mathrm{P}}^{\mathrm{obs}}$ and $\overline{\chi}_{\mathrm{Si}}^{\mathrm{obs}}$. This is necessary because the phosphorus and silicon cycles have no external sources and sinks to set the global mean in steady state. (For phosphate and silicic acid, external sources, e.g., riverine input, and loss to sediment burial are neglected.) We choose the restoring timescale $\gamma_g^{-1} = 10^6$ years ("geological" restoring); there is no sensitivity to the precise value of $\gamma_g$.

Equations (1)–(3) are coupled via the uptake of $PO_4$, which depends on the concentrations of all three nutrients, via the iron scavenging that depends on the export fluxes of organic matter and opal, and via the sedimentary release of dFe, which is keyed to the flux of organic matter onto the sediments (Elrod et al., 2004), as discussed in detail below.

## 2.1 Circulation

We use the data-assimilated, steady (non-seasonal) circulation of Primeau et al. (2013) which has a horizontal resolution of $2° \times 2°$ and 24 vertical levels whose thickness increases with depth. Temperature, salinity, and radiocarbon, $CFC-11$, and $PO_4$ have been used as constraints in the data assimilation. The circulation is constrained dynamically and the data assimilation used the wind-stress climatology of Trenberth et al. (1989) and specified horizontal and vertical viscosities of $5 \times 10^4 \, \mathrm{m^2 \, s^{-1}}$ and $10^{-4} \, \mathrm{m^2 \, s^{-1}}$, respectively. The circulation's advective-diffusive transport operator has fixed horizontal and vertical eddy diffusivities of $10^3 \, \mathrm{m^2 \, s^{-1}}$ and $10^{-5} \, \mathrm{m^2 \, s^{-1}}$, respectively. We emphasize that the circulation effectively provides a ventilation-weighted transport because it has been optimized against $PO_4$ and the ventilation tracers CFC-11 and radiocarbon. The steady model circulation, which has no seasonal cycle, thus does not bias estimates of preformed nutrients in the way an annual-average circulation would.



## 2.2 Biogenic transport

Organic matter sinks as POP, dissolves, and remineralizes at depth. Inverse models of the phosphorus cycle (Primeau et al., 2013; Holzer and Primeau, 2013; Pasquier and Holzer, 2016) suggest that dissolved organic phosphorus (DOP) represents a relatively small fraction of the total dissolved phosphorus that we neglect here for simplicity and numerical efficiency (no

DOP tracer). Because the particle transport is much faster than the fluid transport across a grid box, we approximate particle transport and remineralization, which acts as an interior source of nutrients, as instantaneous. We model this process for each phytoplankton functional class by the "source" operator, $\mathcal{S}_c^{\mathrm{P}}$, which reassigns a "detrital" fraction $f_c$ of the uptake rate to a remineralization rate throughout the water column, while a fraction $1 - f_c$ remineralizes in situ where the uptake occurred. We therefore express $\mathcal{S}_c^{\mathrm{P}}$ in terms of a biogenic redistribution operator $\mathcal{B}^{\mathrm{P}}$ as

$$\mathcal{S}_c^{\mathrm{P}} = 1 - f_c + \mathcal{B}^{\mathrm{P}} f_c \quad . \tag{4}$$

(The operator $\mathcal{B}^{\mathrm{P}}$ does not have a functional class subscript $c$ because it redistributes a unit uptake with the same profile regardless of functional class.) Following Dunne et al. (2005), we model the detrital fraction as decreasing with temperature $T$ so that $f_c = f_c^0 e^{-k_\sigma T}$, with $k_\sigma = 0.032\,°\mathrm{C}^{-1}$ independent of class, $f_{\mathrm{sml}}^0 = 0.14$, $f_{\mathrm{lrg}}^0 = 0.74$, and we assign $f_{\mathrm{dia}} = f_{\mathrm{lrg}}$. We assume that the remineralization of organic matter releases dFe and phosphate in the same ratio with which they were taken

up. Therefore, $\mathcal{S}_c^{\mathrm{Fe}} = \mathcal{S}_c^{\mathrm{P}}$.

Following Najjar et al. (1992), we assume that the detrital production rate is fluxed as POP through the base of the euphotic zone at $z_e = 73.4\,\mathrm{m}$ with $\phi^{\mathrm{POP}}(z_e) = \int_{z_e}^0 f_c U_c dz$, and that the POP flux attenuates with depth according to the Martin power law $\phi^{\mathrm{POP}}(z) = \phi^{\mathrm{POP}}(z_e)(z/z_e)^{-b}$ due to remineralization in the aphotic zone. The operator $\mathcal{B}^{\mathrm{P}}$ therefore injects $\mathrm{PO}_4$ with the divergence of $\phi^{\mathrm{POP}}$ into the aphotic water column. The flux into the ocean bottom is remineralized in the lowest grid box as

in the work of Primeau et al. (e.g., 2013). The exponent $b$ was determined to be $b = 0.82$ using a restoring-type phosphate-only model. (Most parameters were optimized for the full coupled model – for details of our optimization strategy see section 3.3.)

The redistribution operator $\mathcal{B}^{\mathrm{Si}}$ similarly injects silicic acid into the aphotic water column with the divergence of the opal flux, $\phi^{\mathrm{bSi}}$, which attenuates because of temperature dependent opal dissolution following Gnanadesikan (1999) and Holzer et al. (2014). For each latitude and longitude, $\phi^{\mathrm{bSi}}$ is computed as the solution to $\partial_z \phi^{\mathrm{bSi}}(z) = -(\kappa_{\mathrm{Si}}^{\max}/w_{\mathrm{Si}}) \exp(-T_E/T(z)) \phi^{\mathrm{bSi}}(z)$,

with the boundary condition $\phi^{\mathrm{bSi}}(z_e) = \int_{z_e}^0 R^{\mathrm{Si:P}} f_{\mathrm{dia}} U_{\mathrm{dia}} dz$. We use $T_{\mathrm{E}} = 11{,}481\mathrm{K}$ as Gnanadesikan (1999) and the same detrital fraction $f_{\mathrm{dia}}$ for the opal export and diatom POP export. The parameter combination $\kappa_{\mathrm{Si}}^{\max}/w_{\mathrm{Si}}$ has nearly the same value as determined by Holzer et al. (2014), but was re-optimized here for a simple restoring-type model that takes subgrid topography into account (see below).

The scavenging operators $\mathcal{S}^{\mathrm{s,POP}}$ and $\mathcal{S}^{\mathrm{s,bSi}}$ act on $J_{\mathrm{POP}}$ and $J_{\mathrm{bSi}}$ to redistribute a fraction of the iron scavenged at every

layer throughout the water column below the layer. In terms of the corresponding redistribution operators, we write

$$\mathcal{S}^{\mathrm{s,POP}} = f^{\mathrm{POP}} \mathcal{B}^{\mathrm{s,POP}} \qquad \text{and} \qquad \mathcal{S}^{\mathrm{s,bSi}} = f^{\mathrm{bSi}} \mathcal{B}^{\mathrm{s,bSi}} \quad , \tag{5}$$

where the fractions $f^{\mathrm{POP}}$ and $f^{\mathrm{bSi}}$ were both fixed at $0.9$ (see Appendix D). The operators $\mathcal{B}^{\mathrm{s,POP}}$ and $\mathcal{B}^{\mathrm{s,bSi}}$ in effect "recycle" scavenged iron. They are very similar to $\mathcal{B}^{\mathrm{P}}$ and $\mathcal{B}^{\mathrm{Si}}$ but in addition to distributing scavenged iron from the euphotic zone to




the aphotic zone, they also redistribute the scavenging rates of every aphotic layer to a source of redissolving iron with the divergence of the scavenging particle fluxes. The flux of scavenged iron into the bottom is assumed to be lost forever so that there would be iron loss even for 100% efficient recycling of scavenged iron. (For details see Appendices A and B.)

To compute accurate particle fluxes for constructing all $\mathcal{S}$ operators, we take sub-grid topography into account (as done by Moore and Braucher, 2008), using the high-resolution ETOPO2V2c data set (National Geophysical Data Center, 2006). This is done by calculating for each grid box the fractional area occupied by the sub-grid topography, which is also the fraction of the particle flux that is intersected by the sub-grid topography. For each grid box, the fraction of the flux intersected is instantly remineralized or dissolved (details in Appendix B).

### 2.3 Uptake rates

The $PO_4$ uptake rate at a point is a function of the local temperature $T$, irradiance $I$, and nutrient concentrations. The uptake rate for functional class $c$ is calculated as the product of its phytoplankton concentration, $p_c$, and its specific growth rate, $\mu_c$, as

$$U_c = \mu_c\, p_c = \frac{p_c^{\max}}{\tau_c}\, e^{\kappa T}\, \left(F_{I,c}\, F_{\mathrm{N},c}\right)^2 \quad , \tag{6}$$

where $\tau_c$ is the timescale for growth, $p_c^{\max}$ is the phytoplankton concentration under ideal conditions, and $F_{I,c}$ and $F_{\mathrm{N},c}$ are dimensionless factors in the interval $[0,1)$ that represent light and nutrient limitation, respectively, as defined below. We derive equation (6) similarly to Dunne et al. (2005) and Galbraith et al. (2010) as follows.

First, $p_c$ is calculated diagnostically by assuming steady state between growth and mortality, which avoids the need to carry explicit plankton concentration tracers. This is justified by the coarse resolution of our model, which implies transport timescales across a grid box much larger than the typical timescales for phytoplankton growth. Based on Dunne et al. (2005)'s mortality formulation, $p_c$ can be modelled by a logistic equation

$$\partial_t p_c = \mu_c\, p_c - \lambda \left(\frac{p_c}{p_c^*}\right) p_c \quad , \tag{7}$$

where the $p_c/p_c^*$ scales the specific mortality rate $\lambda$, and $p_c^*$ is also referred to as the "pivotal" population density (e.g., Galbraith et al., 2010). Equation (7) has a nontrivial steady state, given by

$$p_c = \left(\frac{\mu_c}{\lambda}\right) p_c^* \quad . \tag{8}$$

We assume that all phytoplankton classes share the same specific mortality rate $\lambda$, which depends only on temperature. Following Eppley (1972), we use $\lambda = \lambda_0\, e^{\kappa T}$, where $\lambda_0$ is a constant and $T$ is in $^\circ C$.

Our formulation differs from that of Dunne et al. (2005) and Galbraith et al. (2010), who raise the ratio $p_c/p_c^*$ to a power $\alpha = 1$ or $\alpha = 1/3$ to differentiate between their small and large phytoplankton classes. Here, we instead differentiate between classes by assigning them different half-saturation rates and maximum uptake rate constants similarly to the work of Matsumoto et al. (2008) (details in sections 2.3.1 and 2.3.3).





We model the specific growth rate $\mu_c$ as multiplicatively colimited (Saito et al., 2008) by temperature, light, and nutrients:

$$\mu_c = \frac{1}{\tau_c} e^{\kappa T} F_{I,c} F_{N,c}, \tag{9}$$

where $\tau$ is the growth timescale at $0°C$ under ideal conditions. For simplicity, the temperature dependence $e^{\kappa T}$ is identical to what we use in the mortality rate (e.g., Galbraith et al., 2010). To group parameters for more efficient optimization, we define

$p_c^{\mathrm{max}} = p_c^*/(\lambda_0 \tau_c)$, so that diagnostic equation (8) for the phytoplankton concentration becomes

$$p_c = F_{I,c} F_{N,c} p_c^{\mathrm{max}} \quad . \tag{10}$$

Substituting (9) and (10) into $U_c^{\mathrm{P}} = \mu_c p_c$ gives (6), which is similar to the uptake formulation of Doney et al. (2006) and Matsumoto et al. (2008).

We note that in the Sea of Japan the model's circulation produces unrealistic nutrient trapping, likely due to under resolved

currents. For this reason we set the specific growth rate in the Sea of Japan to zero, effectively removing it from the computational domain of the biogeochemical model.

### 2.3.1 Nutrient limitation

We model the limitation of functional class $c$ by nutrient $i$ by a Monod function (Monod, 1942) of the concentration, $\chi_i/(\chi_i + k_c^i)$, where $k_c^i$ is the half-saturation constant that determines the scale on which the concentration influences uptake. (Because

only diatoms take up silicon $k_{\mathrm{lrg}}^{\mathrm{Si}} = 0$ and $k_{\mathrm{sml}}^{\mathrm{Si}} = 0$.) For the colimitation of all three nutrients, we use the type-I multiplicative form (Saito et al., 2008)

$$F_{N,c} = \prod_i \frac{\chi_i}{\chi_i + k_c^i} \quad . \tag{11}$$

We chose the Monod model over the arguably more realistic quota model (e.g., Flynn, 2003) for simplicity. Moreover, the shortcomings of the Monod formulation likely only come into play for rapidly evolving transient blooms, which our steady-

state formulation does not attempt to capture.

Using a minimum over nutrient type $i$ (Liebig's rule, e.g., de Baar, 1994), rather than the product (11) is thought to fit the observational data slightly better (e.g., Droop, 2009; Rhee, 1978). However, here we prefer the smoothness of the multiplicative formulation because differentiability is a theoretical requirement for Newton's method to converge (e.g., Kelley, 2003a). A product of $PO_4$, dFe, and irradiance Monod terms was also used by Parekh et al. (2005) and Dutkiewicz et al. (2006) in the

uptake formulation of their coupled phosphorus-iron model.

### 2.3.2 Light limitation

We prescribe irradiance $I$ and model light limitation with a simple Monod factor

$$F_{I,c} = \frac{I}{I + k_{I,c}}, \tag{12}$$





with half-saturation constant $k_{I,c}$ for class $c$ (e.g., Doney et al., 2006). We use an annual mean $I$ derived from photosynthetically active radiation (PAR) measured over the period 2002–2015 by the Modis Aqua satellite (NASA Goddard Space Flight Center, 2014). The surface PAR at location $(x,y)$, denoted by $I_0(x,y)$, was converted to $\mathrm{W\,m^{-2}}$ using $2.77 \times 10^{18}\,\mathrm{quanta\,s^{-1}\,W^{-1}}$ (Morel and Smith, 1974). Irradiance is modelled as exponentially attenuated with depth $z$ so that

$$I(x,y,z) = I_0(x,y)e^{-k_\mathrm{w}z}, \tag{13}$$

with $k_\mathrm{w}^{-1} = 25\,\mathrm{m}$.

### 2.3.3 Elemental uptake ratios

Because we key all biological production to $\mathrm{PO_4}$ utilization, we must specify the Fe : P and Si : P elemental uptake ratios for the iron and silicon cycles. The Fe : P uptake ratio, $R_c^{\mathrm{Fe:P}}$, is known to increase and saturate with increasing dFe concentration (e.g., Sunda and Huntsman, 1997). We follow Galbraith et al. (2010) and model the dFe dependence as a simple Monod term

$$R_c^{\mathrm{Fe:P}} = R_0^{\mathrm{P}}\,\frac{\chi_{\mathrm{Fe}}}{\chi_{\mathrm{Fe}} + k_{\mathrm{Fe:P}}}\quad, \tag{14}$$

where $R_0^{\mathrm{Fe:P}}$ is the maximal Fe : P uptake ratio. In principle, $R_0^{\mathrm{Fe:P}}$ and $k_{\mathrm{Fe:P}}$ could be different for different functional classes. However, we find that when optimized they tend to be nearly equal for different classes, so that we chose for simplicity to use the same values of $R_0^{\mathrm{Fe:P}}$ and $k_{\mathrm{Fe:P}}$ for all classes. As noted by Galbraith et al. (2010), this formulation ignores the effects of light limitation suggested by several studies (e.g., Sunda and Huntsman, 1997; Strzepek et al., 2012). Equation (14) also does not encode a minimum iron requirement. When we introduced such a minimum it tended to be optimized to zero and we therefore ignore it for simplicity. The Monod formulation (14) does capture luxury iron uptake (e.g., Marchetti et al., 2009a) when the half-saturation constant of (14) exceeds the half-saturation constant of the iron limitation in (11), as made explicit by Galbraith et al. (2010). This is the case for our optimized value of $k_{\mathrm{Fe:P}}$ so that phytoplankton has the luxury to increase its iron uptake with increasing dFe concentration even when iron is not limiting.

Our representation of the $R^{\mathrm{Si:P}}$ uptake ratio takes into consideration field studies and iron enrichment experiments, which have indicated that in HNLC regions and upwelling regions iron limitation leads to increased diatom silification, i.e., increased cellular Si : N and Si : P ratios (e.g., Takeda, 1998; Hutchins and Bruland, 1998; Franck et al., 2000; Brzezinski et al., 2003). However, there is no literature consensus on a mechanistic formulation of the iron dependence of silicic-acid uptake. For example, Matsumoto et al. (2013) assume a Si : N uptake ratio inversely proportional to the dFe concentration (capped at a minimum), while Jin et al. (2006) assume the Si : N ratio to depend only on the $\mathrm{Si(OH)_4}$ concentration. Others suggest that the dFe concentration only impacts the diatom growth rate and not the cellular Si : C ratio, while the $\mathrm{Si(OH)_4}$ concentration impacts the cellular Si : C ratio and not growth rate (e.g., Marchetti et al., 2009b; Brzezinski et al., 2011a). Here, we chose to retain the effects of increased silification due to iron limitation and the impact of high $\mathrm{Si(OH)_4}$ concentration on silification (Brzezinski 2016, personal communication). We model these effects with the formulation

$$R^{\mathrm{Si:P}} = R_0^{\mathrm{Si}} + \left(R_\mathrm{m}^{\mathrm{Si}} - R_0^{\mathrm{Si}}\right)\frac{k_{\mathrm{Si:P}}^{\mathrm{Fe}}}{\chi_{\mathrm{Fe}} + k_{\mathrm{Si:P}}^{\mathrm{Fe}}}\frac{\chi_{\mathrm{Si}}}{\chi_{\mathrm{Si}} + k_{\mathrm{Si:P}}^{\mathrm{Si}}}\quad. \tag{15}$$





The ratio involving the $\chi_{\text{Fe}}$ produces increased silification when iron is deficient, while the Monod term for $\chi_{\text{Si}}$ produces increased silification in silicon-replete environments: If $\chi_{\text{Fe}} \to 0$ and $\chi_{\text{Si}} \gg k_{\text{Si:P}}^{\text{Si}}$, then $R^{\text{Si:P}} \to R_{\text{m}}^{\text{Si}}$, while if $\chi_{\text{Fe}} \gg k_{\text{Si:P}}^{\text{Fe}}$ or $\chi_{\text{Si}} \to 0$, then $R^{\text{Si:P}} \to R_0^{\text{Si}}$. The minimum and maximum Si : P ratios $R_0^{\text{Si}}$ and $R_{\text{m}}^{\text{Si}}$, as well as the constants $k_{\text{Si:P}}^{\text{Fe}}$ and $k_{\text{Si:P}}^{\text{Si}}$ were tuned rather than fully optimized to achieve the observation-based fractional uptake of each functional class (see section 3.3 on optimization for details).

### 2.4 Iron model

#### 2.4.1 Iron sources

The aeolian source, $s_{\text{A}}$, is based on the spatial pattern of the surface flux of atmospheric soluble iron of Luo et al. (2008), obtained from an atmospheric model for current climate conditions that includes size-partitioned mineral dust, biomass burning, and industrial emissions. Because the global strength $\sigma_A$ of the aeolian source is highly uncertain (e.g., Tagliabue et al., 2016), we scale the global amplitude of this pattern to an initial guess of the global source strength that is then refined in our final optimization step (see sections 3.3 and 3.4 below). We note that the model of Luo et al. (2008) estimate a soluble aeolian iron flux into the ocean of $\sigma_A \sim 6\,\text{Gmol}\,\text{yr}^{-1}$.

The sedimentary source, $s_{\text{S}}$, has the pattern of the POP flux reaching the sediments (Elrod et al., 2004; Frants et al., 2016) and accounts for both resolved and subgrid topography. The amplitude of this pattern is the global sediment iron source strength $\sigma_S$, which is an optimized parameter. The dependence of the sediment redox reaction on dissolved oxygen (e.g., Galbraith et al., 2010) is ignored here for simplicity and to avoid carrying oxygen as another tracer. Unlike in the model of Frants et al. (2016), the phosphorus cycle and POP flux are not prescribed but coupled to the iron and silicon cycles as described above.

To model the hydrothermal source, $s_{\text{H}}$, we use the $^3$He source pattern of the OCMIP protocol (Dutay et al., 2004), and jointly optimize the hydrothermal iron source strengths $\sigma_{\text{H,ATL}}$, $\sigma_{\text{H,PAC}}$, $\sigma_{\text{H,IND}}$, and $\sigma_{\text{H,SO}}$ of the Atlantic, Pacific, Indian, and Southern Ocean ridge systems, as in the work of Frants et al. (2016).

#### 2.4.2 Iron sinks

Dissolved iron can be chelated by ligands or "free". We assume that the scavenging acts only on the concentration $\chi_{\text{Fe}'}$ of free iron so that chelation by ligands protects dFe from being scavenged. Scavenging is modelled as a first order process (e.g., Aumont et al., 2015) so that the scavenging rate is proportional to the product of $\chi_{\text{Fe}'}$ and the concentration of the scavenging particles $\chi^j$, for $j \in \{\text{POP}, \text{bSi}, \text{dst}\}$, the three types of particles considered. For each particle type, the scavenging rate per unit volume is thus modelled as

$$J_j = \kappa_{\text{scv}}^j \chi^j \chi_{\text{Fe}'} \quad , \tag{16}$$

where the scavenging rate constants $\kappa_{\text{scv}}^{\text{POP}}$, $\kappa_{\text{scv}}^{\text{bSi}}$, or $\kappa_{\text{scv}}^{\text{dst}}$ are optimizable parameters.



To compute the concentration of the scavenging particles, we use the fact that the flux divergences generated by the biogenic transport operators must be balanced by local remineralization or dissolution rates, that is,

$$\mathcal{B}^{\mathrm{P}} \sum_c f_c U_c = \kappa_{\mathrm{P}} \chi^{\mathrm{POP}} \tag{17}$$

and

$$\mathcal{B}^{\mathrm{Si}} f_{\mathrm{dia}} U_{\mathrm{dia}} = \kappa_{\mathrm{Si}}^{\mathrm{max}} e^{-T_E/T} \chi^{\mathrm{bSi}} \quad . \tag{18}$$

Although we use the nominal values of $\kappa_{\mathrm{P}}$ and $\kappa_{\mathrm{Si}}^{\mathrm{max}}$ listed in Table 1, note that these constants only enter the scavenging rates (16) through the combinations $\kappa_{\mathrm{scv}}^{\mathrm{POP}}/\kappa_{\mathrm{P}}$ and $\kappa_{\mathrm{scv}}^{\mathrm{bSi}}/\kappa_{\mathrm{Si}}$, where $\kappa_{\mathrm{scv}}^{\mathrm{POP}}$ and $\kappa_{\mathrm{scv}}^{\mathrm{bSi}}$ are optimized. The concentration of dust particles is modelled as vertically uniform due to sinking dust particles that do not dissolve or re-suspend from sediments (e.g., Moore and Braucher, 2008). We use the geographic pattern of the dust mass flux into the ocean provided by Luo et al. (2008), which we convert to a particle concentration using a nominal sinking speed of $w_{\mathrm{dst}} = 50 \ \mathrm{m \ day}^{-1}$. The exact value of $w_{\mathrm{dst}}$ does not matter because the dust scavenging rate depends only on $\kappa_{\mathrm{scv}}^{\mathrm{dst}}/w_{\mathrm{dst}}$ and $\kappa_{\mathrm{scv}}^{\mathrm{dst}}$ is optimized.

The key control on shaping the free iron concentration, and hence the scavenging, is the ligand concentration $L$. Chemical equilibrium between ligands, total dFe, and free iron determines $\chi_{\mathrm{Fe}'}$ as a quadratic function of the (total) dFe concentration (see, e.g., Frants et al., 2016). We used the same ligand stability constant of $K_L = 8 \times 10^{10} \ \mathrm{kg \ (mol \ Lig)}^{-1}$ as Frants et al. (2016). The ligand concentration itself is modelled to have a uniform background value $L_b$ that can be enhanced in old waters (Misumi et al., 2013) and in hydrothermal plumes (e.g., Bennett et al., 2008; Hawkes et al., 2013), similar to the formulation of Frants et al. (2016). Specifically, we use

$$L = \max(L_{\mathrm{H}} + L_{\mathrm{sw}}, L_{\mathrm{b}}) \quad , \tag{19}$$

where the $L_{\mathrm{H}}$ and $L_{\mathrm{sw}}$ are the hydrothermal and aged "sea water" ligand concentrations defined as follows. The hydrothermal ligand plumes are computed from the souce-sink balance

$$\mathcal{T} L_{\mathrm{H}} = -\frac{1}{\tau_{\mathrm{b}}} L_{\mathrm{H}} - \frac{\Delta_{\mathrm{v}}}{\tau_{\mathrm{v}}} (L_{\mathrm{H}} - L_{\mathrm{v}}), \tag{20}$$

where $\Delta_{\mathrm{v}}$ is a mask that is unity for grid boxes containing vent sites (taken from the OCMIP $^3\mathrm{He}$ source (Dutay et al., 2004)) and zero elsewhere. The timescale $\tau_{\mathrm{v}} = 1 \ \mathrm{s}$ clamps the ligand concentration to $L_{\mathrm{v}}$ at the vents, and the timescale $\tau_{\mathrm{b}}$ controls the plume spread. The ligand concentration $L_{\mathrm{sw}}$ is enhanced in old waters according to

$$L_{\mathrm{sw}}(\boldsymbol{r}) = \frac{\Gamma(\boldsymbol{r})}{\Gamma_{\mathrm{max}}} L_{\mathrm{max}} \quad , \tag{21}$$

where $\Gamma(\boldsymbol{r})$ is the ideal mean water age (easily computed for our model), we choose $\Gamma_{\mathrm{max}} = 1600 \ \mathrm{yr}$ following Frants et al. (2016), and $L_{\mathrm{max}}$ together with $\tau_{\mathrm{b}}$, $L_{\mathrm{v}}$, and $L_{\mathrm{b}}$ are optimizable parameters.

As is the case for most iron models, there is no need to explicitly represent the chemical precipitation of dFe. This is because in most formulations the scavenging rates increase rapidly when dFe exceeds a certain threshold. For our model this threshold is set by the ligand concentration $L$: in chemical equilibrium $\chi_{\mathrm{Fe}'}$ and hence the scavenging rate rise rapidly when dFe concentrations exceed $L$.



## 3 Numerical method, parameter optimization, and family of solutions

### 3.1 Steady state solution

All three-dimensional fields (e.g., the concentrations $\chi_i$) are discretized on our model grid and organized into column vectors (length $n = 191\,169$ at our resolution). Linear operators such as $\mathcal{T}$, $\mathcal{S}_c^i$, and $\mathcal{S}^{\mathrm{s},j}$ are correspondingly organized into $n \times n$ sparse matrices. The steady-state tracer equations (1)–(3) then become a system of $3n \times 3n$ equations that are nonlinear because of the iron scavenging and the colimitation of the $PO_4$ uptake.

The $3n \times 3n$ system is solved efficiently using Newton's method (e.g., Kelley, 2003a, b). Convergence of the Newton method depends on the initial guess for the solution and is not guaranteed. For the initial guess of $\chi_P$ and $\chi_{Si}$ we use the annual mean fields of the World Ocean Atlas (WOA13, Garcia et al., 2014) interpolated to our grid, and for the initial guess of $\chi_{Fe}$ we use the dFe fields estimated by Frants et al. (2016). The Newton solver typically converges to numerical precision in $\sim$10 iterations.

### 3.2 Cost function

We optimize the model parameters by systematically minimizing a quadratic cost function of the mismatch between modelled and observed fields. For $PO_4$ and $Si(OH)_4$, for which gridded climatologies are available, we define the weights based on the grid-box volumes, organized into vector $\boldsymbol{v}$, as

$$\boldsymbol{w}_P = \frac{\boldsymbol{v}}{(\overline{\chi_P^{\mathrm{obs}}})^2 V} \quad , \quad \text{and} \quad \boldsymbol{w}_{Si} = \frac{\boldsymbol{v}}{(\overline{\chi_{Si}^{\mathrm{obs}}})^2 V} \quad , \tag{22}$$

where we have normalized the weights by the total ocean volume $V$ and the squared global mean observed concentrations. This non-dimensionalizes the quadratic cost terms and scales them to the same order of magnitude. For dFe, for which only sparse observations are available, we also define weights $\boldsymbol{w}_{Fe}$ based on grid box volumes, but observations that are part of a vertical profile receive additional weight as detailed in Appendix C.

With diagonal weight matrix $\mathbf{W}_i = \mathbf{diag}(\boldsymbol{w}_i)$ for the $i^{\mathrm{th}}$ nutrient, its cost for the mismatch with observations is then given by

$$E_i = \delta\boldsymbol{\chi}_i^{\mathsf{T}} \mathbf{W}_i \, \delta\boldsymbol{\chi}_i \quad , \tag{23}$$

where $\delta\boldsymbol{\chi}_i \equiv \boldsymbol{\chi}_i - \boldsymbol{\chi}_i^{\mathrm{obs}}$. For $\chi_P^{\mathrm{obs}}$ and $\chi_{Si}^{\mathrm{obs}}$ we use WOA13 fields interpolated to our grid, and for $\chi_{Fe}^{\mathrm{obs}}$ we used the GEO-TRACES intermediate data product (Mawji et al., 2015) and the data set compiled by Tagliabue et al. (2012).

The cost terms for the nutrient mismatch do not provide a strong constraint on the relative sizes of the phytoplankton class because the nutrients are determined by their combined export. We therefore include additional terms in our cost function that constrain the phytoplankton concentrations $p_c$ to the recent satellite derived estimates of Kostadinov et al. (2009). These estimates provide phytoplankton concentrations for picophytoplankton (0.5–2 μm in diameter), nanophytoplankton (2–20 μm), and microphytoplankton (20–50 μm), which we identify with our small, large, and diatom functional classes. We use the entire mission composite data set as the satellite climatology (Kostadinov et al., 2016).





Because of the large dynamic range of the phytoplankton concentrations, we consider mismatches in the log of the concentrations, that is, $\delta\pi_c \equiv \log[(p_c + \epsilon_c)/p_0] - \log[(p_c^{\mathrm{obs}} + \epsilon_c)/p_0]$ , where $\epsilon_c = \overline{p_c^{\mathrm{obs}}}$ is introduced to limit the logarithm where the phytoplankton concentration falls to zero. For each class, we construct normalized weight vectors

$$\boldsymbol{w}_c = \frac{\boldsymbol{v}}{[\log(\overline{p_c^{\mathrm{obs}}}/p_0)]^2 V_{\mathrm{eup}}} \quad , \tag{24}$$

where $V_{\mathrm{eup}}$ is the global euphotic volume, and $p_0 = 1\,\mathrm{mg\,C\,m^{-3}}$ nondimensionalizes the argument of the logarithm.

Organizing mismatches and weights into vectors and diagonal matrices, we calculate the cost for the phytoplankton concentration mismatch as

$$E_{\mathrm{plk}} = \sum_c \delta\boldsymbol{\pi}_c^{\mathsf{T}} \mathbf{W}_c \delta\boldsymbol{\pi}_c \quad , \tag{25}$$

and combine the costs for the nutrient and plankton mismatches into the total cost

$$E_{\mathrm{tot}} = \omega_{\mathrm{P}}\, E_{\mathrm{P}} + \omega_{\mathrm{Si}}\, E_{\mathrm{Si}} + \omega_{\mathrm{Fe}}\, E_{\mathrm{Fe}} + \omega_{\mathrm{plk}}\, E_{\mathrm{plk}} \quad , \tag{26}$$

which we minimize to constrain our model parameters by the available observations. In (26) the $\omega$ weights were chosen such that the four cost terms contribute roughly equally to the total cost for a typical member of our family of solutions. This was achieved with $(\omega_{\mathrm{P}}, \omega_{\mathrm{Si}}, \omega_{\mathrm{Fe}}, \omega_{\mathrm{plk}}) = (1, 0.47, 0.044, 0.30)$, the smaller weight for dFe reflecting its larger root-mean-square (RMS) mismatch and hence much larger cost $E_{\mathrm{Fe}}$.

## 3.3 Optimization strategy

Our model has ∼50 biogeochemical parameters that can in principle be determined through objective optimization given appropriate observational data. However, even with perfect data, some parameters can compensate for others (e.g., two parameters appearing as a ratio) so that not all parameters are independent. Other parameters cannot be optimized because the mismatch with available nutrient and phytoplankton data is not sensitive to their value. In practice, it therefore is not possible to optimize all parameters, and care is needed to optimize only those parameters that independently shape the nutrient and phytoplankton concentrations.

The parameters associated with the remineralization of phosphate and dissolution of opal are well constrained by the high-quality climatologies of $\mathrm{PO_4}$ and $\mathrm{Si(OH)_4}$. However, the iron cycle is relatively poorly constrained because the dFe data is much more sparse in both time and space, and estimates of the iron sources range over two orders of magnitude (e.g., Tagliabue et al., 2016). Moreover, the ligand field that determines the scavengable free iron is highly uncertain. Given these challenges, the recent inverse model of the iron cycle by Frants et al. (2016) considered a family of solutions for a range of external source strengths, an approach we will follow here for our coupled model.

Another key consideration is computational cost. Even with the numerically efficient Newton Solver, optimization typically requires hundreds of solutions of equations (1)–(3) per optimized parameter. We therefore optimized no more than ∼13 parameters at a time. We acknowledge that the minimum attained by sequentially optimizing groups of independent parameters





**Table 1.** Parameters that were prescribed from the literature, or that were separately optimized in a submodel.

| Parameter | Description | Value | Unit | Source |
|---|---|---|---|---|
| $\kappa$ | Growth and mortality temperature coefficient | 0.063 | $(^{\circ}\mathrm{C})^{-1}$ | Eppley (1972) |
| $k_{\mathrm{w}}$ | Irradiance attenuation coefficient | 0.040 | $\mathrm{m}^{-1}$ | Dutkiewicz et al. (2005) |
| $K_L$ | Ligand stability constant | 80000. | $\mathrm{kg\,(mol\,Lig)}^{-1}$ | Galbraith et al. (2010) |
| $\Gamma_{\mathrm{max}}$ | Age coefficient for ligand parameterization | 1600. | yr | Frants et al. (2016) |
| $f^{\mathrm{POP}}$ | Recyclable fraction of POP-scavenged dFe | 0.90 | – | Moore and Braucher (2008) |
| $f^{\mathrm{bSi}}$ | Recyclable fraction of opal-scavenged dFe | 0.90 | – | Moore and Braucher (2008) |
| $f^0_{\mathrm{dia}}$ | Diatom class detrital fraction at $0^{\circ}\mathrm{C}$ | 0.74 | – | Dunne et al. (2005) |
| $f^0_{\mathrm{lrg}}$ | Large class detrital fraction at $0^{\circ}\mathrm{C}$ | 0.74 | – | Dunne et al. (2005) |
| $f^0_{\mathrm{sml}}$ | Small class detrital fraction at $0^{\circ}\mathrm{C}$ | 0.14 | – | Dunne et al. (2005) |
| $\kappa_{\mathrm{P}}$ | POP remineralization rate constant | 0.03 | $\mathrm{d}^{-1}$ | Kriest and Oschlies (2008) |
| $\kappa_{\mathrm{Si}}^{\mathrm{max}}$ | Opal dissolution rate coefficient | $13 \times 10^{15}$ | $\mathrm{d}^{-1}$ | Gnanadesikan (1999) |
| $T_{\mathrm{E}}$ | Temperature scale for opal dissolution | 11481. | K | Gnanadesikan (1999) |
| $k_{\mathrm{dia}}^{\mathrm{Si}}$ | Diatom class $\mathrm{Si(OH)_4}$ half-saturation constant | 1.0 | $\mathrm{mmol\,Si\,m}^{-3}$ | Matsumoto et al. (2013) |
| $w_{\mathrm{Si}}$ | Opal sinking speed | 40. | $\mathrm{m\,d}^{-1}$ | Submodel optimization |
| $b$ | POP flux Martin exponent | 0.82 | – | Submodel optimization |

is generally different than jointly optimizing all independent parameters, but computational and practical considerations demanded a sequential approach. We justify this a posteriori by the fact that we are able to achieve state-of-the-art fits to the nutrient fields. Given these considerations, we adopted the following strategy:

(i) Parameters that are measurable and considered well-known, as well as parameters that are unconstrainable by our cost

function or whose value is not critical because they are strongly compensated by other parameters, were assigned values from the literature as collected in Table 1. The considerations that entered our choice of prescribed parameters are detailed in Appendix D.

(ii) The parameters that set the phosphate remineralization and opal dissolution profiles were optimized by minimizing the mismatch with $\mathrm{PO_4}$ and $\mathrm{Si(OH)_4}$ concentration data from the WOA13 using separate single-nutrient models. For the Si cycle,

we used the model of Holzer et al. (2014) and verified that the opal sinking speed parameter $w_{\mathrm{Si}}$ was not affected by the inclusion of sub-grid topography (Appendix B). For the P cycle, we used a similar conditional restoring model without POP, but with subgrid topography, and optimized the Martin exponent $b$. The resulting values of $w_{\mathrm{Si}}$ and $b$ (Table 1) were held fixed for all optimizations of the coupled nutrient cycling model.

(iii) The remaining parameters were optimized using our coupled model. We first assign initial values for all these parameters

and then sequentially update these initial values by optimizing subsets of parameters as detailed in Appendix D. Both initial




**Table 2.** Optimized parameters and range across family of solutions.

| Parameter | Description | Initial Value | Optimized Value | Range | Unit |
|---|---|---|---|---|---|
| $k_{I,\text{dia}}$ | Diatom class irradiance half-saturation rate | 20. | 8.1 | – | $\text{W m}^{-2}$ |
| $k_{I,\text{lrg}}$ | Large class irradiance half-saturation rate | 20. | 9.0 | – | $\text{W m}^{-2}$ |
| $k_{I,\text{sml}}$ | Small class irradiance half-saturation rate | 20. | 8.8 | – | $\text{W m}^{-2}$ |
| $R_m^{\text{Si}}$ | Diatom maximum Si : P | 160. | 220. | – | $\text{mol Si} \, (\text{mol P})^{-1}$ |
| $R_0^{\text{Si}}$ | Diatom minimum Si : P | 8.0 | 13. | – | $\text{mol Si} \, (\text{mol P})^{-1}$ |
| $k_{\text{Si:P}}^{\text{Si}}$ | Silicon half-saturation constant in Si : P | 30. | 4.0 | – | $\text{mmol Si m}^{-3}$ |
| $k_{\text{Si:P}}^{\text{Fe}}$ | Iron hyperbolic constant in Si : P | 1.0 | 0.077 | – | nM Fe |
| $k_{\text{Fe:P}}$ | Iron half-saturation constant in Fe : P | 0.74 | 0.74 | – | nM Fe |
| $k_{\text{dia}}^{\text{P}}$ | Diatom class $PO_4$ half-saturation constant | 0.39 | 0.72 | – | $\text{mmol P m}^{-3}$ |
| $k_{\text{lrg}}^{\text{P}}$ | Large class $PO_4$ half-saturation constant | 0.39 | 0.72 | – | $\text{mmol P m}^{-3}$ |
| $k_{\text{sml}}^{\text{P}}$ | Small class $PO_4$ half-saturation constant | 0.030 | 0.13 | – | $\text{mmol P m}^{-3}$ |
| $k_{\text{dia}}^{\text{Fe}}$ | Diatom class dFe half-saturation constant | 0.10 | 0.30 | – | nM Fe |
| $k_{\text{lrg}}^{\text{Fe}}$ | Large class dFe half-saturation constant | 0.10 | 0.29 | – | nM Fe |
| $k_{\text{sml}}^{\text{Fe}}$ | Small class dFe half-saturation constant | 0.010 | 0.11 | – | nM Fe |
| $p_{\text{dia}}^{\text{max}}$ | Diatom class maximum concentration | 23. | 42. | – | $\text{mg C m}^{-3}$ |
| $p_{\text{lrg}}^{\text{max}}$ | Large class maximum concentration | 23. | 61. | – | $\text{mg C m}^{-3}$ |
| $p_{\text{sml}}^{\text{max}}$ | Small class maximum concentration | 23. | 21. | – | $\text{mg C m}^{-3}$ |
| $\tau_{\text{dia}}$ | Maximal Diatom growth timescale | 6.0 | 0.65 | – | d |
| $\tau_{\text{lrg}}$ | Maximal Large growth timescale | 6.0 | 1.5 | – | d |
| $\tau_{\text{sml}}$ | Maximal Small growth timescale | 6.0 | 7.4 | – | d |
| $R_0^{\text{Fe:P}}$ | Maximum Fe : P uptake ratio | 5.0 | 2.0 | 0.00047–3.0 | $\text{mmol Fe} \, (\text{mol P})^{-1}$ |
| $\kappa_{\text{scv}}^{\text{POP}}$ | POP scavenging rate constant | 0.13 | 1.0 | 0.015–7.9 | $(\text{mmol POP m}^{-3})^{-1} \, \text{d}^{-1}$ |
| $\kappa_{\text{scv}}^{\text{bSi}}$ | Opal scavenging rate constant | 3.1 | 1.3 | 0.85–13. | $(\text{mol bSi m}^{-3})^{-1} \, \text{d}^{-1}$ |
| $\kappa_{\text{scv}}^{\text{dst}}$ | Dust scavenging rate constant | 10000. | 9.4 | 8.5–10. | $(\text{g dust m}^{-3})^{-1} \, \text{d}^{-1}$ |
| $L_{\text{b}}$ | Background ligand concentration | 1.0 | 0.51 | 0.40–0.72 | nM Lig |
| $L_{\text{v}}$ | Maximal hydrothermal vent ligand conc. | 3.0 | 1.2 | 0.68–1.4 | nM Lig |
| $\tau_{\text{b}}$ | Hydrothermal vent plume restoring timescale | 10. | 5.7 | 3.0–7.5 | yr |
| $L_{\text{max}}$ | Maximal age-enhanced ligand conc. | 2.3 | 0.97 | 0.82–1.3 | nM Lig |
| $\sigma_{\text{A}}$ | Aeolian source strength | 1.9 | 5.3 | 0.63–22. | $\text{Gmol Fe yr}^{-1}$ |
| $\sigma_{\text{S}}$ | Sedimentary source strength | 4.2 | 1.7 | 0.11–22. | $\text{Gmol Fe yr}^{-1}$ |
| $\sigma_{\text{H,ATL}}$ | Hydrothermal source strength, Atlantic | 0.098 | 0.19 | 0.00013–0.50 | $\text{Gmol Fe yr}^{-1}$ |
| $\sigma_{\text{H,PAC}}$ | Hydrothermal source strength, Pacific | 0.21 | 0.42 | 0.035–2.9 | $\text{Gmol Fe yr}^{-1}$ |
| $\sigma_{\text{H,IND}}$ | Hydrothermal source strength, Indian O. | 0.066 | 0.13 | 0.011–0.81 | $\text{Gmol Fe yr}^{-1}$ |
| $\sigma_{\text{H,SO}}$ | Hydrothermal source strength, Southern O. | 0.066 | 0.13 | 0.011–1.2 | $\text{Gmol Fe yr}^{-1}$ |





### 3.4 Family of solutions

Figure 1 shows the quality of the fit to nutrient and phytoplankton data for all our optimized solutions, which span a wide range of source strengths. For ease of presentation, solutions are divided at $\sigma_H = 1 \, \mathrm{Gmol \, yr^{-1}}$ into low and high hydrothermal cases, with $\sigma_H$ spanning a range from 0.073 to 11. $\mathrm{Gmol \, yr^{-1}}$. For high $\sigma_H$, we focused on correspondingly higher aeolian and sedimentary source regimes. Source-parameter space was not explored uniformly because (i) the final step of our optimization adjusted our initial choice of sources, and because (ii) some source choices produced spurious numerical difficulties for the Newton solver.

All solutions fit the macronutrient fields about equally well, but the overall quality of fit as quantified by the square root of the quadratic mismatch ("total cost", top panels of Figure 1) gets systematically worse with increasing aeolian source strength, $\sigma_A$, especially for high hydrothermal sources. This worsening fit for high $\sigma_A$ is reflected in the mismatch of all three nutrients. We define our family of solutions as the set of solutions whose total cost remains within ∼5% of the smallest misfit (total cost less than 15.4), which essentially eliminates solutions with $\sigma_A \gtrsim 22 \, \mathrm{Gmol \, yr^{-1}}$ and $\sigma_H \gtrsim 5 \, \mathrm{Gmol \, yr^{-1}}$ (black crosses in Figure 1). (If we include the "crossed-out" solutions for plots of subsequent sections that show scatter across the family of solutions, the visual impact is virtually imperceptible.) While it is clear from Figure 1 that high-$\sigma_A$ solutions are less likely, we hasten to add that the cost threshold for inclusion in the family is arbitrary as we do not have a formal error covariance to convert the cost into a likelihood.

In terms of total cost, there is little sensitivity to the strength of the sedimentary source – scavenging can be optimized for a sedimentary source ranging over 2 orders of magnitude for an overall similar quality of fit. For low $\sigma_H$, there are small opposing RMS mismatches for $PO_4$ and dFe, with a slightly better $PO_4$ fit for higher sedimentary source and a slightly better dFe fit for lower sedimentary source, although the variation in the mismatch is less that 1% of the global mean concentrations.

While the mismatch for dFe is substantial at ∼45% of the global mean dFe concentration, the smallest dFe mismatch occurs when all three sources are low. The dFe mismatch rapidly increases with $\sigma_A$, consistent with the findings of the much simpler model of Frants et al. (2016). The overall cost and the mismatch for each nutrient are insensitive to the strength of the hydrothermal source.

While Figure 1 shows some variations with the source strengths in the overall quality of the fit, it is clear that the iron sources and scavenging sinks are poorly constrained by the available nutrient and phytoplankton observational data. Given the uncertainties in the sources and the small cost differential between family members, it is not appropriate to single out the solution with the numerically lowest cost as the most realistic solution. We therefore use the entire family of solutions below to assess the robustness of our results in terms of the spread across the family, and to elucidate the systematic variations of the carbon and opal exports with the fractional size of each iron source type (aeolian, sedimentary, hydrothermal).





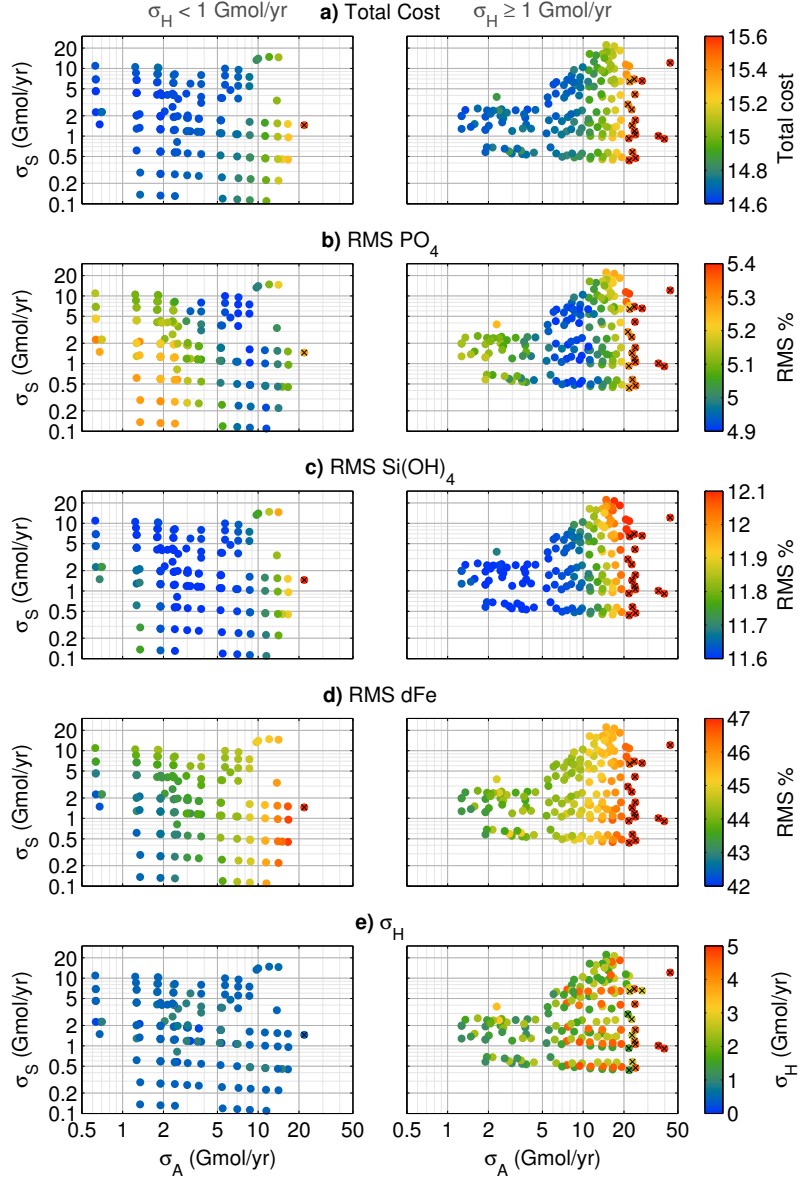

**Figure 1.** Total cost metric and RMS mismatch of the nutrient concentrations as a function of the aeolian, hydrothermal, and sedimentary iron source strengths ($\sigma_A, \sigma_S, \sigma_H$) plotted for all our optimized solutions. Solutions whose total cost exceed 15.4 are indicated by black crosses and were excluded from our family of solutions. Plots on the left show solutions for which $\sigma_H < 1\,\mathrm{Gmol\,yr^{-1}}$, while for plots on the right $\sigma_H \geq 1\,\mathrm{Gmol\,yr^{-1}}$. (a) Square root of the total cost expressed as a nominal percentage representative of the mean RMS mismatch of the nutrient and phytoplankton concentrations. (b) RMS mismatch of the $PO_4$ concentration as a percentage of the global mean $PO_4$ concentration. (c) As (b) for $Si(OH)_4$. (d) As (b) for dFe. (e) The value of the hydrothermal source $\sigma_H$ for each family member.



As a typical representative of our family of solutions, for which we plot patterns and typical results below, we selected the state for $(\sigma_A, \sigma_S, \sigma_H) = (5.3, 1.7, 0.9)\,\mathrm{Gmol\,yr^{-1}}$. This state is typical in that it lies at the mode of the distribution of overall RMS misfit values and, for most quantities, tends to lie in the middle of the range across the family.

We emphasize that the variations across the family of solutions explored here are variations of the *fully optimized* bio-geochemical states. These variations cannot be used to infer the system's response to dFe perturbations for which the other biogeochemical parameters would not change. Such perturbations, which are of great interest in themselves, are beyond the scope of this paper and will be examined in a separate publication.

## 4 Fidelity to observations

We now examine in more detail how well our solutions match the observations against which they were optimized. Where there is little variation across the family, we focus on our typical solution. For iron-related quantities that have by construction significant spread across the family, we will focus on the systematic variations of the optimized states with the dFe sources.

### 4.1 Nutrient concentrations

The nutrient concentrations are well constrained for all members of our family of solutions. We quantify the overall fit of the modelled nutrient concentrations in terms of the joint probability density function (pdf) of the modelled and observed concentrations. This joint pdf may be thought of as the binned scatter plot of the modelled versus observed values for all grid boxes. The binning for a given nutrient was weighted by the corresponding weights of the associated mismatch cost. These joint pdfs are shown in Figure 2 for all three nutrients for our typical solution. Both the $PO_4$ and $Si(OH)_4$ pdfs fall close to the 1:1 line, showing high fidelity to observations. For $PO_4$ the cost-weighted RMS error is $5.0\,\%$ of its global mean of $2.17\,\mu M$.

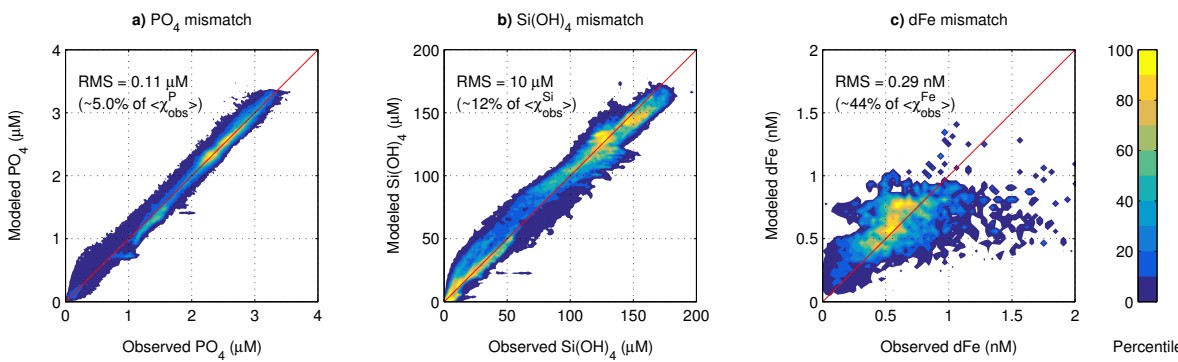

**Figure 2.** Joint distribution of the volume-weighted observed and modelled concentrations of $PO_4$ (a), $Si(OH)_4$ (b), and dFe (c). The percentiles of the cumulative distribution are defined such that $x\,\%$ of the distribution lies outside the $x$-percentile contour. Large percentiles thus correspond to high densities. For $PO_4$ and $Si(OH)_4$, the WOA13 observations were interpolated to our model grid. The dFe observations were interpolated to our model grid from the data compilation of Tagliabue et al. (2012) and the GEOTRACES data (Mawji et al., 2015).





In comparison, Primeau et al. (2013) achieved an RMS mismatch of $3\,\%$ by jointly optimizing the uptake rate of each grid box with the circulation. Silicic acid has a slightly larger RMS mismatch of $12.\,\%$ relative to its global mean $89.1\,\mu M$. This is similar to the $13\,\%$ RMS error reported by Holzer et al. (2014), who used the same circulation but a much simpler model of the silicon cycle.

The global mean $dFe$ concentration is well constrained within the narrow range of $0.56 - 0.68$ nM across the family of solutions. For iron, the joint probability is by necessity computed using only those grid boxes that contain $dFe$ observations. The scatter from the 1:1 line is much larger than for the macronutrients with a substantial RMS mismatch of 0.29 nM, or $44\,\%$ of the mean. This mismatch is comparable to that of other models (e.g., Tagliabue et al., 2016). Compared to the simpler model of Frants et al. (2016), the joint pdf shows that our $dFe$ field has a wider, more realistic dynamic range. We note that while

Frants et al. (2016) report an RMS mismatch of only 0.19 nM, they also employed different weights for the model-observation mismatch. If we re-compute the RMS mismatch of the optimized $dFe$ field of Frants et al. (2016) using the weights of this work, we also obtain a 0.29 nM mismatch.

The relatively large mismatch for $dFe$ not only quantifies model deficiencies, but to a large degree also reflects the fact that we are comparing snapshot observations against a steady-state climatological model. The $dFe$ observations have difficult-to-

quantify temporal and spatial sampling biases, and $dFe$ being a trace element it is sensitive to episodic events in the aeolian source (e.g., Croot et al., 2004), and possibly to internal episodic events such as submarine volcanism (e.g. Massoth et al., 1995).

## 4.2 dFe Profiles

To quantify the spatial structure of the $dFe$ mismatch, we examine vertical profiles for each basin. For both model and ob-

servations, we only use the grid boxes that contain observations and average horizontally over the basins using the $E_{Fe}$ cost weights. The resulting profiles are shown in Figure 3. The family of model profiles generally overlaps with the observational uncertainties. The estimates are particularly close to the observations near the surface. In the abyssal oceans, the spread in the family of profiles is larger. The spread across the family is in part a reflection of the weights in our cost function. Most $dFe$ observations are available in the upper ocean, implying a small variance of the mean concentration and hence large weights,

while deep observations tend to be sparser with smaller weights (for details on the weights see Appendix C).

Figure 3 also shows systematic biases in the inferred $dFe$ concentrations. Biases are particularly strong in the Pacific where the observations tend to be underpredicted by as much as $\sim 0.3\,nM$ above $\sim 1500\,m$ and overpredicted by $\sim 0.2\,nM$ below $\sim 2000\,m$ depth. The typical estimated Pacific profiles is too linear in the upper 1500 m, with vertical gradients that are too weak above $\sim 300\,m$ and too strong below $\sim 1000\,m$. In the Atlantic, a smaller low bias of $\sim 0.15\,nM$ can be seen between

$\sim 500$ and $\sim 1300\,m$ depth.

These biases could be due to deficiencies in our model such as, for example, oversimplified ligand parameterization, but one must also keep in mind that there are hard-to-quantify biases in the observations. The observations are too sparse to form a reliable climatology, and it is remarkable that we can fit the available observations as well as we do. The larger biases in

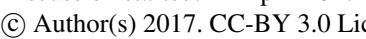



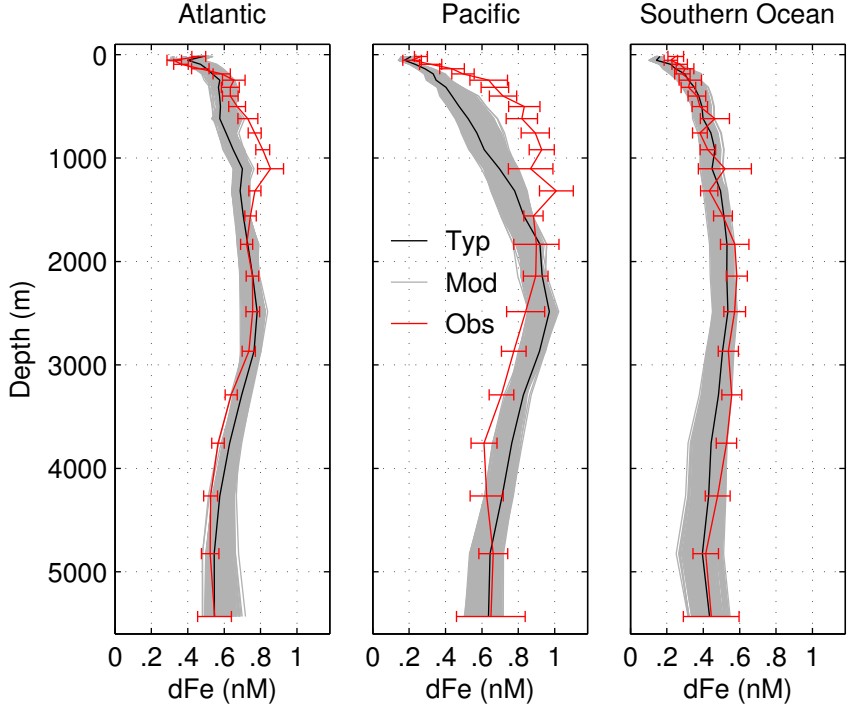

**Figure 3.** Basin-wide, cost-weighted average profiles of the (red) observed and (grey) modelled $dFe$ concentrations for the Atlantic and Pacific oceans (both north of $40°$S), and the Southern Ocean (south of $40°$S). The profiles of our typical solution are highlighted in black. The error bars represent the combined standard error associated with the spatial standard deviation from the basin-mean profile and the observational standard deviation for each grid box. These were added in weighted quadrature using the weights for $dFe$ mismatch from our cost function.

the Pacific could well be due to the absence of Pacific transects in the GEOTRACES Intermediate Data Product 2014, which means that mismatches in the Pacific incur a relatively smaller penalty in our cost function.

### 4.3 Phytoplankton distribution

Figure 4 shows a model–observation comparison of the phytoplankton concentration (plotted in $C$ units using a constant
5  Redfield ratio of $C : P = 106$). Although the distinction between our phytoplankton classes is functional and not determined by size, we compare our Small, Large, and Diatom concentrations with the picophytoplankton ($0.5$–$2 \, \mu m$), nanophytoplankton ($2$–$20 \, \mu m$), and microphytoplankton ($20$–$50 \, \mu m$) of the satellite-based estimates of Kostadinov et al. (2016), consistent with the construction of our $E_{plk}$ cost function.

The inverse-model estimates capture the broad global patterns of the phytoplankton concentrations reasonably well, although
10  some biases are also evident. The observation-based Diatom and Large concentration has a minimum at $\sim 60°$S, a feature our



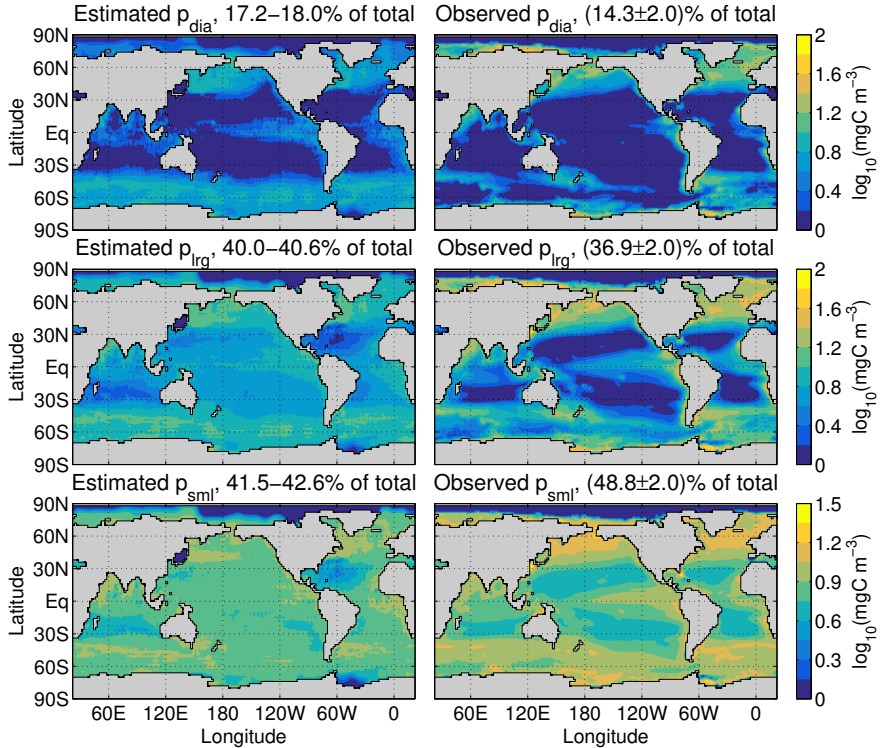

**Figure 4.** Comparison of estimated (left) and observed (right) phytoplankton concentrations averaged vertically over the model euphotic zone. The Diatom concentration is shown in the top row, the Large phytoplankton concentration in the middle row, and the Small phytoplankton concentration in the bottom row. Note the logarithmic colour scales with separate ranges for each functional class.

estimates do not capture. This may be related to seasonality issues, which our approach cannot capture. Our estimates for the Large and Small concentrations have higher concentrations in the subtropical gyres and weaker meridional gradients with lower high-latitude and tropical concentrations than observed. These discrepancies may again be related to the absence of seasonality. Another factor is that the phytoplankton mismatch carries less penalty in our cost than the combined misfit terms

5   of the three nutrient fields.

The global mean phytoplankton concentration of each functional class was remarkably stable across all members of our family of solutions with ranges of $\overline{p_{\mathrm{sml}}} = (2.63 - 2.87)\,\mathrm{mg\,C\,m^{-3}}$, $\overline{p_{\mathrm{lrg}}} = (6.16 - 6.53)\,\mathrm{mg\,C\,m^{-3}}$, and $\overline{p_{\mathrm{dia}}} = (2.63 - 2.87)$ $\mathrm{mg\,C\,m^{-3}}$. This indicates that the satellite data provides a good constraint on the global-scale ecosystem composition.

## 5   Limiting nutrients

10   For a given phytoplankton functional class, different nutrients are known to limit biological production in different parts of the ocean (e.g., Moore et al., 2001). These geographic limitation patterns are a fundamental fingerprint of upper-ocean ecosystem





dynamics. Knowledge of the limitation patterns is important for understanding how the global nutrient cycles operate in the current climate and for assessing possible future changes of the global ocean ecosystem.

Limiting nutrients can be determined observationally (e.g., Moore et al., 2013), and from biogeochemical models (e.g., Moore et al., 2004). Here, we estimate the limitation patterns from our optimized inverse-model solutions. In our model, the

biological uptake of each functional class (6) is limited through $F_{N,c}$, the product defined in (11) of three Monod terms, one for each nutrient. We define the deficiency $D_c^i$ of functional class $c$ in nutrient $i$ as the complement of the corresponding Monod factor, i.e., as $D_c^i \equiv 1 - \frac{\chi_i}{\chi_i + k_c^i}$. We deem nutrient $i$ to be "limiting" class $c$ if $D_c^i > 0.5$ or, equivalently, if $\chi_i < k_c^i$, i.e., if the nutrient concentration falls below its half-saturation value for uptake.

To display the pattern of the nutrient limitations, we could use the fact that we have three nutrients to define an RGB color

as $(D_c^P, D_c^{Si}, D_c^{Fe})$. However, because the resulting colors vary continuously, it is hard to quantify the resulting patterns. We therefore define the limiting RGB color as $(L_c^P, L_c^{Si}, L_c^{Fe})$, where $L_c^i = 1$ if $D_c^i > 0.5$ and $L_c^i = 0$ otherwise. This partitions the RGB color cube into 8 possible colors that define and identify the 8 nutrient limitation regimes shown in Figure 5. Specifically, the resulting colour is black $(0,0,0)$ if all nutrients are available in sufficient quantities so that none are deemed limiting, white $(1,1,1)$ if all three nutrients are limiting, red $(1,0,0)$ if only dFe is limiting, green $(0,1,0)$ if only $Si(OH)_4$ is limiting, and

blue $(0,0,1)$ if only $PO_4$ is limiting. The remaining three possibilities correspond to 2 nutrients being colimiting: magenta $(1,0,1)$ if dFe and $PO_4$ are colimiting, cyan $(1,1,0)$ if $PO_4$ and $Si(OH)_4$ are colimiting, and yellow $(0,1,1)$ if $Si(OH)_4$ and dFe are colimiting. Only a few grid boxes in the Arctic are solely limited by silicic acid (green).

Figure 5 shows the limitation patterns of all three phytoplankton classes. The Large and Diatom classes have similar patterns of iron limitation in the Southern Ocean, eastern tropical Pacific, and North Pacific. For both classes, the Indian Ocean and

North Atlantic are largely $PO_4$ limited. The subtropical gyres of the Indian Ocean and North Atlantic are $PO_4$ limited for the Large class, and $PO_4$–$Si(OH)_4$ colimited for Diatoms. The differences between the Large and Diatom classes come from the $Si(OH)_4$ dependency of diatoms. Because the Large class requires zero silicic acid, its limitation map show no areas where all three nutrients are limiting (white). The subtropical gyres of the Pacific and South Atlantic are dFe and $PO_4$ colimited for the Large class, while for diatoms the centre of these gyres are limited in all three nutrients. For diatoms, the eastern margins of the

Pacific subtropical gyres show Si–Fe colimitation (yellow). The completely nutrient replete regions of the Arctic and Weddell Sea reflect the low biological utilization there driven in our model by light limitation through the prescribed PAR irradiance field.

The small phytoplankton class shows a much simpler limitation pattern. Limitation occurs primarily in the subtropical oceans with small patches of iron limitation also in the Southern Ocean and tropical Pacific. Iron limitation dominates the subtropical

South Pacific, while $PO_4$ limitation occurs primarily in the subtropical gyres of the south Indian Ocean and North Atlantic. The rest of the ocean is largely nutrient replete for the Small functional class.

The broad features of the limitation pattern of Figure 5 are robust across all members of our family of solutions, with areas of each type of limitation generally varying by $\pm 5\%$ or less across the family of solutions.

The general features seen in Figure 5 broadly agree with the observational data (in situ and bottle nutrient addition ex-

periments) reported by Moore et al. (2013). Like our estimates, the observations show Fe limitation in the Southern Ocean,





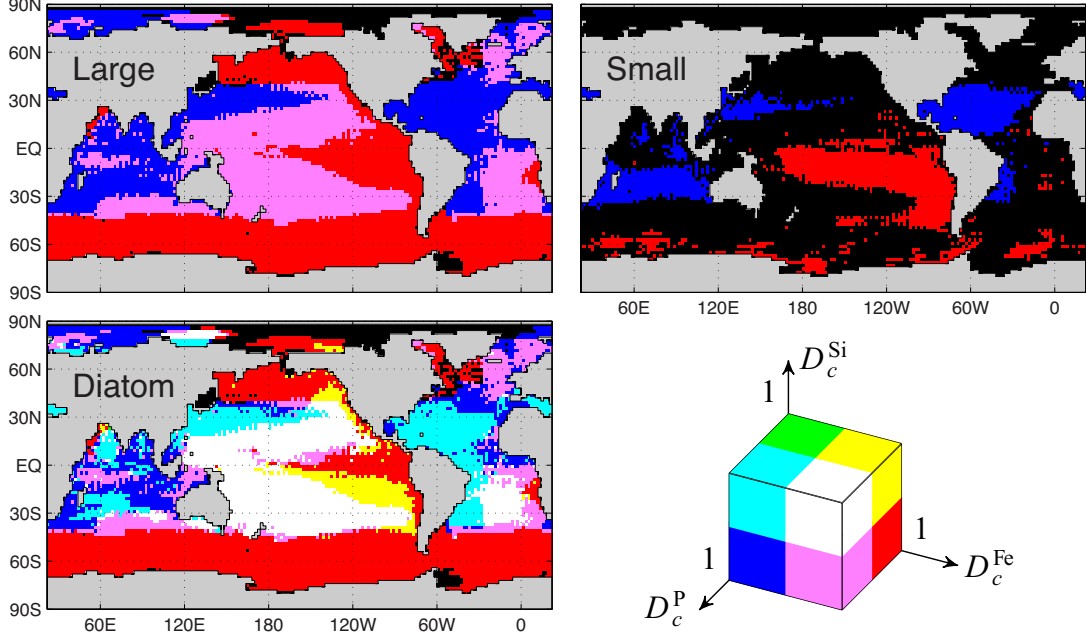

**Figure 5.** The patterns of limiting nutrients for each phytoplankton functional class. The colour cube at the bottom right shows the eight possible limitation regimes of our inverse model: Red corresponds to dFe limitation, blue to P limitation, and green to Si limitation. Cyan, yellow, and magenta correspond to colimitations of P and Si, dFe and Si, and P and dFe, respectively. White corresponds to colimitation of all three nutrients while black indicates no limitation. (See text for the definitions of the deficiencies $D_c^{\mathrm{P}}$, $D_c^{\mathrm{Si}}$, and $D_c^{\mathrm{Fe}}$ of the cube axes.)

subpolar North Pacific, and eastern tropical Pacific. The observations also indicate Fe limitation in the North Atlantic, which for our solutions is also present in small patches in the western subpolar North Atlantic and becomes slightly more pronounced for the family members with higher total iron source. Moore et al. (2013) report Si limitation in the Pacific sector of the Southern Ocean at its northern boundary, where the silicic acid concentration sharply decreases. This is consistent with our yellow region of joint Si and Fe limitation along the eastern edge of the Pacific subtropical gyres. Consistent with our estimates, the observations show $PO_4$ limitation in the North Atlantic subtropical gyre and in the equatorial Atlantic.

Our limitation patterns can also be compared to those calculated for summer conditions in the BEC model of Moore et al. (2004). However, it must be kept in mind that (i) the BEC model has a different circulation and a different representation of the ocean's biogeochemical cycles (e.g., explicitly representing the nitrogen cycle and diazotrophs) and that (ii) Moore et al. (2004) define limitation in terms of the minimum Monod factor, while we use a threshold of $1/2$ for the Monod factors and jointly consider three Monod terms to define the type of limitation. For diatoms, the Fe limitation pattern reported by Moore et al. (2004) is similar to ours including bands of Si limitation surrounding the tongue of Fe limitation in the tropical eastern Pacific. For non-diatom phytoplankton, there are also broad similarities, such as iron limitation in the eastern tropical Pacific, subpolar North Pacific, and Southern Ocean. In the BEC model, most of the Atlantic is phosphate or nitrate limited. While we do not



model nitrate, nitrate limitation in our estimates would be reflected as phosphate limitation, and our limitation patterns show most of the Atlantic deficient in phosphate. The BEC model's Small phytoplankton class shows nitrogen limitation surrounding the tropical tongue of iron limitation in the Pacific, while with our definitions there is very little $PO_4$ limitation in the Pacific for the Small class, which is iron limited or nutrient replete in most of the Pacific. Finally, we note that the annual-mean nature

of our estimates is another possible reason for differences.

## 6   Export production

A key metric of the nutrient cycles is their export production, which determines the strength of the biological pump (e.g., Pasquier and Holzer, 2016). Export production is not directly available from satellite measurements, but observationally constrained estimates are easily calculated from our inverse model. The phosphorus export flux, $\Phi^P$, is simply the flux of organic

phosphorus into the aphotic zone that is remineralized there, which we compute using the operators $\mathcal{S}_c^P$ (sinking and remineralization) as

$$\Phi^P = \sum_{c=1}^{3} \int_a dz \, \mathcal{S}_c^P \, U_c \quad . \tag{27}$$

For plotting, we convert $\Phi^P$ to a carbon export flux using a constant $C:P$ ratio of 106:1. We similarly calculate the opal export as

$$\Phi^{Si} = \int_a dz \, \mathcal{S}_{dia}^P \, R_{dia}^{Si:P} U_{dia} \quad , \tag{28}$$

and the iron export associated with the remineralization of organic matter as

$$\Phi^{Fe} = \sum_{c=1}^{3} \int_a dz \, \mathcal{S}_c^P \, R_c^{Fe:P} U_c \quad , \tag{29}$$

where the vertical integrals are over the model aphotic zone (bottom to $73.4 \, \mathrm{m}$ depth).

    Figure 6a shows a map of the carbon export flux, together with its zonal integral for each member of our family of solutions.

The spatial pattern shows some differences with the estimate of Primeau et al. (2013) (blue curve in Figure 6a). Our estimate of the carbon export has 1.5–2 times larger tropical and high latitude peaks, but is closer to the satellite-derived estimates of Dunne et al. (2007). Our estimate also has sharper meridional gradients, which is arguably more realistic considering the sharp gradients in satellite-derived estimates of production (e.g., Frants et al., 2016). Our globally integrated carbon export of $9.49$–$11.0 \, \mathrm{Pg \, C \, yr^{-1}}$ is also larger than the $\sim 7.5 \pm 2.5 \, \mathrm{Pg \, C \, yr^{-1}}$ estimate of Primeau et al. (2013).

The differences with the estimate of Primeau et al. (2013) are likely due to very different uptake parameterizations: Primeau et al. (2013) consider the phosphorus cycle in isolation and optimize a single spatially varying uptake timescale for each grid box, while we explicitly represent three phytoplankton functional classes with different, optimized globally uniform uptake timescales, $\tau_c$. We note that if we use the same growth timescale for each phytoplankton class, our model's C-export remains close to that of Primeau et al. (2013).





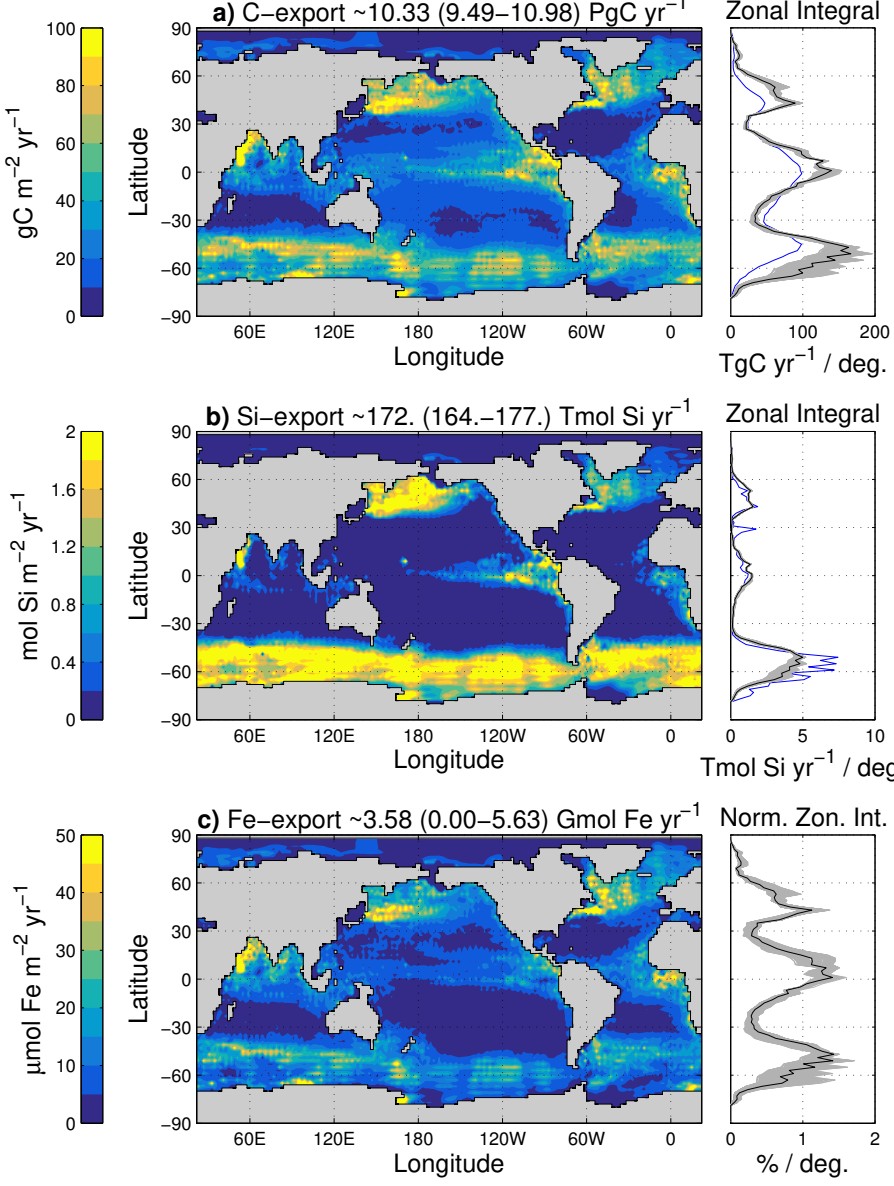

**Figure 6.** Local export production for each nutrient (maps on the left) and its zonal integral (curves on the right). Maps are shown for our typical solution, while we plot the zonal integral of each family member (scaled for $\mathrm{dFe}$) in grey and the typical solution in black. (a) Phosphorus export, expressed in carbon units using $\mathrm{C : P = 106 : 1}$. The blue zonal integral is the export production estimate of Primeau et al. (2013). (b) Opal export, where the blue zonal integral is the estimate of Holzer et al. (2014). (c) Iron export, with its zonal integrals expressed as a percentage of the global iron export.





Our estimates of the carbon-export production compare well with the satellite-based estimates of $9.7$–$12 \, \mathrm{Pg\,C\,yr^{-1}}$ by Gnanadesikan et al. (2004). Our estimates also lie within the wide range of $9$–$28 \, \mathrm{PgC\,yr^{-1}}$ of the Ocean-Carbon Cycle Model Intercomparison Project 2 (OCMIP-2, Najjar et al., 2007), and compare well the OCMIP-2 mean particle export of $13 \pm 3 \, \mathrm{Pg\,C\,yr^{-1}}$. (Because our model does not carry DOP, its particulate carbon export is given by its total carbon export.)

5   Figure 6b shows a map of the opal export, together with its zonal integral. As expected, opal export is most pronounced at high latitudes, particularly in the Southern Ocean. In spite of our relatively complex formulation of silicic acid utilization in terms of colimitations, the spatial pattern of the opal export and its global total of $164$–$177 \, \mathrm{Tmol\,Si\,yr^{-1}}$ compare well with the estimates by Holzer et al. (2014) ($171 \pm 31 \, \mathrm{Tmol\,Si\,yr^{-1}}$). Other estimates of the global opal export range from $69$ to $185 \, \mathrm{Tmol\,Si\,yr^{-1}}$ (e.g., Moore et al., 2004; Sarmiento et al., 2007; Heinze et al., 2003).

10   There is very little spread in the carbon and opal export production across our family of solutions as can be seen by the tightly clustered zonal integrals plotted in grey in Figures 6a,b. This shows that the carbon and opal exports are well constrained despite the wide range of iron inputs. Note, however, that export productions are sensitive to the parameters that set the biological uptake rate, which were optimized before generating the family of optimal solutions with different iron sources.

Figure 6c shows a map of the iron export associated with organic matter, but not including the iron export carried by 15 scavenging particles. The phosphorus and iron exports have broadly similar patterns, with differences that reflect variations in the local $\mathrm{Fe:P}$ uptake ratio. In the iron-deficient Southern Ocean, the $\mathrm{Fe:P}$ ratio is smaller than its global mean, which results in Southern Ocean iron export that is less efficient than that of phosphorus (for iron, the peak Southern Ocean export relative to the tropical peak is lower than for phosphorus). As expected from the widely varying iron source strengths across our family of solutions, the globally integrated iron export covers a wide range of $9.6 \times 10^{-4}$–$5.6 \, \mathrm{Gmol\,Fe\,yr^{-1}}$. However, the 20 geographic pattern of the iron export is robust across the family: the zonally integrated iron exports normalized by their global integrals collapse onto a well-defined cluster of curves. The spread in the thus normalized iron export is similar to the spread in the (un-normalized) carbon export, but slightly larger due to variations in the $\mathrm{Fe:P}$ ratio.

All export fields of Figure 6 show near-zero export in the Weddell Sea, in contrast to what restoring-type models tend to show. For example, the opal export estimated by Holzer et al. (2014) has a local maximum in the Weddell Sea. The Weddell 25 Sea minimum here is due to near-zero satellite measurements of photosynthetically active radiation in this region. This may well be an artifact of the satellite data, for which the irradiance in the Weddell Sea varies substantially depending on which years are averaged.

Figure 7 shows the carbon export partitioned according to each functional class. The bulk of the carbon export is accomplished by the Large and Diatom phytoplankton classes ($\sim 53$ and $41\,\%$, respectively), while the remainder ($\sim 6\,\%$) is exported 30 by the Small class. In the tropics, the Large class dominates the C-export, while in the high latitudes, and particularly in the Southern Ocean, the Diatom class dominates. This partition between the three phytoplankton classes is the result of the adjustments of the class-specific growth timescales, $\tau_c$, and of the parameters of the $\mathrm{Si:P}$ ratio to bring the fractional uptake by each class into alignment with the satellite-based estimates of Uitz et al. (2010). For our typical solution, these uptake fractions are $38\,\%$, $42\,\%$, and $30\,\%$ for the Diatom, Large, and Small classes, respectively. This compares to $32\,\%$, $44\,\%$, and 35 $24\,\%$ for micro$-$, nano$-$, and picophytoplankton, respectively, in the satellite-based estimates. (We find that if we use only a





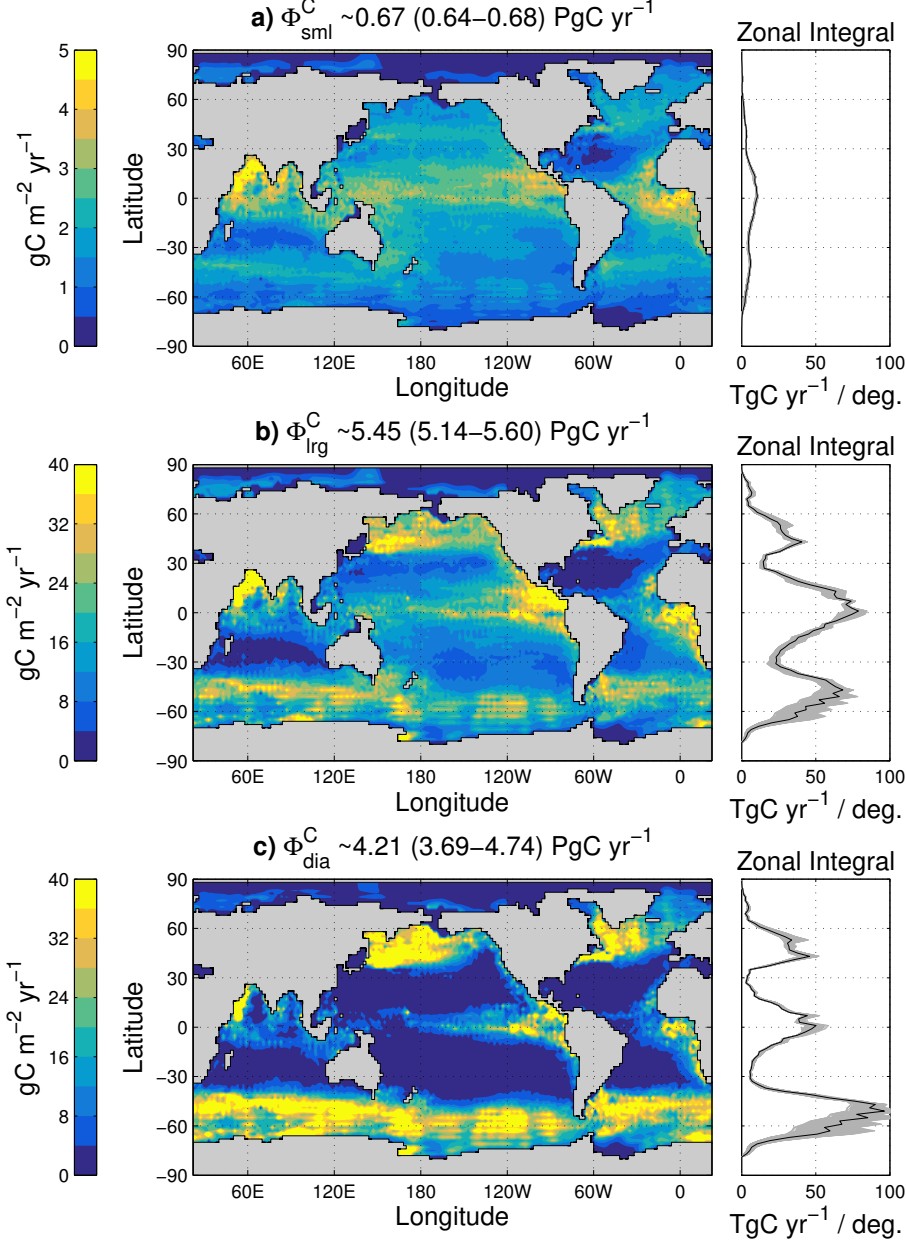

**Figure 7.** Local export production (maps on left) and its zonal integral (curves on the right) expressed in carbon units (using C : P = 106 : 1). Maps are shown for our typical solution, while we plot the zonal integral of each family member in grey and the typical solution in black. The export productions are plotted for each phytoplankton functional class: Small (top plots, a), Large (middle plots, b), and Diatom (bottom plots, c). Note the different colour scale for the Small class.





single optimized timescale for all three classes, the Small phytoplankton class completely dominates the phosphorus export, underlining the need for class-specific growth timescales.)

Export by the Diatom class has the sharpest meridional gradients. This is due to the diatoms' larger half-saturation rates, larger maximum phytoplankton concentrations, and their Si-limitation. Conversely, the carbon export by the Small class has the least spatial variation. This is consistent with satellite estimates of plankton concentration that show the least spatial variability for picophytoplankton (Kostadinov et al., 2016): Ubiquitous small phytoplankton are better adapted to nutrient-limited conditions, but do not benefit from nutrient abundance as much as larger phytoplankton.

## 7    Iron cycle

Here we document some of the key features of the iron cycle as constrained by our inverse model. Certain features such as the dFe concentration field are robustly constrained by the observations regardless of iron source strengths, while other features such as, e.g., the relative importance of hydrothermal iron, vary systematically with the source strengths.

### 7.1    Iron sources and sinks

The pattern of the aeolian source is identical for all family members because we only vary its global source strength, $\sigma_A$. The sediment source is keyed to export production, which is well constrained across the family of solutions. Therefore, the sedimentary iron source patterns are very similar across all solutions, with only the global strength $\sigma_S$ of the pattern varying among solutions. The initial hydrothermal pattern is set by the OCMIP $^3$He source (Dutay et al., 2004), but for total hydrothermal sources larger than $\sim 0.5\,\mathrm{Gmol\,yr}^{-1}$, the optimized contributions from each basin changed substantially. Across our family of solutions the mean and standard deviations of the percentage contributions from each basin to the total hydrothermal source are $(15 \pm 9)\%$ for the Atlantic, $(52 \pm 6)\%$ for the Pacific, $(16 \pm 2)\%$ for the Indian Ocean, and $(17 \pm 3)\%$ for the Southern Ocean (south of $40°$S). For reference, the vertically integrated iron sources of our typical solution are plotted in Appendix E.

Because of the small variations in the source patterns, the vertically integrated total sinks of dFe, $\int dz (1 - \mathcal{S}_s^{\mathrm{P}}) J_{\mathrm{POP}}$, $\int dz (1 - \mathcal{S}_s^{\mathrm{Si}}) J_{\mathrm{bSi}}$, and $\int dz\, J_{\mathrm{dst}}$, have patterns that also vary little across the family of solutions (see Appendix E for plots of the vertically integrated sinks of our typical solution). Note that these sinks balance the total source exactly because we are in steady state. The dominant iron sink is due to scavenging by organic particles (POP). The sink due to scavenging by opal accounts for roughly half of the total sink. The patterns of POP and opal scavenging are determined by the phosphorus and opal exports and by the concentration of free iron. Consequently, the organic scavenging sink is strongest in the tropics, and the opal scavenging sink is strongest in the Southern Ocean. The sink due to mineral dust scavenging reflects the pattern of the aeolian dust input modulated by the free iron concentration. However, for our family of solutions, the sink due to dust scavenging is essentially negligible, being about 3 orders of magnitude smaller than the POP and opal scavenging.





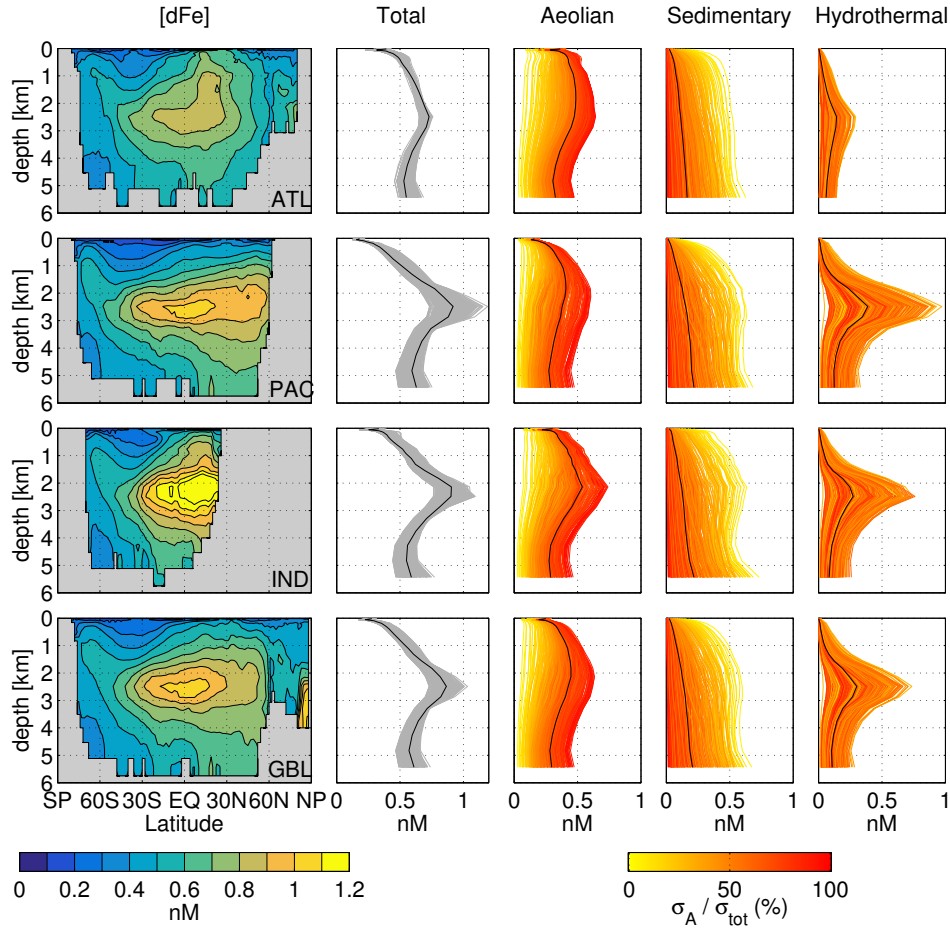

**Figure 8.** Estimates of the dFe concentration in each basin (ATL, PAC, IND) and globally (GBL). The zonal averages in latitude-depth space on the left show the total dFe field of our typical solution. The corresponding horizontally averaged profiles of total dFe are shown in grey for each family member and in black for the typical solution. The three columns of plots on the right show the source-partitioned dFe profiles, i.e., the profiles of the aeolian, sedimentary, and hydrothermal component of dFe for each family member. The individual source-partitioned profiles are colour coded according to the percent contribution of the aeolian iron source to the total iron source, with our typical solution in black.

## 7.2 dFe Concentration and source attribution

Figure 8 shows our typical solution's zonally averaged dFe concentration for each basin and for the global ocean. For each zonal average, we also show the corresponding profile of horizontally averaged dFe for each member of our family of solutions. The profiles are tightly clustered showing that the large-scale features of the dFe field are well constrained despite the large

5   variations of the iron sources. The inverse model fits the observed dFe field for widely different sources by adjusting the





corresponding scavenging. While these adjustments keep the total dFe field close to the observations, the relative contributions from the aeolian, sediment, and hydrothermal sources are unconstrained and can vary widely.

We calculate dFe concentrations due to each source following Holzer et al. (2016) by replacing the dFe concentration tracer equation (3) by an equivalent linear diagnostic system that has the same solution. This linear system, corresponding

to a given solution of the full nonlinear system, is obtained by replacing the iron uptake and scavenging by linear operators. Specifically, the dFe uptake $R^{\mathrm{Fe:P}}U_c$ is replaced with $L_{U,c}\chi_{\mathrm{Fe}}$ and the scavenging rate $J_j$ with $L_{J,j}\chi_{\mathrm{Fe}}$, where the linear operators, organized into matrix form, are simply specified from the uptake and scavenging rates of the nonlinear solution as $\mathbf{L}_{U,c} = \mathbf{diag}(R^{\mathrm{Fe:P}}U_c/\chi_{\mathrm{Fe}})$ and $\mathbf{L}_{J,j}\chi_{\mathrm{Fe}} \equiv \mathbf{diag}(J_j/\chi_{\mathrm{Fe}})$. The dFe concentration $\chi_{\mathrm{Fe}}^k$ due to source $s_k$ (with $k \in \{\mathrm{A,S,H}\}$) is then computed by replacing the total source in the linear equivalent system by the source $s_k$ of the component of interest and

computing the corresponding concentration.

Figure 8 also shows the profiles of the individual source components of dFe, colour coded according to the fractional strength of the aeolian source. In contrast to the profiles of the total dFe, these individual source components vary widely across the family of solutions, but in such a way that the total concentration $\chi_{\mathrm{Fe}} = \chi_{\mathrm{Fe}}^{\mathrm{A}} + \chi_{\mathrm{Fe}}^{\mathrm{S}} + \chi_{\mathrm{Fe}}^{\mathrm{H}}$ is tightly constrained. For example, for low aeolian sources (yellow profiles in Figure 8), the concentration of aeolian iron $\chi_{\mathrm{Fe}}^{\mathrm{A}}$ is relatively low,

but largely compensated by a relatively larger sediment contribution $\chi_{\mathrm{Fe}}^{\mathrm{A}}$. The concentrations of hydrothermal iron vary less systematically with the aeolian source, but all family members have very similarly shaped hydrothermal dFe profiles. However, the amplitudes of the hydrothermal dFe profiles can be seen to vary by roughly an order of magnitude across the majority of solutions, effectively fine tuning the total dFe concentration to be as close to the observations as possible.

### 7.3   Iron-type attributed export

### 7.3.1   Phosphorus export

We quantify the contribution of each iron type to the export production as follows. In our formulation, non-zero dFe is necessary for non-zero phosphate uptake $U_c$. The uptake $U_c(\boldsymbol{r})$ inferred at point $\boldsymbol{r}$ is supported by the dFe concentration at $\boldsymbol{r}$, which is a mixture of aeolian, sedimentary, and hydrothermal dFe. Thus, the uptake supported by dFe type $k$ is given by $(\chi_{\mathrm{Fe}}^k(\boldsymbol{r})/\chi_{\mathrm{Fe}}(\boldsymbol{r}))\,U_c(\boldsymbol{r})$, that is, the local uptake supported by dFe of type $k$ must be in proportion to the concentration fraction

$\chi_{\mathrm{Fe}}^k/\chi_{\mathrm{Fe}}$ (note that $\sum_k \chi_{\mathrm{Fe}}^k/\chi_{\mathrm{Fe}} = 1$). For a given nonlinear solution, the export phosphorus production supported by dFe of type $k$, denoted by $\phi_k^{\mathrm{P}}$ is therefore calculated by replacing the uptake $U_c$ in (27) with $(\chi_{\mathrm{Fe}}^k/\chi_{\mathrm{Fe}})\,U_c(\boldsymbol{r})$.

While the total export production is well constrained regardless of the chosen iron source strengths, the production supported by a given dFe type varies substantially with the magnitude of the corresponding source. (Summing over the three dFe types yields the well-constrained total.) However, regardless of the source amplitudes, the patterns of the export supported by each

dFe type is similar across the entire family of solutions.

Figure 9 shows $\widehat{\Phi}_k^{\mathrm{P}} \equiv \Phi_k^{\mathrm{P}}/\langle\Phi_k^{\mathrm{P}}\rangle$, which is the export flux supported by dFe type $k$ normalized by the global mean export $\langle\Phi_k^{\mathrm{P}}\rangle$. The $\widehat{\Phi}_k^{\mathrm{P}}$ patterns are plotted for our typical solution, together with zonal averages of $\widehat{\Phi}_k^{\mathrm{P}}$ for all family members. The patterns can be seen to differ little among family members. Even the pattern of the export supported by hydrothermal dFe,





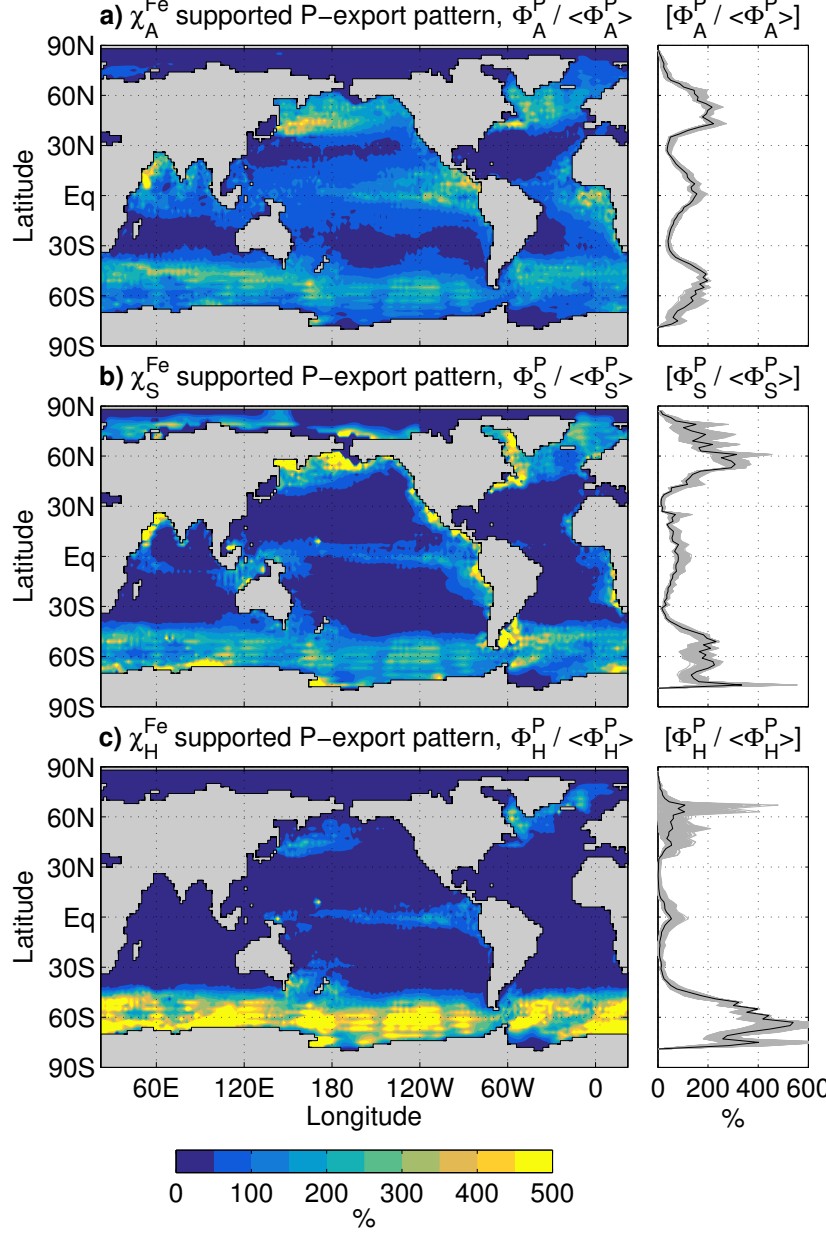

**Figure 9.** Phosphorus export supported by each dFe type [aeolian (a), sedimentary (b), hydrothermal (c)] normalized by its global mean. The maps show our typical solution, while zonal averages of the normalized phosphorus export are shown for each family member in grey, with the typical solution in black.





which varies most across the family, has the broadly similar features of peak export in the Southern Ocean, with a secondary peaks in the tropics and in the Northern Hemisphere subpolar oceans, for all family members.

Figure 9a shows that aeolian iron supports export primarily in the tropics and in the subpolar oceans. The tropics receive direct input of fresh aeolian iron, while the subpolar oceans receive upwelling regenerated iron (Holzer et al., 2016). The

aeolian-iron-supported export pattern is very similar to the pattern of the total export flux shown in Figure 6a (note that here we plot zonal averages, while Figure 6 shows zonal integrals). For sedimentary $dFe$ to support export it must be transported from the ocean bottom into the euphotic zone. Consequently, the pattern of the export supported by sedimentary $dFe$ (Fig. 9b) is dominated by regions of upwelling in the tropical and subpolar oceans and by regions of shallow depth (both resolved and subgrid) where there is high organic matter flux, such as the seas around Indonesia. The meridional gradients of the

sedimentary-iron-supported export are much sharper than for the aeolian-iron-supported export, presumably because of the lack of large-scale direct surface input for sedimentary $dFe$. The pattern of export supported by hydrothermal $dFe$ (Fig. 9c) is dominated by the Southern Ocean, where most of the density classes into which hydrothermal fluid is injected outcrop. Secondary regions of hydrothermal-iron-supported export are associated with upwelling in the tropics and in the subpolar oceans of the Northern Hemisphere.

Underscoring the similar source distribution of hydrothermal $dFe$ and mantle ${}^3$He, the pattern of hydrothermal-iron-supported export production is similar to the pattern with which mantle ${}^3$He outgases to the atmosphere (e.g., Holzer et al., 2017). We do not expect an exact correspondence in the patterns because hydrothermal $dFe$ is subject to scavenging losses, while ${}^3$He is not, and our ratio of hydrothermal $dFe$ source to mantle ${}^3$He source is different for different basins. (The ranges of the ratio of the optimized hydrothermal iron source to the mantle ${}^3$He source across the family of inverse-model estimates were $(0.0-3.3)$,

$(0.097-8.2)$, $(0.20-15.)$, and $(0.025-2.8)$ in units of $(\mathrm{Mmol\,Fe\,yr}^{-1})\,(\mathrm{mol\,{}^3He\,yr}^{-1})^{-1}$, for the Atlantic, Pacific, Indian, and Southern Ocean basins, respectively.)

While the total phosphorus export is well constrained and varies little across our family of solutions, the magnitude of the $dFe$-type-supported export production varies systematically with the relative source strength of each iron type. To quantify these systematic variations, Figure 10 plots the fraction $\langle \Phi^{\mathrm{P}}_k \rangle / \langle \Phi^{\mathrm{P}} \rangle$ of the globally averaged iron-type-$k$-supported export to

the total global export as a function of the corresponding fractional global iron source strengths $\sigma_k/\sigma_{\mathrm{tot}}$, where $\sigma_{\mathrm{tot}} \equiv \sum_k \sigma_k$. Note that if a given source strength $\sigma_k$ constitutes 100% of the total, then it must support 100% of the export and that if $\sigma_k = 0$ then it supports 0% of the export.

Figure 10 shows a number of interesting systematic features of our family of solutions. First, the aeolian $dFe$ supports $\sim$ 20–100% of the global export, with the low end of the range corresponding to an aeolian source of only $\sim$5% of the total source

(we did not explore lower fractional aeolian sources). Sedimentary iron supports $\sim$ 0–80% of the global export, with the high end of the range corresponding to a sediment source as high as $\sim$ 90% of the total source. Hydrothermal iron supports the least export ranging from $\sim$ 0–18% for fractional hydrothermal sources as large as $\sim$45% of the total source.

A key point of Figure 10 is that aeolian iron may be considered to be the most important type of iron for supporting export production: For a given fraction of the total source, the fraction of export supported by aeolian iron is larger (i.e.,

the aeolian points all lie above the 1:1 line). For the fractional aeolian sources between $\sim$20 and $\sim$50%, the corresponding





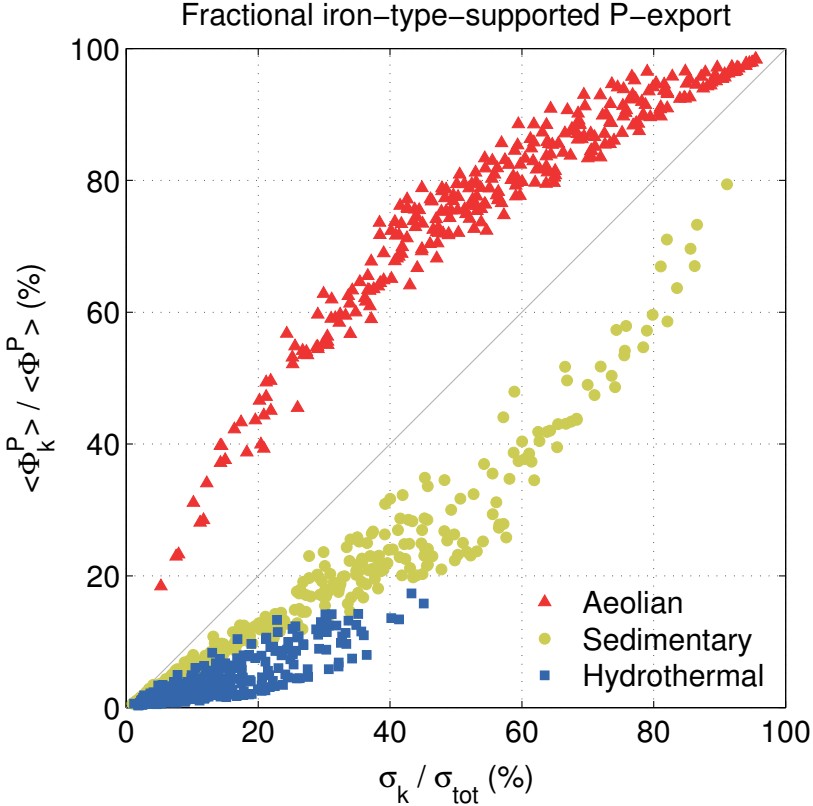

**Figure 10.** Percent global phosphorus export (equivalently carbon export) supported by each iron type (aeolian, sedimentary, hydrothermal) versus the corresponding fractional source of that iron type.

export fraction is as much as ∼30% greater than the fractional source. As the factional aeolian source approaches 100% from below or 0% from above, it comes close to supporting an equal fraction of the export because the relation between fractional production and fractional source must pass through the points $(0\%, 0\%)$ and $(100\%, 100\%)$. The ability of aeolian iron to make disproportionately large contributions to supporting phosphate and carbon export is presumably due to fresh aeolian iron being

5 directly injected into the euphotic zone. This view is supported by the fact that if we compute the fractional export only for the Southern Ocean, where the aeolian source is small and most aeolian iron is supplied as upwelled regenerated iron (Holzer et al., 2016), the deviation of the fractional aeolian-iron-supported export from the 1:1 line is roughly halved.

Sedimentary and hydrothermal dFe make fractional contributions to the export that are less than their fractional sources (the sedimentary and hydrothermal points lie below the 1:1 line by as much as ∼20%). Over the range of the fractional hydrothermal

10 source, the hydrothermal-iron-supported export fraction is roughly two thirds the corresponding sedimentary-iron-supported export fraction. The lower efficiency of hydrothermal iron presumably reflects the fact that most large sedimentary sources are relatively shallow and can thus supply iron more readily to the euphotic zone compared to the deep hydrothermal sources.



This is consistent with the fact that hydrothermal and sedimentary iron have roughly equal efficiency for the fractional exports computed for the Southern Ocean only (not shown).

### 7.3.2 Opal export

The opal export supported by each dFe type can be calculated analogously, and the corresponding geographic patterns are shown in Figure 11. Similar to the total opal export (Figure 6b), the patterns of the opal export supported by each iron type emphasize regions with high diatom concentrations, namely the Southern Ocean and subpolar North Pacific and North Atlantic where there is also upwelling and/or vertical mixing. Aeolian-iron-supported opal export (Fig. 11a) is large in the Southern Ocean, but most pronounced in the subpolar North Pacific, where both diatom production is significant and aeolian input is high downwind from Asia's deserts. While tropical opal export is of secondary importance, the tropics are most pronounced for aeolian-supported export, again because of the direct source there. The pattern of sedimentary-iron-supported opal export (Fig. 11b) is broadly similar to that for aeolian dFe, but weaker in the tropics. The pattern of hydrothermal-iron-supported opal export (Fig. 11c) is dominated by the Southern Ocean, where diatom production is high and where most hydrothermal iron upwells. The patterns of iron-type-supported opal export have tightly clustered zonal means with a similar spread across the family as for the phosphorus export.

The amplitude of the opal-export patterns varies systematically with the iron source strength as summarized in Figure 12, which shows the fractional iron-type-supported opal export as a function of the corresponding fractional dFe source. While aeolian dFe is still the most efficient iron type for supporting opal export, aeolian dFe is less efficient for opal export than for phosphorus export (the aeolian points fall closer to the 1:1 line by ∼10% fractional export). For a given fractional aeolian source strength between 20% and 60%, the supported fractional opal export is typically ∼20% larger, though there is significant scatter. Conversely, sedimentary iron is slightly more efficient in supporting opal export than in supporting phosphorus export, and hydrothermal dFe is only slightly less efficient than sedimentary dFe.

The lower per-source efficiency of aeolian iron for supporting opal export is consistent with the fact that opal export occurs primarily in the Southern Ocean, where direct aeolian input is small. Similarly, the greater per-source efficiency of sedimentary and hydrothermal iron is consistent with the bulk of the opal export occurring in the upwelling regions of the Southern Ocean where access to deep iron sources is greatest. This is supported by the fact that a plot of the fractional *phosphorus* export of the Southern Ocean only (not shown) versus the fractional iron sources is nearly identical to the plot of fractional *global* opal export of Figure 12.

## 8 Discussion and caveats

Our approach has a number of limitations that must be kept in mind. Most importantly, inverse-model estimates are only as good as the data used to constrain them. The dFe observations are too sparse in space and time to construct a gridded annual mean climatology like those available for $PO_4$ and $Si(OH)_4$. We averaged the available dFe data to minimize observational biases, but in many places observations are only available for one time of the year and likely contain seasonal biases. Other





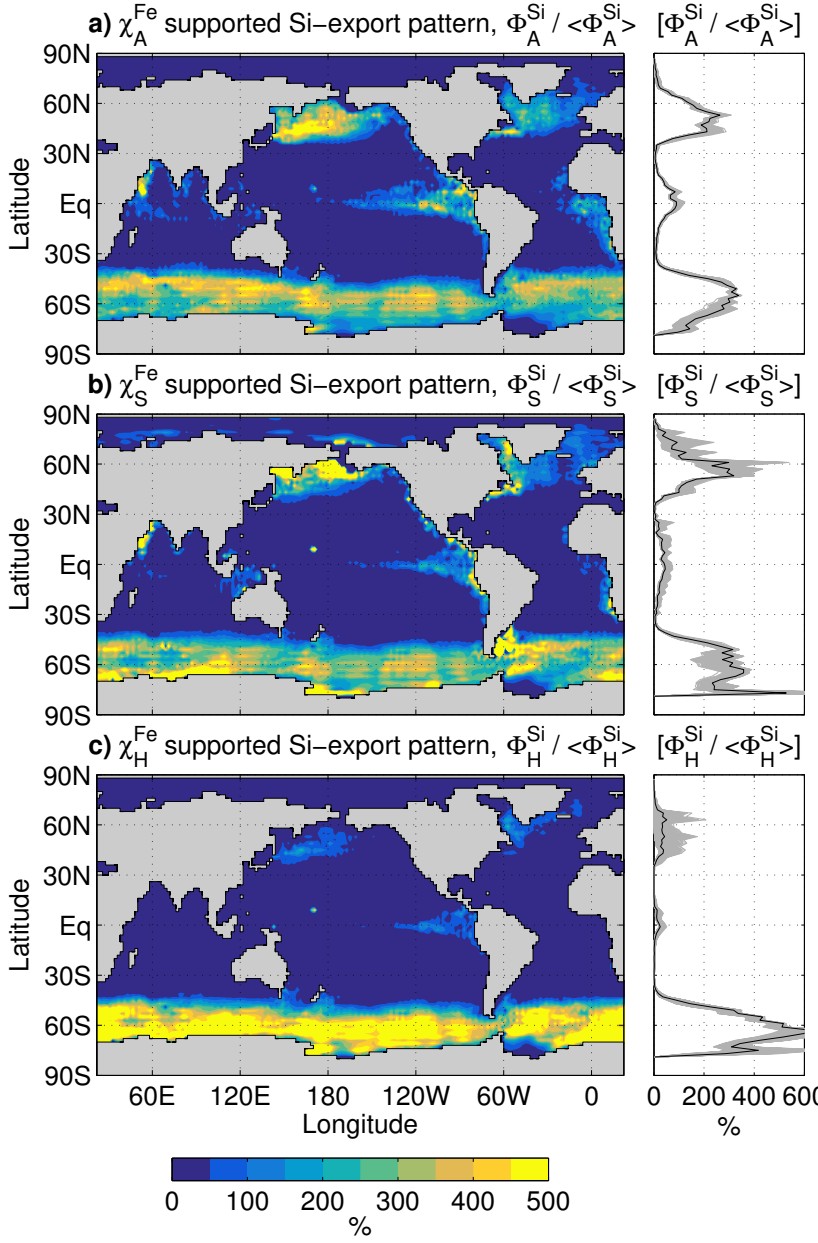

**Figure 11.** Opal export supported by each dFe type [aeolian (a), sedimentary (b), hydrothermal (c)] normalized by its global mean. The map is for our typical solution, while zonal averages of the normalized opal export are shown for each family member in grey, with the typical solution in black.





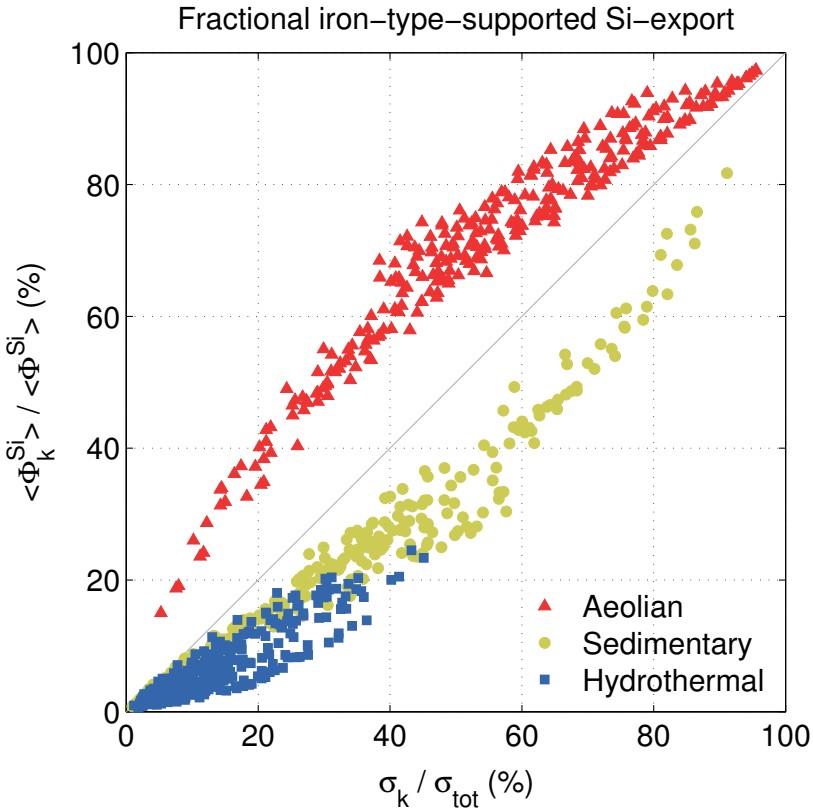

**Figure 12.** Percent global opal export supported by each iron type (aeolian, sedimentary, hydrothermal) versus the corresponding fractional source of that iron type.

biases are likely introduced when dFe measurements aliase episodic source events such mineral dust downwind from the major deserts (e.g., Croot et al., 2004; Johnson et al., 2010). In the near future, GEOTRACES will release an expanded data product that will include Pacific transects that were not available for the intermediate data product used here. The additional dFe observations will help constrain the hydrothermal sources, particularly the strength of the Pacific source relative to that of

5     the other basins.

      Important non-nutrient observational fields for our inverse model are the satellite-measured photosynthetically active radiation (PAR) and ocean-colour-derived estimates of the size-partitioned phytoplankton concentrations. Small-scale features of the PAR field, e.g., in the Weddell Sea where ice and cloud cover play a role, are uncertain with the PAR for different time averages showing different features. The satellite-based estimates of phytoplankton concentrations also carry unquantified un-

10    certainties due to a number of assumptions (Kostadinov et al., 2016). In our inverse model, these estimates provide crucial constraints on how carbon export is partitioned among the different functional classes. The unquantified uncertainties warrant re-evaluation as independent satellite-derived estimates become available in the future.

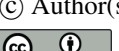



Most biogeochemical parameters are determined through objective optimization against available observations, but the construction of the cost function, and the choice of which parameters are optimized and which are prescribed, necessarily involves subjective choices. For example, choosing a different set of weights $(\omega_P, \omega_{Fe}, \omega_{Si}, \omega_{plk})$ to combine the four terms of the cost function would result in different optimal parameters. Similarly, assigning greater weight to dFe data measured as part of a vertical profile introduces another arbitrary weight. As for any nonlinear least-squares problem, it is also important to recognize that any minimum of the cost function found numerically is not guaranteed to be the global minimum and it is always possible that a better fit exists for a different set of parameters. Conversely, depending on the choice of initial state, the optimizer may find a local minimum that has grossly unrealistic features and must be rejected.

We addressed the uncertainty in our estimates that results from the nearly unconstrained strengths of the external iron sources. The uncertainty in key metrics (e.g., global carbon export) was quantified in terms of their spread across our family of solutions and in terms of systematic variations with the iron source strengths. While our efficient numerics allow us to easily determine the linear sensitivities of any metric with respect to all parameters (from which one can also estimate uncertainty), we did not do so here because the spread in the metric across the family is more relevant. Given the large set of parameters $x_j$ and several interesting metrics $M_i$, a detailed investigation of all the sensitivities $\partial M_i / \partial x_j$ evaluated at the optimal states is beyond the scope of this study. In principle, one can estimate the uncertainty of the optimal parameters themselves using a Bayesian framework (e.g., Teng et al., 2014). However, this requires the construction of suitable covariances and is also beyond the scope of this study.

A key limitation of our approach is that seasonality is ignored and we use a steady circulation. This circulation is constructed so that its transport reproduces the annual-mean observed temperature, salinity, CFC, radiocarbon, and $PO_4$ fields with minimal error. The circulation is hence not a simple average, but an effective ventilation-weighted mean. However, we acknowledge that effects due to the seasonal covariance of biological production and circulation cannot be captured.

Our model of the nutrient cycles makes a number of simplifying assumptions. We ignore external inputs of silicic acid and therefore also neglect permanent burial of opal in sediments. While this approximation has been shown to have negligible impact on particle fluxes (Sarmiento et al., 2007), we acknowledge that our estimates will miss features such as, e.g., silicic-acid plumes due to crustal fluid venting (Johnson et al., 2006). The uncertainty of the silicon cycle that is most difficult to quantify stems from our simple parameterization of opal dissolution, which does not account for partial frustule protection by decaying organic material or the effect of digestion by zooplankton. Another key uncertainty lies in our parameterization of the Si : P uptake ratio, particularly its dependence on dFe. While our empirical formulation captures known dependencies qualitatively, a first-principles derivation based on cell biology is currently lacking. These remarks apply equally to the Fe : P uptake ratio.

Although our model of the iron cycle includes an explicit representation of the redissolution of scavenged iron, effects of subgrid topography, and dynamic coupling to the phosphorus and silicon cycles, and is thus much more complex and mechanistic than the iron model of Frants et al. (2016), it was still necessary to make a number of simplifying assumptions. Specifically, we do not model ligands dynamically, ignore colloidal iron (e.g., Fitzsimmons and Boyle, 2014), and do not represent some iron sources that may be locally important such as input from icebergs (Klunder et al., 2011, 2014). We also

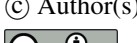



assume that $PO_4$ and dFe are remineralized with the same Martin curve and in the same ratio in which they were utilized. The recent work by Twining et al. (2014) suggests that sinking diatoms release phosphorus higher in the water column than iron, but we do not have sufficient information to model these effects. Given the large uncertainties in the external iron sources, the neglected details are likely of second order for estimating the large-scale dFe concentration.

Other uncertainties concern the phosphorus cycle to which the uptake of the other elements is keyed. While the optimized phosphate fields have the smallest misfit with observations, our model of the phosphorus cycle makes several simplifying assumptions that should be kept in mind. The Martin exponent is assumed to be globally uniform although in reality it almost certainly varies spatially (Weber et al., 2016), potentially leading to underestimated gradients in our model. To avoid carrying an additional tracer, we approximated DOP to have zero lifetime. In reality, DOP has a wide range of lifetimes, and the lifetime

of semi-labile DOP is typically assumed to be a fraction of a year (e.g., Primeau et al., 2013). However, the neglect of DOP is unlikely to seriously affect our estimates. DOP represents only a tiny fraction (less that 1%) of the total phosphorus pool (e.g., Pasquier and Holzer, 2016) and by using a Martin exponent optimized for a restoring model without DOP, we were able to match $PO_4$ concentrations to within 5% of the observations.

We emphasize that the carbon export reported here was simply our estimate of the phosphorus export converted to carbon

units. No effort was made to compute a more realistic carbon export such as could be achieved with an explicit representation of the carbon cycle (which would require additional tracers and was numerically too expensive) and the $C : P$ export ratio was treated as globally uniform. While a globally uniform export ratio is acceptable for a unit conversion, the true $C : P$ export ratio is now known to vary spatially (Teng et al., 2014).

## 9  Summary and Conclusions

We have formulated a steady-state model of the coupled phosphorus, silicon, and iron cycles that is embedded in a steady data-assimilated global circulation. The model is of intermediate complexity and couples the nutrient cycles through colimitations on biological uptake and through the scavenging of iron by organic particles. The concentrations of the small, large, and diatom phytoplankton functional classes are calculated diagnostically, which avoids the need for plankton concentration tracers. We explicitly represent iron scavenging by POP, opal, and mineral-dust particles, and the redissolution of POP- and

opal-scavenged iron. Subgrid topography is parameterized for the sedimentary iron sources and intercepts all vertical fluxes. The relative simplicity of the biogeochemical model and the matrix formulation of the steady-state advective diffusive transport afford highly efficient numerics. Steady-state solutions are readily found using a Newton solver, which permits the model to be used in inverse mode to constrain many of the biogeochemical parameters through objective optimization. The optimization minimizes the mismatch with the observed WOA13 nutrient concentrations and with satellite-derived estimates of

phytoplankton concentrations (Uitz et al., 2010).

Our optimized estimates of the macronutrient concentrations closely match the observational climatology with volume-weighted RMS errors of $5\,\%$ for phosphate and $12\%$ for silicic acid relative to the global mean concentrations. The modelled dFe concentration has a larger cost-weighted RMS mismatch of $\sim 45\,\%$ of the global mean dFe concentration. However, the





cost-weighted basin-averaged vertical dFe profiles for the Atlantic and Southern Ocean generally lie within the observational uncertainties. The Pacific dFe profiles show systematic biases, in part because the Pacific basin contains relatively few dFe observations in the currently available data sets. The estimated phytoplankton concentrations have generally weaker meridional gradients than observed possibly because of our use of a globally uniform Martin exponent. However, the fraction of the global plankton biomass represented by each functional class lies within at most $\sim$4% of the fraction in the corresponding size class as estimated by Kostadinov et al. (2009) from ocean-colour observations.

Given that even the order of magnitude of the iron sources is uncertain, we generated a family of solutions for a wide range of source strengths. Because our inverse-model estimates compensate for different source strengths by adjusting the scavenging parameters, each family member fits the available observations with roughly the same fidelity. This means that the available observed dFe and phytoplankton concentrations by themselves are insufficient to constrain the sources. This conclusion can also be gleaned from the model intercomparison of Tagliabue et al. (2016) and was reached using an inverse model by Frants et al. (2016), who also considered a family of solutions. However, while Frants et al. (2016) varied primarily the aeolian source, our family of solutions explores a range of sedimentary and hydrothermal source strengths in addition to a much wider range of aeolian source strengths.

We partitioned the dFe concentration field into its aeolian, sedimentary, and hydrothermal components without perturbing the system using the approach of Holzer et al. (2016). We find that the individual source components vary widely depending on the source strengths, but that the total dFe concentration given by the sum of the source components is well constrained. Variations in the aeolian component are compensated primarily by sedimentary dFe. Both the compensations between different iron types and between effective sources and sinks suggest that a more dense sampling of the ocean's dFe field by future measurement campaigns may not provide the information necessary for constraining the source strengths. The required information may ultimately have to come from better direct quantification of the source and/or scavenging processes themselves.

Nutrient limitation patterns were defined by jointly considering if the $PO_4$, $Si(OH)_4$, and dFe concentrations fell below their half-saturation values for uptake. Iron limitation was thus deemed to occur where only dFe fell below its half-saturation value, phosphate–iron colimitation where both $PO_4$ and dFe fell below their half-saturation values, and so on. The resulting limitation patterns are robust across our family of solutions and broadly consistent with direct observations (Moore et al., 2013) and with alternatively defined limitation patterns in the BEC model (Moore et al., 2004). The Large and Diatom functional classes show iron limitation in the Southern Ocean, eastern tropical Pacific and subpolar North Pacific, with $PO_4$–dFe (for diatoms $PO_4$–$Si(OH)_4$–dFe) colimitations in the Pacific and South Atlantic subtropical gyres. The Indian Ocean, tropical Atlantic, and North Atlantic are largely iron replete (i.e., not limited in the sense defined) with $PO_4$ limitation (for diatoms $PO_4$–$Si(OH)_4$ colimitation).

The export productions of phosphorus and opal are well constrained across our family of solutions, both in terms of pattern and magnitude. Because we model three phytoplankton functional classes with distinct, optimized uptake timescales, our carbon export of $9.5 - 11.$ Pg C yr$^{-1}$ is $\sim$30 – 45% larger than that estimated by Primeau et al. (2013) and closer in spatial pattern to the empirical satellite-based estimates of Dunne et al. (2007). The range of our opal export of $164 - 177$ Tmol Si





yr$^{-1}$ overlaps with the estimate of Holzer et al. (2014), who used a simple restoring-type model of the silicon cycle uncoupled from other nutrients.

We estimate the percentage of the global carbon export due the Small, Large, and Diatom functional classes to be $(6.4 \pm 0.3)\%$, $(52 \pm 2)\%$, and $(41 \pm 3)\%$, respectively. These percentages were not well constrained by the nutrient and phytoplankton concentration data and required joint tuning of the parameters of the Si : P uptake ratio and of the uptake timescales so that the fractional uptake by each class approximately matched the satellite-based estimates of Uitz et al. (2010).

We quantified the role of the iron cycle in shaping the carbon and opal export productions. We find that each iron source type (aeolian, sedimentary, hydrothermal) supports phosphorus and opal export with a distinct geographic pattern that is robust across the family of solutions. The export pattern supported by a given iron type reflects the nature of its source. Sedimentary- and hydrothermal-iron-supported carbon export is dominantly shaped by the large-scale patterns of upwelling that brings these iron types to the surface. Aeolian-iron-supported export is shaped by both the pattern of direct aeolian input and by large-scale upwelling, which brings regenerated aeolian iron, as well as redissolved scavenged aeolian iron, back into the euphotic zone. For opal export, the signature of each iron type is qualitatively similar, but compared to carbon export, the opal export patterns tend to be weaker in the tropics and stronger at high latitudes, especially in the Southern Ocean where diatom concentrations and silicon trapping are strongest.

The amplitude of the export pattern supported by a given iron type varies systematically with the fractional global source of that iron type. These variations summarize and quantify the efficiency of each source for supporting export. Aeolian iron is most efficient in the sense that the percentage of the export it supports exceeds the percentage of the aeolian source by as much as ∼30% for carbon export and ∼20% for opal export. Globally, sedimentary and hydrothermal iron are inefficient, supporting a fraction of export that is up to ∼20% less than their fractional sources. For carbon export, hydrothermal iron is least efficient supporting about 1/3 less fractional export than sedimentary iron. When the effect of direct aeolian deposition is minimized by calculating the exports for the Southern Ocean only, the efficiency of a given iron type is similar regardless of whether carbon or opal export is considered. This reflects the similar oceanic transport paths of all iron types into the Southern Ocean euphotic zone, where the bulk of aeolian iron is supplied in regenerated form through upwelling.

Our optimized model is ideally suited for investigating the response of the global ocean ecosystem to a variety of perturbations. In future publications, we will report on the model's response to perturbations in the iron supply and analyze the detailed workings of the iron cycle more comprehensively.

## 10 Data availability

The temperature, phosphate, and silicic-acid data used in this study are available from the World Ocean Atlas v2 2013 (www.nodc.noaa.gov/OC5/woa13/woa13data.html). The dFe data used in this study, including the TFE data set, are available from GEOTRACES (www.bodc.ac.uk/geotraces/data). The satellite estimates of the concentrations of picophytoplankton, nanophytoplankton, and microphytoplankton are available from the PANGAEA data repository (doi:10.1594/PANGAEA.859005). The yearly irradiance data from NASA's MODIS Aqua PAR are available from the OceanColor website (oceandata.sci.gsfc.nasa.gov).





## Appendix A: Recycling operators for scavenged iron

The recycling operator for POP-scavenged iron, $\mathcal{B}^{\mathrm{s,POP}}$, comprises two parts: For dFe scavenged in the euphotic layer, i.e., above $z_{\mathrm{e}}$, $\mathcal{B}^{\mathrm{s,POP}}$ acts identically to $\mathcal{B}^{\mathrm{P}}$. For dFe scavenged below $z_{\mathrm{e}}$, we solve the flux equation of scavenged iron for continuous $z$. We assume that iron scavenged by POP below the mixed layer continuously sinks and can be recycled in the same grid cell in which it was scavenged. (We assume that the mixed layer coincides with the euphotic zone.) As shown by Kriest and Oschlies (2008), the Martin curve can be simply modeled with a sinking speed linearly increasing with depth, an approach we follow here. The equation for the flux or iron, $\phi_{\mathrm{Fe}}^{\mathrm{POP}}$, that was scavenged by POP below the mixed layer is thus

$$\frac{\partial \phi_{\mathrm{Fe}}^{\mathrm{POP}}}{\partial z} + \frac{b}{z}\phi_{\mathrm{Fe}}^{\mathrm{POP}} = -f^{\mathrm{POP}}J_{\mathrm{POP}} \quad , \tag{A1}$$

where $b$ is the Martin exponent of the POP flux, and with the condition that $\phi_{\mathrm{Fe}}^{\mathrm{POP}} = 0$ for $z > z_{\mathrm{e}}$ because here we only consider dFe scavenged below $z_{\mathrm{e}}$. The solution is given by

$$\phi_{\mathrm{Fe}}^{\mathrm{POP}}(z) = -\int_{z}^{z_{\mathrm{e}}} \left(\frac{z}{z'}\right)^{-b} f^{\mathrm{POP}}J_{\mathrm{POP}}(z')dz' \quad . \tag{A2}$$

And the rate per unit volume at which POP-scavenged dFe is recycled is thus given by

$$\mathcal{B}^{\mathrm{s,POP}}f^{\mathrm{POP}}J_{\mathrm{POP}} = \mathcal{B}^{\mathrm{P}}f^{\mathrm{POP}}J_{\mathrm{POP}} + \frac{b}{z}\phi_{\mathrm{Fe}}^{\mathrm{POP}} \quad , \tag{A3}$$

where the first term is for iron that was scavenged in the euphotic zone, and the second term for iron that was scavenged in the interior.

Similarly, the recycling operator for opal-scavenged iron, $\mathcal{B}^{\mathrm{s,bSi}}$, has a euphotic part identical to $\mathcal{B}^{\mathrm{Si}}$, and an aphotic interior part. In the interior, we solve the continuous equation for the flux of iron, $\phi_{\mathrm{Fe}}^{\mathrm{bSi}}$, that was scavenged by opal below the mixed layer. The flux obeys

$$\frac{\partial \phi_{\mathrm{Fe}}^{\mathrm{bSi}}}{\partial z} + \frac{\kappa_{\mathrm{Si}}^{\mathrm{max}}}{w_{\mathrm{Si}}}e^{-T_E/T(z)}\phi_{\mathrm{Fe}}^{\mathrm{bSi}} = -f^{\mathrm{bSi}}J_{\mathrm{bSi}} \quad , \tag{A4}$$

with the condition that $\phi_{\mathrm{Fe}}^{\mathrm{bSi}} = 0$ for $z > z_{\mathrm{e}}$. The solution is of the form

$$\phi_{\mathrm{Fe}}^{\mathrm{bSi}}(z) = -\int_{z}^{z_{\mathrm{e}}} Q(z,z')f^{\mathrm{bSi}}J_{\mathrm{bSi}}(z')dz' \quad , \tag{A5}$$

so that the flux of opal-scavenged iron at $z$ integrates all the scavenging of dFe by opal that occurs above $z$ (and below $z_{\mathrm{e}}$) and that is not recycled before reaching $z$. This is accomplished by $Q(z,z')$, given by

$$Q(z,z') = \exp\left(-\int_{z}^{z'} \frac{\kappa_{\mathrm{Si}}^{\mathrm{max}}}{w_{\mathrm{Si}}}e^{-T_E/T(z'')}dz''\right) \quad , \tag{A6}$$

which removes all the recycling that occurs between the current depth $z$ and the depth of scavenging $z'$. The rate per unit volume at which opal-scavenged dFe is recycled is thus given by

$$\mathcal{B}^{\mathrm{s,bSi}}f^{\mathrm{bSi}}J_{\mathrm{bSi}} = \mathcal{B}^{\mathrm{P}}f^{\mathrm{bSi}}J_{\mathrm{bSi}} + \frac{\kappa_{\mathrm{Si}}^{\mathrm{max}}}{w_{\mathrm{Si}}}e^{-T_E/T(z)}\phi_{\mathrm{Fe}}^{\mathrm{bSi}} \quad , \tag{A7}$$



where the first term is for iron that was scavenged in the euphotic zone, and the second term for iron that was scavenged in the aphotic interior.

## Appendix B: Biogenic transport operators with subgrid topography

We follow Moore and Braucher (2008) to include the effects of the more realistic, high-resolution topographic data from the
National Geophysical Data Center (2006). The subgrid topography parameterization must be reflected in the redistribution operators. Here we explain how this is done, based on the biogenic redistribution operators $\mathcal{B}^{\mathrm{P}}$ and $\mathcal{B}^{\mathrm{Si}}$, which link biological production in the euphotic zone to remineralization or redissolution in the aphotic zone. The operators $\mathcal{B}^{\mathrm{P}}$ and $\mathcal{B}^{\mathrm{Si}}$ are related to the divergence of the aphotic particle fluxes through

$$\mathcal{B}^{\mathrm{P}} f_c U_c = \frac{\partial}{\partial z}\left[\Theta(z - z_{\mathrm{b}})\phi_{\mathrm{POP}}(z)\right] \quad, \tag{B1}$$

and

$$\mathcal{B}^{\mathrm{Si}} f_{\mathrm{dia}} U_{\mathrm{dia}} = \frac{\partial}{\partial z}\left[\Theta(z - z_{\mathrm{b}})\phi_{\mathrm{bSi}}(z)\right] \quad, \tag{B2}$$

where $z_{\mathrm{e}}$ and $z_{\mathrm{b}}$ are the height coordinates at the base of the euphotic zone, and at the ocean bottom, respectively. The Heaviside function, $\Theta(z - z_{\mathrm{b}})$, ensures that the fraction of POP that reaches $z_{\mathrm{b}}$ is redissolved and remineralized there (Primeau et al., 2013; Holzer et al., 2014).

The subgrid topography parameterization is implemented by applying equations (B1)–(B2) to the high-resolution topography. In practice, for each model grid cell, we calculate the fraction of the flux that should remineralize where it hits the topography. Note, to ensure that $\mathcal{B}^{\mathrm{P}}$ and is $\mathcal{B}^{\mathrm{Si}}$ are conservative, special care is required where subgrid topography is present within euphotic grid cells. In this case, a corresponding fraction of sinking particles must remineralize in that cell. We similarly implemented the same subgrid topography parameterization to the scavenging redistribution operators, $\mathcal{B}^{\mathrm{s,POP}}$ and $\mathcal{B}^{\mathrm{s,bSi}}$.

## Appendix C: Weights for dFe mismatch

We use the dFe observations of both the global data set compiled by Tagliabue et al. (2012) and the GEOTRACES Intermediate Data Product v3 (Mawji et al., 2015). We combine both data sets and remove dFe observations above 2.71 nM which probably correspond to transient states with short timescales that cannot be captured by our steady-state model. In order to compensate the fact that most dFe observations are close to the surface, we give more weight to observations that are part of a "profile". (A dFe observation is deemed to belong to a "profile" if there are 10 or more observations at the same latitude and longitude, and if one of those was recorded deeper than 2000 m.) Because the dFe observations do not sample the seasonal cycle uniformly, we adopt an approach similar to Frants et al. (2016) to reduce potential sampling bias when we interpolate the data to our model grid: If multiple dFe observations lie in the same grid cell, we first take the seasonal averages, which we then averaged again to estimate the annual mean.





As in equations (22) for $PO_4$ and $Si(OH)_4$, we use volume weights to evaluate the $dFe$ concentration mismatch with observations. However, because not all model grid cells contain $dFe$ observations, we define a $dFe$-specific vector of grid-box volumes, $\boldsymbol{v}_{Fe}^{all}$, which has nonzero elements only for grid boxes that contain at least one $dFe$ observation. We also define a $dFe$ "profile-specific" vector, $\boldsymbol{v}_{Fe}^{pro}$, which is nonzero only for grid boxes that contain "profile" observations. The corresponding

weights are defined by

$$\boldsymbol{w}_{Fe}^{all} = \frac{\boldsymbol{v}_{Fe}^{all}}{\left(\overline{\chi}_{Fe}^{obs}\right)^2 V_{Fe}^{all}} \qquad \text{and} \qquad \boldsymbol{w}_{Fe}^{pro} = \frac{\boldsymbol{v}_{Fe}^{pro}}{\left(\overline{\chi}_{Fe}^{obs}\right)^2 V_{Fe}^{pro}} \quad , \tag{C1}$$

where $V_{Fe}^{all}$ is the total volume of grid cells which contain a $dFe$ observation, and $V_{Fe}^{pro}$ the total volume of grid cells containing "profile" observations. We define the total $dFe$ weight vector, $\boldsymbol{w}_{Fe}$, for the mismatch with observations in equation (23), by

$$\boldsymbol{w}_{Fe} = \frac{1}{5}\boldsymbol{w}_{Fe}^{all} + \frac{4}{5}\boldsymbol{w}_{Fe}^{pro} \quad , \tag{C2}$$

where we give extra weight to the "profile" observations. The $1:4$ ratio was manually adjusted until "profile" observations were deemed to have sufficiently strong influence on the solutions. We also tried different approaches to weight the model-observation $dFe$ mismatch, including the use of inverse variances (Frants et al., 2016), but we found no significant difference in our results.

## Appendix D: Optimization strategy details

**D0.1   Prescribed parameters**

The following considerations determined which parameters were not optimized and how their values were chosen. The recyclable fractions of POP and opal scavenging, $f^{POP}$ and $f^{bSi}$, compensate with the maximum $Fe:P$ uptake ratio, $R_0^{Fe:P}$, and thus were prescribed at $90\,\%$ (Moore and Braucher, 2008). (This compensation results from the biological iron pump having almost the same effect as the combination of scavenging and recycling iron.) Similarly, the detrital fractions, $f_c^0$, which set the

particle export ratio, are directly compensated by all the other parameters in the uptake formulation. We therefore followed Dunne et al. (2005) and assigned their "small" detrital fraction to $f_{sml}^0$ and their "large" detrital fraction to both $f_{lrg}^0$ and $f_{dia}^0$. When trying to optimize the silicon half-saturation rate $k_{dia}^{Si}$, starting from a value of $1\,\mathrm{mmol\,m^{-3}}$ (e.g., Matsumoto et al., 2013), we found that the optimal value always remained within a few percent of this initial value. This is in part due to the fact that in regions of high diatom concentration the Monod term for silicic acid is near saturation so that there is little sensitivity to

the precise value of $k_{dia}^{Si}$. Moreover, there appears to be consistency across the literature that $k_{dia}^{Si} = 1\,\mathrm{mmol\,m^{-3}}$. We therefore simply fixed $k_{dia}^{Si}$ at this value for numerical efficiency.

**D0.2   Choice of initial parameter values**

We first chose an initial set of values for the remaining parameters as collected in Table 2. The parameters of the iron cycle

were taken from of the typical solution of Frants et al. (2016) except for the half-saturation constant of the $Fe:P$ ratio, which





was taken from the work of Galbraith et al. (2010), and the scavenging-rate parameters. The initial parameters for POP and opal scavenging, $\kappa_{\mathrm{scv}}^{\mathrm{POP}}$ and $\kappa_{\mathrm{scv}}^{\mathrm{bSI}}$, where determined so that the globally integrated scavenging of each process was initially $\sim 5$ $\mathrm{Gmol\,Fe\,yr}^{-1}$ (the typical total source/sink strength reported by Frants et al. (2016)). The initial value of the dust scavenging rate parameter, $\kappa_{\mathrm{dst}}^{\mathrm{POP}}$, was chosen so that the sink due to dust scavenging was $\sim 10\%$ of the total sink of the initial state.

The initial irradiance half-saturation constants were taken from the work of Doney et al. (2006). The initial uptake half-saturation constants $k_c^i$ were taken from the work of Matsumoto et al. (2013). The uptake timescales $\tau_c$ were set to an initial value of 6 days and optimized subject to the constraint $\tau_{\mathrm{sml}} \geq \tau_{\mathrm{lrg}} \geq \tau_{\mathrm{dia}}$. The initial values of the maximum phytoplankton concentrations were calculated as $p_c^{\max} = p^*/(\lambda_0 \tau_c)$ using $p^* = 0.018\,\mathrm{mmol\,P\,m}^{-3}$ (Galbraith et al., 2010) and $\lambda_0^{-1} = 5.26\,\mathrm{d}$ (Dunne et al., 2005).

The initial values of the parameters of the Si : P ratio were set so that $k_{\mathrm{Si:P}}^{\mathrm{Fe}}$ and $k_{\mathrm{Si:P}}^{\mathrm{Si}}$ where on the order of typical dFe and $\mathrm{Si(OH)}_4$ concentrations, while $R_0^{\mathrm{Si}}$ and $R_m^{\mathrm{Si}}$ were based on corresponding Si : N uptake ratios found in the literature and converted using N : P=16 : 1. Thus, in terms of Si : N units, $R_0^{\mathrm{Si}}$ was chosen to be on the order of the minimum Si : N uptake ratio used by Matsumoto et al. (2013) and $R_m^{\mathrm{Si}}$ was chosen to be on the order of the maximum Si : N uptake ratio reported by Franck et al. (2000) and Brzezinski et al. (2002).

**D0.3    Sequential optimization steps**

(a) We first optimized the hydrothermal iron source parameters $\sigma_{\mathrm{H,ATL}}$, $\sigma_{\mathrm{H,PAC}}$, $\sigma_{\mathrm{H,IND}}$, and $\sigma_{\mathrm{H,SO}}$ forcing $\sigma_{\mathrm{H,IND}} = \sigma_{\mathrm{H,SO}}$. Although we adjusted the overall source strength of the hydrothermal iron source when generating our family of iron cycling estimates, we did not re-optimize the relative strength of the 4 basin amplitudes until the final step of our strategy.

(b) We jointly optimize the three irradiance half-saturations $k_c^I$ and then keep these fixed because of potential compensation 20   with the half-saturation constants $k_c^i$.

(c) We jointly optimized the half-saturations $k_c^i$ of the nutrient-limitation Monod terms.

(d) We were not able to optimize the Si : P uptake ratio parameters, because they are not well constrained due to compensation with the parameters that set the uptake by diatoms. We therefore separately tuned the parameters $R_m^{\mathrm{Si}}$, $R_0^{\mathrm{Si}}$, $k_{\mathrm{Si:P}}^{\mathrm{Si}}$, and $k_{\mathrm{Si:P}}^{\mathrm{Fe}}$ together with the three growth timescales $\tau_c$ to match the fractions of total uptake by each phytoplankton class as estimated by 25   Uitz et al. (2010).

(e) Because of compensation with the maximum Fe : P uptake ratio, the associated half-saturation rate, $k_{\mathrm{Fe:P}}$, was optimized on its own.

(f) We then jointly re-optimized the 13 parameters $k_c^{\mathrm{P}}$, $k_c^{\mathrm{Fe}}$, $k_{\mathrm{dia}}^{\mathrm{Si}}$, $p_c^{\max}$, and $\tau_c$.

(g) Only the parameters of the iron cycle remain to be optimized. Iron source and sink parameters cannot jointly be optimized 30   because of strong local compensation. (Although the aeolian source injects dFe at the surface, uptake and scavenging export iron to depth, thus creating an effective interior source.) To generate our family of estimates, we therefore first assigned the aeolian, sedimentary, and hydrothermal source-strength parameters (keeping the same ratio of basin hydrothermal source strengths to global hydrothermal source strength), and held these fixed while jointly optimizing the parameters determining the iron scavenging, namely $R_0^{\mathrm{Fe:P}}$, $\kappa_{\mathrm{scv}}^{\mathrm{POP}}$, $\kappa_{\mathrm{scv}}^{\mathrm{bSi}}$, $\kappa_{\mathrm{scv}}^{\mathrm{dst}}$, $L_{\mathrm{v}}$, $L_{\max}$, $L_{\mathrm{b}}$, and $\tau_{\mathrm{b}}$.





(h) As a final step, we jointly optimized all source-strength parameters $\sigma_A$, $\sigma_S$, $\sigma_{H,ATL}$, $\sigma_{H,PAC}$, $\sigma_{H,IND}$, and $\sigma_{H,SO}$ primarily to give the hydrothermal source pattern (relative strength in each basin) a chance to adjust from its initial state. We find that if the total hydrothermal source strength, $\sigma_H$, is below $1\,\mathrm{Gmol\,yr}^{-1}$, the final optimization step hardly changes $\sigma_H$, while larger hydrothermal source strengths tend to be reduced by up to $\sim 0.5\,\mathrm{Gmol\,yr}^{-1}$. If $\sigma_H < 0.5\,\mathrm{Gmol\,yr}^{-1}$, the fractional hydrothermal source strength of each basin tends to remain unchanged during this last step, while if $\sigma_H \geq 0.5\,\mathrm{Gmol\,yr}^{-1}$, the fractional hydrothermal sources for the Pacific and Southern Ocean tend to increase by order 10%, the fractional Indian Ocean source tends to decrease order 5%, and the Atlantic source is typically reduced by order 10% and for some solutions to near zero. The aeolian iron source strength, $\sigma_A$, tends to be reduced by $\sim 10\%$, while the sedimentary iron source strength, $\sigma_S$, stays within 10% of its previous value for most family members but can more than double for cases with high hydrothermal and aeolian sources.

## Appendix E: Iron source and sink patterns

Figure 13 shows the vertically integrated sources of dFe with a logarithmic colour scale. The aeolian soluble iron deposition pattern is identical to that of the study of Luo et al. (2008), albeit limited to the oceans. The tropical Atlantic close to the Sahara, the Arabian Sea, and the Bay of Bengal are the regions of largest aeolian iron deposition. The hydrothermal iron sources follow the mid-ocean ridges with the pattern of the OCMIP protocol, but independently scaled for the Atlantic, Pacific, Indian, and Southern Ocean basins. Sedimentary iron is more intense where export production is large and in areas where oceans are shallower, because in both cases, a large flux of organic matter (or POP in our model) reaches the sediment. The sub-grid topography plays a significant role in the pattern of sedimentary iron, in particular for coastal regions and large underwater plateaus, e.g., near the Kerguelen islands or the Falkland islands. Because of unrealistic circulation features in the Sea of Japan, we zero all sources there consistent with our zeroing out production in the Sea of Japan.

Figure 14 shows the vertically integrated sinks that balance the sources of Figure 13. The scavenging due to sinking mineral dust particles is about 3 orders of magnitude smaller than the sink due to organic and opal particle scavenging and could be neglected without changing our estimates appreciably. Although the pattern of the scavenging sinks has significant local variations among the members of our family of solutions, the zonally averaged pattern (vertically integrated sink normalized by its global mean) is broadly similar across the family.

*Acknowledgements.* We thank François Primeau for making the data-assimilated circulation available to us, Natalie Mahowald for providing the aeolian iron flux estimate of Luo et al. (2008), and Marina Frants for discussions. This work was supported by ARC grant DP120100674 (MH). BP gratefully acknowledges scholarship support from the Government of Monaco, the Scientific Centre of Monaco, the Frères Louis et Max Principale Foundation, and the Cuomo Foundation.




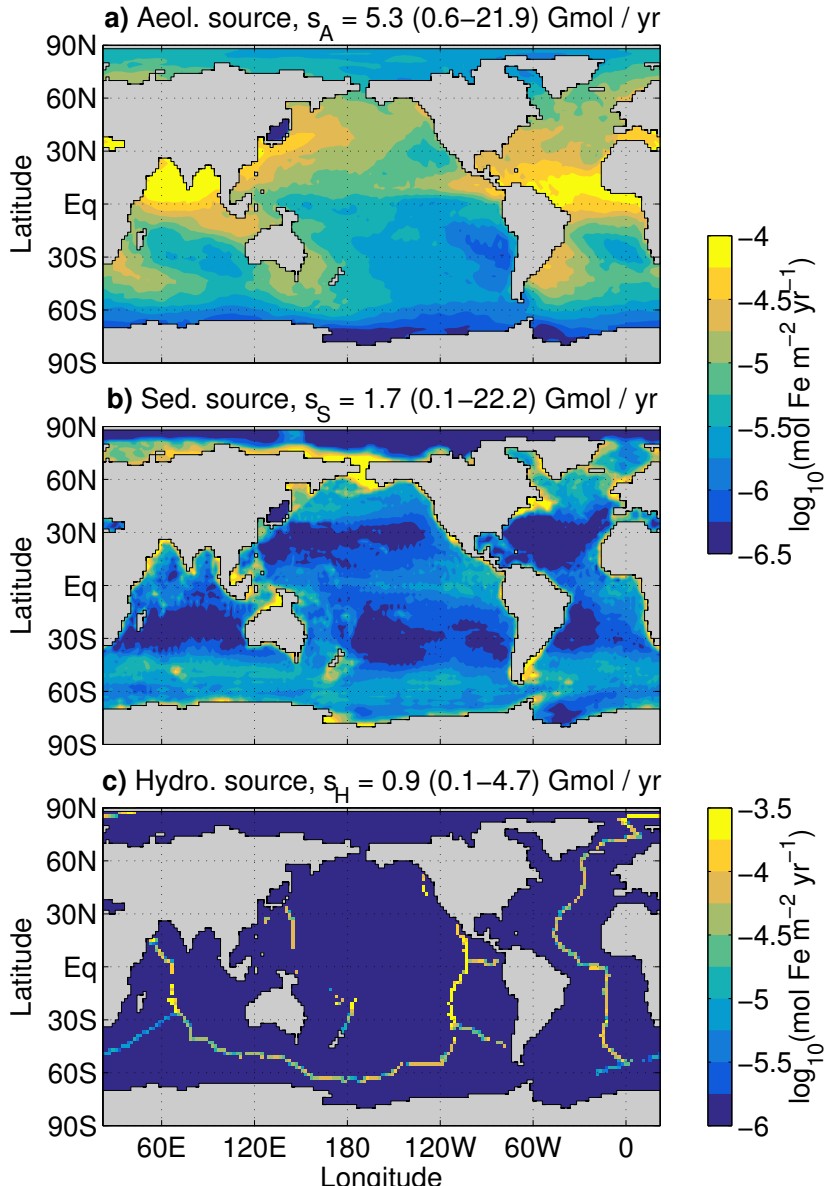

**Figure 13.** Vertically integrated sources of aeolian dFe (top, a), sedimentary dFe (middle, b), and hydrothermal dFe (bottom, c) for our typical solution. Note the logarithmic colour scales.

## References

Aumont, O., Ethé, C., Tagliabue, A., Bopp, L., and Gehlen, M.: PISCES-v2: an ocean biogeochemical model for carbon and ecosystem studies, Geoscientific Model Development, 8, 2465–2513, doi:10.5194/gmd-8-2465-2015, 2015.





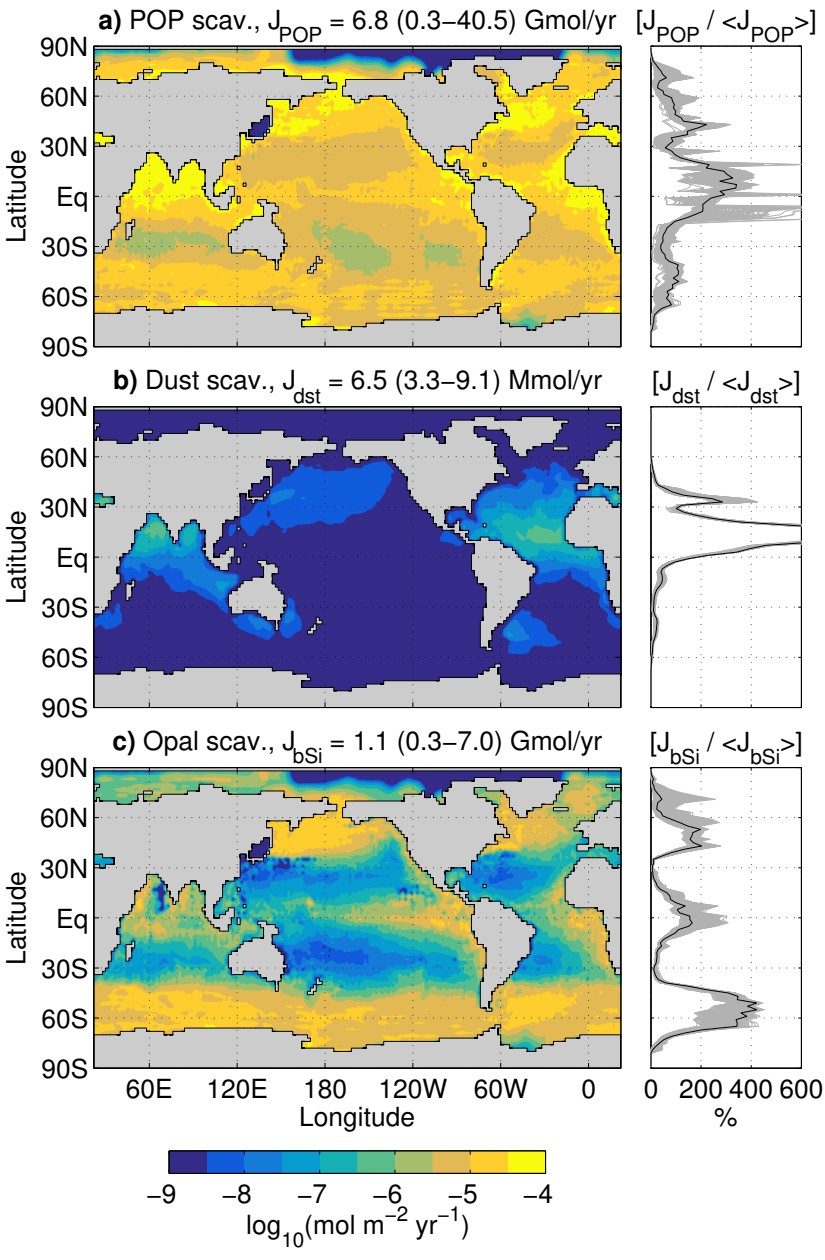

**Figure 14.** Maps of the vertically integrated iron sinks of our typical solution due to (a) POP, (b) mineral dust, and (c) sinking opal particles. Plotted to the right are the zonal averages of the vertically integrated sinks normalized by their global mean, with each family member in grey and our typical solution in black.





Bennett, S. A., Achterberg, E. P., Connelly, D. P., Statham, P. J., Fones, G. R., and German, C. R.: The distribution and stabilisation of dissolved Fe in deep-sea hydrothermal plumes, Earth and Planetary Science Letters, 270, 157 – 167, doi:10.1016/j.epsl.2008.01.048, 2008.

Boyd, P. W. and Ellwood, M. J.: The biogeochemical cycle of iron in the ocean, Nature Geosci, 3, 675–682, doi:10.1038/ngeo964, 2010.

Boyd, P. W., Jickells, T., Law, C. S., Blain, S., Boyle, E. A., Buesseler, K. O., Coale, K. H., Cullen, J. J., de Baar, H. J. W., Follows, M., Harvey, M., Lancelot, C., Levasseur, M., Owens, N. P. J., Pollard, R., Rivkin, R. B., Sarmiento, J., Schoemann, V., Smetacek, V., Takeda, S., Tsuda, A., Turner, S., and Watson, A. J.: Mesoscale Iron Enrichment Experiments 1993-2005: Synthesis and Future Directions, Science, 315, 612–617, doi:10.1126/science.1131669, 2007.

Brzezinski, M. A., Pride, C. J., Franck, V. M., Sigman, D. M., Sarmiento, J. L., Matsumoto, K., Gruber, N., Rau, G. H., and Coale, K. H.: A

switch from $Si(OH)_4$ to $NO_3^-$ depletion in the glacial Southern Ocean, Geophys. Res. Lett., 29, 1564, doi:10.1029/2001GL014349, 2002.

Brzezinski, M. A., Dickson, M.-L., Nelson, D. M., and Sambrotto, R.: Ratios of Si, C and N Uptake by Microplankton in the Southern Ocean, Deep Sea Research Part II: Topical Studies in Oceanography, 50, 619–633, doi:10.1016/s0967-0645(02)00587-8, 2003.

Brzezinski, M. A., Baines, S. B., Balch, W. M., Beucher, C. P., Chai, F., Dugdale, R. C., Krause, J. W., Landry, M. R., Marchi, A., Measures, C. I., Nelson, D. M., Parker, A. E., Poulton, A. J., Selph, K. E., Strutton, P. G., Taylor, A. G., and Twining, B. S.: Co-Limitation of

Diatoms by Iron and Silicic Acid in the Equatorial Pacific, Deep Sea Research Part II: Topical Studies in Oceanography, 58, 493–511, doi:10.1016/j.dsr2.2010.08.005, 2011a.

Brzezinski, M. A., Krause, J. W., Church, M. J., Karl, D. M., Li, B., Jones, J. L., and Updyke, B.: The annual silica cycle of the North Pacific subtropical gyre, Deep Sea Research Part I: Oceanographic Research Papers, 58, 1988–1001, doi:10.1016/j.dsr.2011.08.001, 2011b.

Buesseler, K. O.: The decoupling of production and particulate export in the surface ocean, Global Biogeochem. Cycles, 12, 297–310,

doi:10.1029/97GB03366, 1998.

Croot, P. L., Streu, P., and Baker, A. R.: Short residence time for iron in surface seawater impacted by atmospheric dry deposition from Saharan dust events, Geophys. Res. Lett., 31, L23S08, doi:10.1029/2004GL020153, 2004.

de Baar, H.: von Liebig's law of the minimum and plankton ecology (1899–1991), Progress in Oceanography, 33, 347–386, doi:10.1016/0079-6611(94)90022-1, 1994.

de Baar, H. J. W., de Jong, J. T. M., Bakker, D. C. E., Loscher, B. M., Veth, C., Bathmann, U., and Smetacek, V.: Importance of iron for plankton blooms and carbon dioxide drawdown in the Southern Ocean, Nature, 373, 412–415, doi:10.1038/373412a0, 1995.

Doney, S. C., Lindsay, K., Fung, I., and John, J.: Natural Variability in a Stable, 1000-Yr Global Coupled Climate-Carbon Cycle Simulation, J. Climate, 19, 3033–3054, doi:10.1175/JCLI3783.1, 2006.

Droop, M. R.: The nutrient status of algal cells in continuous culture, Journal of the Marine Biological Association of the United Kingdom,

54, 825–855, doi:10.1017/S002531540005760X, 2009.

Dunne, J. P., Armstrong, R. A., Gnanadesikan, A., and Sarmiento, J. L.: Empirical and Mechanistic Models for the Particle Export Ratio, Global Biogeochem. Cycles, 19, doi:10.1029/2004gb002390, 2005.

Dunne, J. P., Sarmiento, J. L., and Gnanadesikan, A.: A synthesis of global particle export from the surface ocean and cycling through the ocean interior and on the seafloor, Global Biogeochem. Cycles, 21, GB4006, doi:10.1029/2006GB002907, 2007.

Dutay, J.-C., Jean-Baptiste, P., Campin, J.-M., Ishida, A., Maier-Reimer, E., Matear, R., Mouchet, A., Totterdell, I., Yamanaka, Y., Rodgers, K., Madec, G., and Orr, J.: Evaluation of OCMIP-2 ocean models' deep circulation with mantle helium-3, Journal of Marine Systems, 48, 15–36, doi:10.1016/j.jmarsys.2003.05.010, 2004.



Dutkiewicz, S., Follows, M. J., and Parekh, P.: Interactions of the iron and phosphorus cycles: A three-dimensional model study, Global Biogeochem. Cycles, 19, GB1021, doi:10.1029/2004GB002342, 2005.

Dutkiewicz, S., Follows, M. J., Heimbach, P., and Marshall, J.: Controls on ocean productivity and air-sea carbon flux: An adjoint model sensitivity study, Geophys. Res. Lett., 33, L02603, doi:10.1029/2005GL024987, 2006.

Elrod, V. A., Berelson, W. M., Coale, K. H., and Johnson, K. S.: The flux of iron from continental shelf sediments: A missing source for global budgets, Geophys. Res. Lett., 31, doi:10.1029/2004GL020216, L12307, 2004.

Eppley, R. W.: Temperature and Phytoplankton Growth in the Sea, Fish. Bull, 70, 1063–1085, 1972.

Fitzsimmons, J. N. and Boyle, E. A.: Both soluble and colloidal iron phases control dissolved iron variability in the tropical North Atlantic Ocean, Geochimica et Cosmochimica Acta, 125, 539–550, doi:10.1016/j.gca.2013.10.032, 2014.

Flynn, K. J.: Modelling multi-nutrient interactions in phytoplankton - balancing simplicity and realism, Progress in Oceanography, 56, 249–279, doi:10.1016/s0079-6611(03)00006-5, 2003.

Franck, V. M., Brzezinski, M. A., Coale, K. H., and Nelson, D. M.: Iron and Silicic Acid Concentrations Regulate Si Uptake North and South of the Polar Frontal Zone in the Pacific Sector of the Southern Ocean, Deep Sea Research Part II: Topical Studies in Oceanography, 47, 3315–3338, doi:10.1016/S0967-0645(00)00070-9, US Southern Ocean JGOFS Program (AESOPS), 2000.

Frants, M., Holzer, M., DeVries, T., and Matear, R.: Constraints on the Global Marine Iron Cycle from a Simple Inverse Model, Journal of Geophysical Research: Biogeosciences, 121, 28–51, doi:10.1002/2015jg003111, 2016.

Galbraith, E. D., Gnanadesikan, A., Dunne, J. P., and Hiscock, M. R.: Regional Impacts of Iron-Light Colimitation in a Global Biogeochemical Model, Biogeosciences, 7, 1043–1064, doi:10.5194/bg-7-1043-2010, 2010.

Garcia, H. E., Locarnini, R. A., Boyer, T. P., Antonov, J. I., Baranova, O. K., Zweng, M. M., Reagan, J. R., and Johnson, D. R.: World Ocean

Atlas 2013, NOAA Atlas NESDIS 76, U.S. Government Printing Office, Washington, D.C., Volume 4: Dissolved Inorganic Nutrients (phosphate, nitrate, silicate), S. Levitus, Ed., A. Mishonov Technical Ed., 2014.

Gnanadesikan, A.: A global model of silicon cycling: Sensitivity to eddy parameterization and dissolution, Global Biogeochem. Cycles, 13, 199–220, doi:10.1029/1998GB900013, 1999.

Gnanadesikan, A., Dunne, J. P., Key, R. M., Matsumoto, K., Sarmiento, J. L., Slater, R. D., and Swathi, P. S.: Oceanic ventilation and

biogeochemical cycling: Understanding the physical mechanisms that produce realistic distributions of tracers and productivity, Global Biogeochem. Cycles, 18, GB4010, doi:10.1029/2003GB00209, 2004.

Gran, H. H. et al.: On the conditions for the production of plankton in the sea, Rapp. Proc. Verb. Cons. Int. Explor. Mer., 75, 37–46, 1931.

Hart, T. J.: On the phytoplankton of the south-west Atlantic and the Bellingshausen Sea, 1929–31, Discovery Reports, VIII, l–268, 1934.

Hawkes, J., Connelly, D., Gledhill, M., and Achterberg, E.: The stabilisation and transportation of dissolved iron from high temperature

hydrothermal vent systems, Earth and Planetary Science Letters, 375, 280–290, doi:10.1016/j.epsl.2013.05.047, 2013.

Heinze, C., Hupe, A., Maier-Reimer, E., Dittert, N., and Ragueneau, O.: Sensitivity of the marine biospheric Si cycle for biogeochemical parameter variations, Global Biogeochem. Cycles, 17, 1086, doi:10.1029/2002GB001943, 2003.

Holzer, M. and Primeau, F. W.: Global teleconnections in the oceanic phosphorus cycle: patterns, paths, and timescales, J. Geophys. Res., 118, 1775–1796, doi:10.1002/jgrc.20072, 2013.

Holzer, M., Primeau, F. W., DeVries, T., and Matear, R.: The Southern Ocean silicon trap: Data-constrained estimates of regenerated silicic acid, trapping efficiencies, and global transport paths, Journal of Geophysical Research: Oceans, 119, 313–331, doi:10.1002/2013JC009356, 2014.



Holzer, M., Frants, M., and Pasquier, B.: The age of iron and iron source attribution in the ocean, Global Biogeochem. Cycles, doi:10.1002/2016GB005418, 2016GB005418, 2016.

Holzer, M., DeVries, T., Bianchi, D., Newton, R., Schlosser, P., and Winckler, G.: Objective estimates of mantle $^3$He in the ocean and implications for constraining the deep ocean circulation, Earth Planet. Sci. Lett., 458, 305–314, doi:10.1016/j.epsl.2016.10.054, 2017.

Howarth, R. W.: Nutrient Limitation of Net Primary Production in Marine Ecosystems, Annual Review of Ecology and Systematics, 19, 89–110, doi:10.1146/annurev.es.19.110188.000513, 1988.

Hutchins, D. A. and Bruland, K. W.: Iron-Limited Diatom Growth and Si:N Uptake Ratios in a Coastal Upwelling Regime, Nature, 393, 561–564, doi:10.1038/31203, 1998.

Jin, X., Gruber, N., Dunne, J. P., Sarmiento, J. L., and Armstrong, R. A.: Diagnosing the contribution of phytoplankton functional groups

to the production and export of particulate organic carbon, CaCO$_3$, and opal from global nutrient and alkalinity distributions, Global Biogeochem. Cycles, 20, GB2015, doi:10.1029/2005GB002532, 2006.

Johnson, H. P., Hautala, S. L., Bjorklund, T. A., and Zarnetske, M. R.: Quantifying the North Pacific silica plume, Geochem. Geophys. Geosyst., 7, Q05011, doi:10.1029/2005GC001065, 2006.

Johnson, M. S., Meskhidze, N., Solmon, F., Gassó, S., Chuang, P. Y., Gaiero, D. M., Yantosca, R. M., Wu, S., Wang, Y., and Carouge,

C.: Modeling dust and soluble iron deposition to the South Atlantic Ocean, J. Geophys. Res., 115, D15202, doi:10.1029/2009JD013311, 2010.

Kelley, C. T.: Solving Nonlinear Equations with Newton's Method, chap. 1. Introduction, pp. 1–25, SIAM, doi:10.1137/1.9780898718898.ch1, 2003a.

Kelley, C. T.: Solving Nonlinear Equations with Newton's Method, chap. 2. Finding the Newton Step with Gaussian Elimination, pp. 27–55,

SIAM, doi:10.1137/1.9780898718898.ch2, 2003b.

Klunder, M., Laan, P., Middag, R., Baar, H. D., and van Ooijen, J.: Dissolved iron in the Southern Ocean (Atlantic sector), Deep-Sea Research II, 58, 2678–2694, doi:10.1016/j.dsr2.2010.10.042, 2011.

Klunder, M. B., Laan, P., Baar, H. J. W. D., Middag, R., Neven, I., and Ooijen, J. V.: Dissolved Fe across the Weddell Sea and Drake Passage: impact of DFe on nutrient uptake, Biogeosciences, 11, 651–669, doi:10.5194/bg-11-651-2014, 2014.

Kostadinov, T. S., Siegel, D. A., and Maritorena, S.: Retrieval of the particle size distribution from satellite ocean color observations, Journal of Geophysical Research: Oceans, 114, doi:10.1029/2009JC005303, 2009.

Kostadinov, T. S., Milutinović, S., Marinov, I., and Cabré, A.: Carbon-based phytoplankton size classes retrieved via ocean color estimates of the particle size distribution, Ocean Science, 12, 561–575, doi:10.5194/os-12-561-2016, 2016.

Kostadinov, T. S., Milutinović, S., Marinov, I., and Cabré, A.: Size-partitioned phytoplankton carbon concentrations retrieved from ocean

color data, links to data in netCDF format, doi:10.1594/PANGAEA.859005, 2016.

Kriest, I. and Oschlies, A.: On the treatment of particulate organic matter sinking in large-scale models of marine biogeochemical cycles, Biogeosciences, 5, 55–72, doi:10.5194/bg-5-55-2008, 2008.

Kwon, E. Y. and Primeau, F.: Sensitivity and Optimization Study of a Biogeochemistry Ocean Model using an Implicit Solver and In-Situ Phosphate Data, Global Biogeochem. Cycles, 20, GB4009, doi:10.1029/2005GB002631, 2006.

Kwon, E. Y. and Primeau, F.: Optimization and sensitivity of a global biogeochemistry ocean model using combined in situ DIC, alkalinity, and phosphate data, Journal of Geophysical Research: Oceans, 113, doi:10.1029/2007JC004520, C08011, 2008.





Lundry, M. R., Barber, R. T., Bidare, R. R., Chai, F., Coale, K. H., Dam, H. G., Lewis, M. R., Lindley, S. T., McCarthy, J. J., Roman, M. R., Stoecker, D. K., Verity, P. G., and White, J. R.: Iron and grazing constraints on primary production in the central equatorial Pacific: An EqPac synthesis, Limnology and Oceanography, 42, 405–418, doi:10.4319/lo.1997.42.3.0405, 1997.

Luo, C., Mahowald, N., Bond, T., Chuang, P. Y., Artaxo, P., Siefert, R., Chen, Y., and Schauer, J.: Combustion iron distribution and deposition, Global Biogeochemical Cycles, 22, doi:10.1029/2007GB002964, GB1012, 2008.

Marchetti, A., Parker, M. S., Moccia, L. P., Lin, E. O., Arrieta, A. L., Ribalet, F., Murphy, M. E. P., Maldonado, M. T., and Armbrust, E. V.: Ferritin is Used for Iron Storage in Bloom-Forming Marine Pennate Diatoms, Nature, 457, 467–470, doi:10.1038/nature07539, 2009a.

Marchetti, A., Varela, D. E., Lance, V. P., Lance, V. P., Palmucci, M., Giordano, M., and Armbrust, E. V.: Iron and Silicic Acid Effects on Phytoplankton Productivity, Diversity, and Chemical Composition in the Central Equatorial Pacific Ocean, Limnol. Oceanogr., 55, 11–29, doi:10.4319/lo.2010.55.1.0011, 2009b.

Martin, J. H.: Glacial-Interglacial $CO_2$ Change: The Iron Hypothesis, Paleoceanography, 5, 1–13, doi:10.1029/PA005i001p00001, 1990.

Martin, J. H. and Fitzwater, S. E.: Iron deficiency limits phytoplankton growth in the north-east Pacific subarctic, Nature, 331, 341–343, doi:10.1038/331341a0, 1988.

Massoth, G. J., Baker, E. T., Feely, R. A., Butterfield, D. A., Embley, R. E., Lupton, J. E., Thomson, R. E., and Cannon, G. A.: Observations of manganese and iron at the CoAxial Seafloor Eruption Site, Juan de Fuca Ridge, Geophys. Res. Lett., 22, 151–154, doi:10.1029/94GL02662, 1995.

Matsumoto, K., Tokos, K. S., Price, A. R., and Cox, S. J.: First description of the Minnesota Earth System Model for Ocean biogeochemistry (MESMO 1.0), Geoscientific Model Development, 1, 1–15, doi:10.5194/gmd-1-1-2008, 2008.

Matsumoto, K., Tokos, K., Huston, A., and Joy-Warren, H.: MESMO 2: a Mechanistic Marine Silica Cycle and Coupling to a Simple Terrestrial Scheme, Geoscientific Model Development, 6, 477–494, doi:10.5194/gmd-6-477-2013, 2013.

Mawji, E., Schlitzer, R., Dodas, E. M., Abadie, C., Abouchami, W., Anderson, R. F., Baars, O., Bakker, K., Baskaran, M., Bates, N. R., Bluhm, K., Bowie, A., Bown, J., Boye, M., Boyle, E. A., Branellec, P., Bruland, K. W., Brzezinski, M. A., Bucciarelli, E., Buesseler, K., Butler, E., Cai, P., Cardinal, D., Casciotti, K., Chaves, J., Cheng, H., Chever, F., Church, T. M., Colman, A. S., Conway, T. M., Croot, P. L., Cutter, G. A., de Baar, H. J., de Souza, G. F., Dehairs, F., Deng, F., Dieu, H. T., Dulaquais, G., Echegoyen-Sanz, Y., Edwards, R. L., Fahrbach, E., Fitzsimmons, J., Fleisher, M., Frank, M., Friedrich, J., Fripiat, F., Galer, S. J., Gamo, T., Solsona, E. G., Gerringa, L. J., Godoy, J. M., Gonzalez, S., Grossteffan, E., Hatta, M., Hayes, C. T., Heller, M. I., Henderson, G., Huang, K.-F., Jeandel, C., Jenkins, W. J., John, S., Kenna, T. C., Klunder, M., Kretschmer, S., Kumamoto, Y., Laan, P., Labatut, M., Lacan, F., Lam, P. J., Lannuzel, D., le Moigne, F., Lechtenfeld, O. J., Lohan, M. C., Lu, Y., Masqué, P., McClain, C. R., Measures, C., Middag, R., Moffett, J., Navidad, A., Nishioka, J., Noble, A., Obata, H., Ohnemus, D. C., Owens, S., Planchon, F., Pradoux, C., Puigcorbé, V., Quay, P., Radic, A., Rehkämper, M., Remenyi, T., Rijkenberg, M. J., Rintoul, S., Robinson, L. F., Roeske, T., Rosenberg, M., van der Loeff, M. R., Ryabenko, E., Saito, M. A., Roshan, S., Salt, L., Sarthou, G., Schauer, U., Scott, P., Sedwick, P. N., Sha, L., Shiller, A. M., Sigman, D. M., Smethie, W., Smith, G. J., Sohrin, Y., Speich, S., Stichel, T., Stutsman, J., Swift, J. H., Tagliabue, A., Thomas, A., Tsunogai, U., Twining, B. S., van Aken, H. M., van Heuven, S., van Ooijen, J., van Weerlee, E., Venchiarutti, C., Voelker, A. H., Wake, B., Warner, M. J., Woodward, E. M. S., Wu, J., Wyatt, N., Yoshikawa, H., Zheng, X.-Y., Xue, Z., Zieringer, M., and Zimmer, L. A.: The GEOTRACES Intermediate Data Product 2014, Marine Chemistry, 177, Part 1, 1 – 8, doi:10.1016/j.marchem.2015.04.005, biogeochemistry of trace elements and their isotopes, 2015.

Misumi, K., Lindsay, K., Moore, J. K., Doney, S. C., Tsumune, D., and Yoshida, Y.: Humic substances may control dissolved iron distributions in the global ocean: Implications from numerical simulations, Global Biogeochem. Cycles, 27, 450–462, doi:10.1002/gbc.20039, 2013.



Monod, J.: Microbiologie: Recherches sur la croissance des cultures bactériennes. I, Actualités scientifiques et industrielles, Hermann & cie, 1942.

Moore, C. M., Mills, M. M., Arrigo, K. R., Berman-Frank, I., Bopp, L., Boyd, P. W., Galbraith, E. D., Geider, R. J., Guieu, C., Jaccard, S. L., Jickells, T. D., Roche, J. L., Lenton, T. M., Mahowald, N. M., Maranon, E., Marinov, I., Moore, J. K., Nakatsuka, T., Oschlies, A., Saito, M. A., Thingstad, T. F., Tsuda, A., and Ulloa, O.: Processes and patterns of oceanic nutrient limitation, Nature Geoscience, 6, 701–710, doi:10.1038/NGEO1765, 2013.

Moore, J., Doney, S. C., Glover, D. M., and Fung, I. Y.: Iron cycling and nutrient-limitation patterns in surface waters of the World Ocean, Deep Sea Research Part II: Topical Studies in Oceanography, 49, 463 – 507, doi:10.1016/S0967-0645(01)00109-6, the US JGOFS Synthesis and Modeling Project: Phase 1, 2001.

Moore, J. K. and Braucher, O.: Sedimentary and mineral dust sources of dissolved iron to the world ocean, Biogeosciences, 5, 631–656, doi:10.5194/bg-5-631-2008, 2008.

Moore, J. K., Doney, S. C., and Lindsay, K.: Upper ocean ecosystem dynamics and iron cycling in a global three-dimensional model, Global Biogeochem. Cycles, 18, GB4028, doi:10.1029/2004GB002220, 2004.

Morel, A. and Smith, R. C.: Relation between total quanta and total energy for aquatic photosynthesis, Limnology and Oceanography, 19, 591–600, doi:10.4319/lo.1974.19.4.0591, 1974.

Najjar, R. G., Sarmiento, J. L., and Toggweiler, J. R.: Downward transport and fate of organic matter in the ocean: Simulations with a general circulation model, Global Biogeochem. Cycles, 6, 45–76, doi:10.1029/91GB02718, 1992.

Najjar, R. G., Jin, X., Louanchi, F., Aumont, O., Caldeira, K., Doney, S. C., Dutay, J.-C., Follows, M., Gruber, N., Joos, F., Lindsay, K., Maier-Reimer, E., Matear, R. J., Matsumoto, K., Monfray, P., Mouchet, A., Orr, J. C., Plattner, G.-K., Sarmiento, J. L., Schlitzer, R., Slater, R. D., Weirig, M.-F., Yamanaka, Y., and Yool, A.: Impact of circulation on export production, dissolved organic matter, and dissolved oxygen in the ocean: Results from Phase II of the Ocean Carbon-cycle Model Intercomparison Project (OCMIP-2), Global Biogeochem. Cycles, 21, doi:10.1029/2006GB002857, GB3007, 2007.

NASA Goddard Space Flight Center, Ocean Ecology Laboratory, O. B. P. G.: MODIS-Aqua Ocean Color Data, doi:10.5067/AQUA/MODIS_OC.2014.0, 2014.

National Geophysical Data Center: ETOPO2v2 2-minute Global Relief Model, doi:10.7289/V5J1012Q, 2006.

Nelson, D. M., Tréguer, P., Brzezinski, M. A., Leynaert, A., and Quéguiner, B.: Production and dissolution of biogenic silica in the ocean: Revised global estimates, comparison with regional data and relationship to biogenic sedimentation, Global Biogeochem. Cycles, 9, 359–372, doi:10.1029/95GB01070, 1995.

Nickelsen, L. and Oschlies, A.: Enhanced sensitivity of oceanic $CO_2$ uptake to dust deposition by iron-light colimitation, Geophys. Res. Lett., 42, 492–499, doi:10.1002/2014GL062969, 2014GL062969, 2015.

Parekh, P., Follows, M. J., and Boyle, E.: Decoupling of iron and phosphate in the global ocean, Global Biogeochem. Cycles, 19, GB2020, doi:10.1029/2004GB002280, 2005.

Pasquier, B. and Holzer, M.: The plumbing of the global biological pump: Efficiency control through leaks, pathways, and time scales, Journal of Geophysical Research: Oceans, 121, 6367–6388, doi:10.1002/2016JC011821, 2016.

Primeau, F. W., Holzer, M., and DeVries, T.: Southern Ocean nutrient trapping and the efficiency of the biological pump, J. Geophys. Res., 118, 2547–2564, doi:10.1002/jgrc.20181, 2013.

Rhee, G.-Y.: Effects of N:P atomic ratios and nitrate limitation on algal growth, cell composition, and nitrate uptake 1, Limnology and Oceanography, 23, 10–25, doi:10.4319/lo.1978.23.1.0010, 1978.



Saito, M. A., Goepfert, T. J., and Ritt, J. T.: Some thoughts on the concept of colimitation: Three definitions and the importance of bioavailability, Limnology and Oceanography, 53, 276–290, doi:10.4319/lo.2008.53.1.0276, 2008.

Sarmiento, J. L., Simeon, J., Gnanadesikan, A., Gruber, N., Key, R. M., and Schlitzer, R.: Deep ocean biogeochemistry of silicic acid and nitrate, Global Biogeochem. Cycles, 21, GB1S9, doi:10.1029/2006GB002720, 2007.

Smith, S. V.: Phosphorus versus nitrogen limitation in the marine environment, Limnology and Oceanography, 29, 1149–1160, doi:10.4319/lo.1984.29.6.1149, 1984.

Strzepek, R. F., Hunter, K. A., Frew, R. D., Harrison, P. J., and Boyd, P. W.: Iron-Light Interactions Differ in Southern Ocean Phytoplankton, Limnol. Oceanogr., 57, 1182–1200, doi:10.4319/lo.2012.57.4.1182, 2012.

Sunda, W. G. and Huntsman, S. A.: Interrelated Influence of Iron, Light and Cell Size on Marine Phytoplankton Growth, Nature, 390, 389–392, doi:10.1038/37093, 1997.

Tagliabue, A., Mtshali, T., Aumont, O., Bowie, A. R., Klunder, M. B., Roychoudhury, A. N., and Swart, S.: A global compilation of dissolved iron measurements: focus on distributions and processes in the Southern Ocean, Biogeosciences, 9, 2333–2349, doi:10.5194/bg-9-2333-2012, 2012.

Tagliabue, A., Aumont, O., and Bopp, L.: The impact of different external sources of iron on the global carbon cycle, Geophysical Research Letters, 41, 920–926, doi:10.1002/2013GL059059, 2013GL059059, 2014.

Tagliabue, A., Aumont, O., DeAth, R., Dunne, J. P., Dutkiewicz, S., Galbraith, E., Misumi, K., Moore, J. K., Ridgwell, A., Sherman, E., Stock, C., Vichi, M., Völker, C., and Yool, A.: How Well do Global Ocean Biogeochemistry Models Simulate Dissolved Iron Distributions?, Global Biogeochem. Cycles, 30, 149–174, doi:10.1002/2015gb005289, 2016.

Takeda, S.: Influence of Iron Availability on Nutrient Consumption Ratio of Diatoms in Oceanic Waters, Nature, 393, 774–777, doi:10.1038/31674, 1998.

Teng, Y., Primeau, F. W., Moore, J. K., Lomas, M. W., and Martiny, A.: Global-scale variations of the ratios of carbon to phosphorus in exported marine organic matter, Nature Geosci., 7, 895–898, doi:10.1038/NGEO2303, 2014.

Trenberth, K. E., Large, W. G., and Olson, J. G.: A global ocean wind stress climatology based on ECMWF analyses, Tech. Rep. TN-3381STR, National Center for Atmospheric Research, doi:10.5065/D6ST7MR9, 1989.

Twining, B. S., Nodder, S. D., King, A. L., Hutchins, D. A., LeCleir, G. R., DeBruyn, J. M., Maas, E. W., Vogt, S., Wilhelm, S. W., and Boyd, P. W.: Differential remineralization of major and trace elements in sinking diatoms, Limnol. Oceanogr., 59, 689–704, doi:10.4319/lo.2014.59.3.0689, 2014.

Uitz, J., Claustre, H., Gentili, B., and Stramski, D.: Phytoplankton class?specific primary production in the world's oceans: Seasonal and interannual variability from satellite observations, Global Biogeochem. Cycles, 24, GB3016, doi:10.1029/2009GB003680, 2010.

Weber, T., Cram, J. A., Leung, S. W., DeVries, T., and Deutsch, C.: Deep ocean nutrients imply large latitudinal variation in particle transfer efficiency, Proc. Natl. Acad. Sci., 113, 8606–8611, doi:10.1073/pnas.1604414113, 2016.