# Peer review of "Inverse-model estimates of the ocean's coupled phosphorus, silicon, and iron cycles"

_Biogeosciences, 2017_

## Referee Comment (RC1) · Anonymous Referee #1 · 11 May 2017

This manuscript presents the formulation of a global biogeochemistry-ocean circulation model that considers the phosphorus, silica, and iron cycles. Results are presented from a family of solutions that fit the data (dissolved phosphate, silicate, iron, and phytoplankton distributions) equally well but explore the sensitivity to the unconstrained external iron sources to the ocean. Metrics related to global carbon and opal export, limiting nutrients, and iron based export production patterns are presented. The presented modeling framework is at the state-of-the-art for building a 3D global biogeochemical model with the solution computed in offline mode and is of high interest to the ocean modeling and marine biogeochemistry communities. The main advance of the work is to show that the global biogenic carbon and opal exports are well constrained

by the available nutrient and satellite phytoplankton data even though the external and internal ocean iron fluxes are not. The family of most probable model solutions given the sensitivity in assumptions on iron cycling mostly converge on 10 Pg C yr-1 and 170 Tmol Si yr-1 global exports.

My two main comments concern the sensitivity of their calculated global carbon export flux to their omission of DOM cycling and variable C:P stoichiometry in organic matter production/export. DOC has been estimated to contribute 20% (Hansell et al., 2009, Oceanography) to 25% (Letscher et al., 2015, Biogeosciences) of global carbon export production. In the model presented by the authors they chose to omit DOP cycling, with their argument being that DOP cycling represents a small to negligible contribution to the biological phosphorus cycle. They also rationalize that DOP typically has lifetimes <1 year in surface waters such that it is not significantly advected with the ocean circulation and can instead focus on vertical redistribution of particles as the dominant export process in their model. However DOC has longer lifetimes in surface waters on the order of a couple years and does accumulate to large enough quantities to be an important part of the carbon export term. Can the authors address the sensitivity of their calculated global carbon export flux to this omission of DOM cycling in their model? Should DOC export be considered as an addition to the computed ∼10 Pg C yr-1 flux? Or is the DOC export flux somehow already included in their computations from their model solution?

Secondly, recent global datasets and model inversions of nutrient data have shown/predicted that the production and export of organic matter from the surface ocean is not constant and exhibits latitudinal and ocean biome-level variability (e.g. Martiny et al., 2013, Nature Geoscience; Teng et al., 2014, Nature Geoscience; Devries & Deutsch, 2014, Nature Geoscience; Galbraith & Martiny, 2015, PNAS). The authors chose to calculate all of their carbon export metrics using a constant Redfield ratio of 106:1 C:P to get carbon units from their model which is in phosphorus units. How much would their global estimates of carbon export change if a variable C:P of

organic matter production/export were used? For example, a regionally variable C:P could be computed using the relationship predicted with surface phosphate conditions from Galbraith & Martiny, 2015 (PNAS) using the model simulated phosphate fields. Alternatively, the twelve-biome inferred export C:P ratios from Teng et al., 2014 (Nature Geoscience) could be used to calculate the regionally variable C:P of export from the authors model. It seems given what we now know vis-à-vis regionally variable organic matter stoichiometry, it would be remiss not to include that knowledge to update the global C export flux from the authors' model solution.

Other comments:

Pg 20 L1-5: The authors blame phytoplankton biomass mismatches between the model and satellite observations based on a lack of seasonality in the model but aren't these steady-state satellite climatologies they are comparing against, and therefore seasonality is averaged over?

Pg 23 L20-25: The authors argue that the sharper meridional gradient in C export is more realistic because there is a sharp gradient in satellite NPP. But they don't include DOC export. DOC export is estimated to be 1/5 to 1/4 of total global C export with its larger contribution occurring in the subtropical gyre systems due to large-scale downwelling.

Pg 27 L24-30. One statement says the dominant Fe sink is from POP scavenging. The very next statement says that opal scavenging accounts for half of Fe sinks. The next statement says that dust scavenging is negligible for Fe sinks. Why not just say that POP and opal scavenging account about equally to Fe sinks?

---

## Referee Comment (RC2) · Anonymous Referee #2 · 20 Jun 2017

The submission by Pasquier and Holzer uses a new ocean bioegeochemical and ecosystem model, embedded within a data-constrained steady-state circulation model, to explore linkages between the phosphorous, silicon, and iron cycles. The model uses interesting methods to represent nutrient uptake by multiple-species phytoplankton communities without explicitly resolving their biomass, allowing for efficient simulations and parameter optimization. Based on previous work, the authors understand that no single optimal solution for the Fe cycle can be obtained because certain source and sink processes have overlapping effects on the Fe distribution. They therefore explore a "family" of solutions with different source strengths, which are independently optimized and then compiled into a "typical" solution and uncertainty range. There are a number of interesting outcomes that are robust across the family of solutions, for

example the patterns of phosphorous export supported by each iron source and the "efficiency" of each source at supporting export. Atmospheric Fe supports most export relative to the magnitude of its source, followed by benthic and then hydrothermal Fe.

I think this paper takes an interesting approach and has the potential to be a valuable contribution to the literature. Nevertheless, I have two main critiques of the paper in its current form. First, the paper lacks a clear direction from the outset. The introduction does not lay out any specific questions or hypotheses that the new model is designed to address, nor does it identify the particular gaps in our understanding of the Fe cycle that the authors aim to close. Instead, the goal is simply states as "to constrain a model of the coupled nutrient cycles by optimizing the biogeochemical parameters against available observations", which does not seem like a strong motivation. The purpose of an inverse model should be to extract new information from the available observations, not just to match the observations. The authors should begin by clearly stating what new information they aim to extract by explicitly simulating the coupling of Si, P and Fe, relative to their previous work.

This same mentality extends throughout the paper, where numerous model-data comparisons are presented without properly highlighting what new has been learned in the process. For example, one of the key benefits of this coupled model is the ability to assess the relative Fe-scavenging efficiency of different particle types (organic, silica, dust), which remains an open question in Fe biogeochemistry. While this result is part of the model solution, it receives very little attention in the text – it is briefly noted that on a global basis, organic matter and silica are equally responsible for Fe removal from the ocean, and a figure is shown in the Appendix. But the authors should discuss which particle type is the stronger Fe scavenger on a per-gram basis, whether this is robust across the family of solutions. This would be a new interesting result of this study.

As another example, the new model seems to be the ideal tool for examining differences in Fe quotas among phytoplankton types – another open question in Fe cycle research. The authors briefly mention that they experimented with different Fe quotas, but abandoned the approach when the parameters converged to similar values. If the model selects similar Fe quotas for all plankton groups, and this is robust across the whole family of solutions, it would be an interesting result indeed and worth of some attention in the paper! Especially if the authors could demonstrate that there is no evidence for enhanced Fe quotas in subtropical gyres where diazotrophic plankton are common, given that there is ongoing debate about the relative Fe requirements of N-fixing and non-fixing plankton.

My second main critique of the paper is that it doesn't present the model-data comparisons for Fe that would be best suited to support the conclusions. If one of the main goals of the paper is to understand the relative contribution of each Fe source to organic matter export, one would want to show that the model accurately reproduces the locations and transport trajectories of the sources. By design, many of the GEO-TRACES transects sampled different source regions of Fe, and show clear signatures of these sources and their transport across basins. For example, GA03 and GP16 both traverse benthic and hydrothermal source regions. Plotting cross-sections of modeled and observed Fe along these transects would give a clearer visual impression of the model's performance than the summary statistics and basin-wide profiles that are presented. The reader would want to ensure that these source signatures and transport trajectories are well reproduced, before considering the export contribution analysis.

In addition, I have the following minor comments:

1. I agree with Reviewer #1 that caveats of neglecting DOP cycling need to be more carefully considered. Ignoring DOP will not only bias the total estimated export, but also its pattern and therefore potentially the contribution of different Fe sources to export. Particularly, DOP convergence is thought to provide a significant P supply to subtropical gyres, and essentially "relocates" export downstream, from tropical and coastal upwelling zones into the gyres. Given that benthic Fe supports most export in upwelling zones and atmospheric Fe supports most export in gyres, relocating exporting between those two regimes seems important.

2. What is the justification for choosing such widely different export ratios between plankton types (page 5, line 13)? The authors cite Dunne 2005, but there have been other studies since (e.g. Richardson 2007) that suggest small plankton contribute as much export, relative to their NPP, as large plankton.

3. What is the justification for not prescribing a minimum Fe:P quota in equation 14? It is impossible to sustain phytoplankton growth with no Fe, so if the model is optimizing towards zero it means that model is straying into unrealistic parameter space, not that this parameter can be neglected. The authors should set a reasonable lower limit during the optimization (e.g. low end of the range shown in Moore et al 2013), rather than allowing the Fe quota to approach zero at low [Fe].

---

## Author Response (AR1)

**Authors' response on* "Inverse-model estimates of the ocean's coupled phosphorus, silicon, and iron cycles" *by* Benoît Pasquier and Mark Holzer**

Benoît Pasquier and Mark Holzer

July 24, 2017

Submitted manuscript for discussion available at:
https://doi.org/10.5194/bg-2017-122

Dear Dr. Slomp,

Below please find the comments from the two Reviewers reproduced in bold followed by our point-by-point replies along with the corresponding changes to the manuscript in blue (page numbers refer to the revised version).

The manuscript version with tracked changes appended here shows the deleted parts struck through and in red and the revised parts in blue.

In addition to the revisions in response to the Reviewers, we have made minor revisions to help the reader interpret Figures 8 and 10: We have added fits to simple theoretical curves with some corresponding discussion on the efficiency of each iron type in supporting export (pages 28–34). We believe this strengthens one of the main points of our manuscript, namely that per source-injected molecule aeolian iron supports more phosphorus and opal export than sedimentary and hydrothermal iron. We have correspondingly edited the abstract and the conclusions.

**Reply to comments by Reviewer #1**

We appreciate the referee's comments. The referee's main points concern our statements on carbon export, specifically the lack of DOC and constant C:P stoichiometry. Before we address these in detail, we would like to emphasize that we model the phosphorus, silicon, and iron cycles and not the carbon cycle. Carbon export is plotted merely to show the phosphorus export in convenient units that make it easy to compare to other model and observational estimates. We therefore used a constant C:P ratio as a convenient unit conversion not intended to quantitatively estimate carbon export. The advantage of a constant Redfield ratio is that it does not distort the pattern of the phosphorus export. While we did point out that the carbon export flux is merely meant as a conversion of the phosphorus export after Eq. (27), we should have been more explicit that we are discussing phosphorus export in the abstract and in some of the figure discussions. In response, we have revised the manuscript (especially the abstract) to make clear that we quantify phosphorus export and to mention throughout that the plots of carbon export show phosphorus export expressed in carbon units.

The referee's comments raise the interesting questions as to how much dissolved organic matter transport is implicitly captured by our approach, and how different the inferred global carbon export would be if one used more careful estimates of the C:P ratio for the export of particulate and dissolved organic matter. We will address these issues in our point-by-point responses below.

**This manuscript presents the formulation of a global biogeochemistry-ocean circulation model that considers the phosphorus, silica, and iron cycles. Results are presented from a family of solutions that fit the data (dissolved phosphate, silicate, iron, and phytoplankton distributions) equally well but explore the sensitivity to the unconstrained external iron sources to the ocean. Metrics related to global carbon and opal export, limiting nutrients, and iron based export production patterns are presented. The presented modeling framework is at the state-of-the-art for building a 3D global biogeochemical model with the solution computed in offline mode and is of high interest to the ocean modeling and marine biogeochemistry communities. The main advance of the work is to show that the global biogenic carbon and opal exports are well constrained by the available nutrient and satellite phytoplankton**

**data even though the external and internal ocean iron fluxes are not. The family of most probable model solutions given the sensitivity in assumptions on iron cycling mostly converge on 10 Pg C yr-1 and 170 Tmol Si yr-1 global export.**

**My two main comments concern the sensitivity of their calculated global carbon export flux to their omission of DOM cycling and variable C:P stoichiometry in organic matter production/export. DOC has been estimated to contribute 20% (Hansell et al., 2009, Oceanography) to 25% (Letscher et al., 2015, Biogeosciences) of global carbon export production. In the model presented by the authors they chose to omit DOP cycling, with their argument being that DOP cycling represents a small to negligible contribution to the biological phosphorus cycle. They also rationalize that DOP typically has lifetimes <1 year in surface waters such that it is not significantly advected with the ocean circulation and can instead focus on vertical redistribution of particles as the dominant export process in their model. However DOC has longer lifetimes in surface waters on the order of a couple years and does accumulate to large enough quantities to be an important part of the carbon export term. Can the authors address the sensitivity of their calculated global carbon export flux to this omission of DOM cycling in their model? Should DOC export be considered as an addition to the computed $\sim$10 Pg C yr$^{-1}$ flux? Or is the DOC export flux somehow already included in their computations from their model solution?**

Our estimates of phosphorus export effectively capture the export of both POP and DOP, despite DOP not being explicitly represented as a separate tracer. This is because the phosphate export of our model is set by the strength of $PO_4$ uptake and by the Martin exponent $b$ of the remineralization profile, both of which are optimized by minimizing the mismatch between modelled and observed $PO_4$ concentrations. Because the real ocean's (regenerated) $PO_4$ concentrations result from both POP and DOP export, the effect of DOP export on the remineralization field is implicitly accounted for in our optimized states. In response, we have added a short statement on the fact that the effects of DOP are captured by our model on page 5.

We agree that the lack of explicit representation of DOC results in an underestimation of the carbon export as inferred from the phosphorus export. We estimate the "missing" carbon export by explicitly calculating both POP and

(semilabile) DOP export using the data-assimiled phosphorus cycle of Primeau et al. (JGR, 2013), which carries explicit $PO_4$ and DOP tracers and uses the same circulation as in our manuscript. (The DOP export was computed as the aphotic remineralization rate of DOP produced in the euphotic layer.) We find a DOP export of $0.70 \, \text{Tmol P yr}^{-1}$ and a total phosphorus export of $5.79 \, \text{Tmol P yr}^{-1}$. If one applies C:P=106:1 to the total phosphorus export, this converts to a total carbon export of $7.37 \, \text{Pg C yr}^{-1}$. However, if one uses C:P=225:1 for DOP (as determined by the DOM OPT simulation of Letscher et al. (2015) for semilabile DOM), the total carbon export becomes $8.37 \, \text{Pg C yr}^{-1}$. Thus, by not representing DOC explicitly, one "misses" $\sim$12% of the total carbon export. (In this calculation, the DOC export represents 23% of the total carbon export, consistent with the estimate of Letscher et al., (2015).) In response, we have added a paragraph briefly discussing the fact that phosphorus export converted to carbon units underestimates carbon export by roughly 12% due to the missing DOC export after Equation (27), page 23–24.

While we agree that accounting for DOC export gives more realistic estimates for the total carbon export, we reiterate that we use our model to calculate phosphorus export and merely convert to carbon units when we think this is useful (e.g., for comparison to the results of Primeau et al., 2013).

**Secondly, recent global datasets and model inversions of nutrient data have shown/predicted that the production and export of organic matter from the surface ocean is not constant and exhibits latitudinal and ocean biome-level variability (e.g. Martiny et al., 2013, Nature Geoscience; Teng et al., 2014, Nature Geoscience; Devries & Deutsch, 2014, Nature Geoscience; Galbraith & Martiny, 2015, PNAS). The authors chose to calculate all of their carbon export metrics using a constant Redfield ratio of 106:1 C:P to get carbon units from their model which is in phosphorus units. How much would their global estimates of carbon export change if a variable C:P of organic matter production/export were used? For example, a regionally variable C:P could be computed using the relationship predicted with surface phosphate conditions from Galbraith & Martiny, 2015 (PNAS) using the model simulated phosphate fields. Alternatively, the twelve-biome inferred export C:P ratios from Teng et al., 2014 (Nature Geoscience) could be used to calculate the regionally variable C:P of export from the authors model. It seems given what we now know vis-à-vis regionally variable organic matter stoichiometry, it would be remiss not to include that knowledge to update the global C export flux from the authors' model solution.**

We agree with the referee that if the objective were to provide an accurate estimate of carbon export, using recently constrained regionally varying C:P ratios would be more realistic. However, using the regionally varying C:P ratios suggested by the referee has only modest effects on the globally integrated carbon export:

(i) Applying the P:C relation of Galbraith and Martiny (2015) to the phosphate export of our typical state gives a carbon export of $8.5 \pm 0.4\,\mathrm{Pg\,C\,yr^{-1}}$ or $9.4 \pm 0.9\,\mathrm{Pg\,C\,yr^{-1}}$ when we use their log-binned parameter values.

(ii) Applying the regional C:P inverse-model estimates of Teng et al. (2014) gives a carbon export of $10. \pm 2.\,\mathrm{Pg\,C\,yr^{-1}}$. Both this estimate and the one based on the log-binned regression agree within their uncertainties with our simple unit-conversion value of $10.3 \pm 0.4\,\mathrm{Pg\,C\,yr^{-1}}$.

In response, we have added some discussion of these calculations to the caveats section where we discuss regionally variable C:P ratios (page 36). We have also revised the manuscript to make explicit throughout that we compute phosphorus export converted to carbon units.

**Other comments:**

**Pg 20 L1-5: The authors blame phytoplankton biomass mismatches between the model and satellite observations based on a lack of seasonality in the model but aren't these steady-state satellite climatologies they are comparing against, and therefore seasonality is averaged over?**

We are alluding to the fact that there is a difference between the fields produced by a steady-state model and the long-term average of a seasonally varying field. (For example, the circulation we use is *not* a simple annual-mean circulation, but rather a steady-state circulation whose transport reproduces the annually averaged tracers used in the assimilation.) However, we agree that this passage could have been clearer. In response, concerning the mismatch at ∼60°S, we have revised the manuscript to point out that the strong seasonality of the Southern Ocean, with its large seasonal cycle in sea-ice coverage, is not captured by our model and that this could conceivably be the reason for the mismatches in that region (page 43). We have deleted the second mention of seasonality in this paragraph, which was unintentional repetition.

**Pg 23 L20-25: The authors argue that the sharper meridional gradient in C export is more realistic because there is a sharp gradient in satellite NPP. But they don't include DOC export. DOC export is estimated to be 1/5 to 1/4 of total global C export with its larger contribution occurring in the subtropical gyre systems due to large-scale downwelling.**

We agree that the inclusion of DOC would result in more realistic estimates of carbon export. However, in this passage we are comparing our phosphorus export (converted to C units) to POC export as estimated from satellite-derived NPP, and by definition POC export does not include DOC export. In response, we have revised this passage to make clear that we are comparing phosphorus export converted to carbon units, and to state that sharp meridional gradients can also be seen in the satellite-derived NPP (pages 23–24). We have deleted the statement about the sharp gradients being "more realistic" to avoid confusion.

**Pg 27 L24-30. One statement says the dominant Fe sink is from POP scavenging. The very next statement says that opal scavenging accounts for half of Fe sinks. The next statement says that dust scavenging is negligible for Fe sinks. Why not just say that POP and opal scavenging account about equally to Fe sinks?**

Thank you for catching this. (There was some left-over wording from a previous iteration.) We have revised as suggested (page 26).

**Reply to comments by Reviewer #2**

Reviewer 2 raises some interesting points (reproduced here in bold) to which we respond in detail below. The Reviewer also points out that our Introduction section did not explicitly articulate the main science questions addressed by our study, which can be easily remedied in the revised manuscript. Reviewer 2 then asks if our model can be used to constrain class-dependent Fe quotas and the scavenging efficiency of different particle types. However, these quantities are not robustly constrainable within our framework. Moreover, these questions are tangential to our study whose main goal is to obtain data-constrained estimates of the coupled iron–macronutrient cycles and to elucidate the relative importance of the different iron sources for supporting export production. The Reviewer also makes some "minor" points regarding detrital fractions and a minimum iron quota. Thinking about these helped us deepen our understanding and improve our manuscript.

**The submission by Pasquier and Holzer uses a new ocean bioegeochemical and ecosystem model, embedded within a data-constrained steady-state circulation model, to explore linkages between the phosphorous, silicon, and iron cycles. The model uses interesting methods to represent nutrient uptake by multiple-species phytoplankton communities without explicitly resolving their biomass, allowing for efficient simulations and parameter optimization. Based on previous work, the authors understand that no single optimal solution for the Fe cycle can be obtained because certain source and sink processes have overlapping effects on the Fe distribution. They therefore explore a "family" of solutions with different source strengths, which are independently optimized and then compiled into a "typical" solution and uncertainty range. There are a number of interesting outcomes that are robust across the family of solutions, for example the patterns of phosphorous export supported by each iron source and the "efficiency" of each source at supporting export. Atmospheric Fe supports most export relative to the magnitude of its source, followed by benthic and then hydrothermal Fe.**

We would like to point out that our work additionally shows that part of the uncertainty in the iron sources comes from the fact that, for very similar overall mismatch with observations, the three sources of iron can compensate for

each other. We now explicitly make this point in the revised Introduction section.

**I think this paper takes an interesting approach and has the potential to be a valuable contribution to the literature. Nevertheless, I have two main critiques of the paper in its current form. First, the paper lacks a clear direction from the outset. The introduction does not lay out any specific questions or hypotheses that the new model is designed to address, nor does it identify the particular gaps in our understanding of the Fe cycle that the authors aim to close. Instead, the goal is simply states as "to constrain a model of the coupled nutrient cycles by optimizing the biogeochemical parameters against available observations", which does not seem like a strong motivation. The purpose of an inverse model should be to extract new information from the available observations, not just to match the observations. The authors should begin by clearly stating what new information they aim to extract by explicitly simulating the coupling of Si, P and Fe, relative to their previous work.**

We agree that the Introduction section could have more clearly articulated the key points of our paper. The purpose of the inverse model is definitely not to build a model and compare to observations. In response, we have revised the Introduction section, which now clearly states that our paper makes the following advances:

A. We are building an inverse model of the coupled Fe, P, and Si cycles so that the macronutrients and organic matter export can respond to changes in the iron supply. This is a key advance over the work of *Frants et al.* [2016] where the phosphate cycle was prescribed. Our new inverse model provides, for the first time, a family of data-constrained state estimates of the coupled Fe-P-Si cycles for a wide range of not only aeolian, but also hydrothermal and sedimentary sources. Analysis of this family of estimates allows us to show that the uncertainty in the iron sources stems not only from compensation between sources and scavenging sinks, but also from the fact that the different types of iron sources (aeolian, hydrothermal, and sedimentary) can compensate each other.

B. We use our state estimates to address an important open question about the marine iron cycle: What are the relative contributions of the different iron sources to supporting the world ocean's export production? While there have

been perturbation experiments with forward models where one type of source (e.g., hydrothermal or sedimentary) was shut down to assess the importance of dFe from the missing source [e.g., *Tagliabue et al.*, 2009, 2010, 2014], such experiments cannot quantify the true contribution of hydrothermal or sedimentary iron to biological production because of the nonlinearities of the iron cycle [*Holzer et al.*, 2016]. Moreover such experiments were conducted with definite but highly uncertain choices of the iron sources, and the models were not objectively constrained by the available observational data. Thus, in addition to presenting the first inverse model of the coupled Fe-P-Si cycle, the central scientific objectives of our study are to answer the following key questions:

1. How well can the observed dFe, $PO_4$, and $Si(OH)_4$ concentrations be fitted to observations for widely differing iron sources, and are there limits on the iron source strengths that are consistent with the observed dFe concentrations?

2. What are the limitation patterns that emerge from the data-constrained estimates of the coupled nutrient cycles, given that direct observational data on these patterns is very sparse?

3. How well constrained are the phosphorus and opal exports for optimized state estimates with widely different iron sources?

4. What fractions of phosphorus and opal export are supported by aeolian, hydrothermal, and sedimentary iron, and how do these fractions vary with the relative iron source strengths?

We think that the underlying theme and science objectives of our work should now be clear to the reader.

In addition, we have added a "road map" paragraph at the end of the Introduction section, so that the reader knows what to expect for the rest of the paper even without looking at the section headings.

**This same mentality extends throughout the paper, where numerous model-data comparisons are presented without properly highlighting what new has been learned in the process. For example, one of the key benefits of this coupled model is the ability to assess the relative Fe-scavenging efficiency of different particle types (organic, silica, dust), which remains an open question in Fe biogeochemistry. While this result is part of the**

**model solution, it receives very little attention in the text — it is briefly noted that on a global basis, organic matter and silica are equally responsible for Fe removal from the ocean, and a figure is shown in the Appendix. But the authors should discuss which particle type is the stronger Fe scavenger on a per-gram basis, whether this is robust across the family of solutions. This would be a new interesting result of this study.**

We would like to point out that we showed only three figures (joint pdfs, mean iron profiles, and phytoplankton concentrations) that compare model against observations, and we note that the Reviewer requests a yet more detailed comparison with GEOTRACES sections below. These comparisons are not made to answer new science questions per se, only to quantify the degree to which the dFe and macronutrient concentrations can be matched to the observations.

Regarding the suggestion to delve further into the scavenging efficiency of different types of particles (POP, opal, and dust), the partitioning of the scavenging among the different particle types is not something that can be constrained robustly from our inverse model. The scavenging by one particle type can be compensated by another particle type because of overlap in the spatial pattern of their fluxes. The nutrient and phytoplankton data used do not provide separate constraints on the scavenging by each particle type, only on the total amount of scavenging. For these reasons this aspect of our model is not a focus in our manuscript.

Nevertheless, one may of course ask how the partition among particles types varies across our family of estimates. We find that scavenging by dust is negligible for all our state estimates, while the fraction scavenged by POP ranges from ∼10% for the lowest iron source cases and saturates near ∼100% for the highest iron sources considered. (The remaining fraction is due to opal scavenging.)

In response to this comment, we have added a brief discussion that states that the partition among particle types is likely not robust, and that details the systematic dependence on the total iron source strength (page 27).

**As another example, the new model seems to be the ideal tool for examining differences in Fe quotas among phytoplankton types — another open question in Fe cycle research. The authors briefly mention that they ex-**

**perimented with different Fe quotas, but abandoned the approach when the parameters converged to similar values. If the model selects similar Fe quotas for all plankton groups, and this is robust across the whole family of solutions, it would be an interesting result indeed and worth of some attention in the paper! Especially if the authors could demonstrate that there is no evidence for enhanced Fe quotas in subtropical gyres where diazotrophic plankton are common, given that there is ongoing debate about the relative Fe requirements of N-fixing and non-fixing plankton.**

Reviewer 2 is correct that we do not distinguish the Fe:P parameters of different functional classes based on experiments where we optimized class-dependent Fe:P parameters. While we agree that establishing any differences in Fe quotas among N-fixing and non-fixing plankton is an interesting open question, we would like to remind the Reviewer that we do not model the nitrogen cycle at all so that this issue is beyond the scope of our paper.

We cannot use our inverse model to robustly constrain different values of $R_0^{\mathrm{Fe:P}}$ and $k_{\mathrm{Fe:P}}$ for each class because these six parameters (two per class) would directly compensate for one another in the global Fe export. We therefore decided to use only the two class-independent parameters, $R_0^{\mathrm{Fe:P}}$ and $k_{\mathrm{Fe:P}}$, and optimized them sequentially. Constraining class-dependent Fe quota is beyond the scope of what is possible with our inverse model, which we now explicitly point out in the revised manuscript (page 9).

**My second main critique of the paper is that it doesn't present the model-data comparisons for Fe that would be best suited to support the conclusions. If one of the main goals of the paper is to understand the relative contribution of each Fe source to organic matter export, one would want to show that the model accurately reproduces the locations and transport trajectories of the sources. By design, many of the GEOTRACES transects sampled different source regions of Fe, and show clear signatures of these sources and their transport across basins. For example, GA03 and GP16 both traverse benthic and hydrothermal source regions. Plotting cross-sections of modeled and observed Fe along these transects would give a clearer visual impression of the model's performance than the summary statistics and basin-wide profiles that are presented. The reader would want to ensure that these source signa-**

**tures and transport trajectories are well reproduced, before considering the export contribution analysis.**

The Reviewer is correct that a key point of our analysis is to quantify the relative contribution of each iron type to organic matter export and to explore the systematics of these contributions over a wide range of iron source strengths. While source patterns and transport are important for this, we do not think one can reasonably expect our model to reproduce the GEOTRACES transects with complete fidelity. First, we use a coarse-resolution, steady-state inverse model, while the GEOTRACES sections provide snapshots in space and time. Therefore, as we point out in our manuscript, our model cannot capture any transient plumes (e.g., from an African dust event) that are highly localized and episodic. Our state estimates can only capture the long-term average concentration, coarse-grained to $2° \times 2°$ resolution. Second, we had only the Intermediate Data Product available when developing this model. Pacific features are therefore only constrained from the older data compilation by *Tagliabue et al.* [2012]. In terms of capturing hydrothermal plumes, we note that our model uses a data-assimilated circulation, but this circulation only assimilated $T$, $S$, $PO_4$, and $^{14}C$ but not $^3He$. Therefore, there are likely still some biases in the abyssal circulation, which contribute to the fact that we do not perfectly match the observed hydrothermal iron plumes. However, what matters for our analysis is the large-scale transport into the euphotic zone, particularly the transport into iron-limited regions such as the Southern Ocean. We have no reason to think that this large-scale transport is suspect as evidenced by realistic large-scale patterns of production that are robust across a family of states with widely varying iron source strengths. We emphasize that production in our inverse model is mechanistically driven by dFe and macronutrient availability.

Of course, we are happy to show a direct comparison with the GEOTRACES sections subject to the caveats discussed above. Figure 1 herein compares the main transects included in the Intermediate Data Product with our typical state estimate. The coarse resolution model does capture the large-scale features, but localized high concentrations cannot be captured at our resolution.

We respectfully disagree that we did not present the model–data comparison best suited to supporting our conclusions. We think that Figures 1d, 2c, and 3 of our manuscript are the most relevant and appropriate *quantitative* comparisons between estimated and observed dFe, given that essentially raw bottle data is compared with a coarse-resolution steady-state model. Figure 1d plots the

RMS cost-weighted mismatch for the whole family of estimates. This shows how varying iron source strengths affects our ability to match dFe observations. Figure 2c plots the cost-weighted joint distribution of modelled and observed dFe for our typical estimate. This shows just how difficult it is to match the sparse dFe observations, at least compared to the macronutrients for which gridded climatologies are available. Figure 3 shows the basin dFe profiles of each family member. Because these profiles average a large number of observations, they provide a robust metric for assessing the model's ability to capture the large-scale vertical gradients and a realistic nutricline.

In response, we have added an appendix showing the GEOTRACES comparison of Figure 1 herein, plus a brief discussion on what features one should not expected to be captured by our coarse-resolution steady-state estimates (Appendix G, pages 45–46).

**In addition, I have the following minor comments:**

**1. I agree with Reviewer #1 that caveats of neglecting DOP cycling need to be more carefully considered. Ignoring DOP will not only bias the total estimated export, but also its pattern and therefore potentially the contribution of different Fe sources to export. Particularly, DOP convergence is thought to provide a significant P supply to subtropical gyres, and essentially "relocates" export downstream, from tropical and coastal upwelling zones into the gyres. Given that benthic Fe supports most export in upwelling zones and atmospheric Fe supports most export in gyres, relocating exporting between those two regimes seems important.**

We agree that not carrying DOP has (minor) implications for converting our phosphorous (P) export to carbon export as discussed in response to Reviewer 1. However, our optimized P export effectively includes DOP effects in spite of DOP not being explicitly represented. The optimization of the P-cycle parameters implicitly accounts for all P export by minimizing the mismatch with the observed distribution of $PO_4$, which in the real ocean is determined by the remineralization of DOP (and possibly also of POP directly). We are therefore not concerned about missing or "relocated" export due to lateral transport

and utilization of DOP. This reasoning is supported by the fact that our POP export averaged over the subtropical gyres matches the estimates of *Letscher et al.* [Nature Geoscience, 2016] when we use the same masks (interpolated to our grid) to define the subtropical gyres. *Letscher et al.* [2016] explicitly model DOP transport and utilization and find a mean subtropical-gyre POP export of $10\pm2\,\mathrm{mmol\,P\,m^{-2}\,yr^{-1}}$, which agrees with our corresponding export of $10\pm1\,\mathrm{mmol\,P\,m^{-2}\,yr^{-1}}$ (mean and standard deviation across our family of estimates). Our estimates agree with *Letscher et al.*'s despite the fact that we do not have DOP contributing to biological production, which underscores that our estimates of POP export implicitly account for DOP effects.

Regarding the patterns of dFe-supported P export, it is true that aeolian dFe support is more important than sedimentary dFe support in the subtropical gyres, although both aeolian and benthic dFe are important in upwelling regions (Figure 9 of our manuscript). However, as discussed above, we capture effects due to DOP implicitly and there is no reason to think that export has been relocated out of the subtropical gyres. Our estimates of the Fe-type-supported export patterns are by construction consistent with the available nutrient data and we showed that they are robust for a range of Fe sources with widely different ratios of the benthic to aeolian source strengths.

In response to this comment, and in addition to our revisions in response to Referee 1, we have added some discussion to the manuscript about the agreement with the subtropical POP exports of *Letscher et al.* [2016] when we more fully discuss our choice of omitting an explicit representation of DOP (page 24).

**2. What is the justification for choosing such widely different export ratios between plankton types (page 5, line 13)? The authors cite Dunne 2005, but there have been other studies since (e.g., Richardson 2007) that suggest small plankton contribute as much export, relative to their NPP, as large plankton.**

Reviewer 2 is correct that the values of our detrital fractions, $f_c$, are the optimized values from the work of *Dunne et al.* [2005], that is, $f_{\mathrm{sml}} = 0.14$ and $f_{\mathrm{lrg}} = f_{\mathrm{dia}} = 0.74$. This means that export due to large phytoplankton is $\sim$5 times larger than export due to small phytoplankton for the same amount of production. We prescribed the $f_c$ parameters because they are not constrainable from the nutrient and plankton concentration data alone. The $f_c$ values set the fractional

export of each phytoplankton class. However, the nutrient observations constrain only the total export of all classes, while the phytoplankton data [*Kostadinov et al.*, 2016] constrain only the concentration of the different functional classes and thus their respective *uptake*, but not their respective *export*. We think that prescribing $f_c$ to take the values from the study of *Dunne et al.* [2005] was an appropriate choice, and we will argue below that this is not contradicted by the findings of *Richardson and Jackson* [2007], although we acknowledge that the precise $f_c$ values are uncertain.

*Richardson and Jackson* [2007] suggest that export efficiencies of large and small phytoplankton should be closer to each other, specifically that "the relative contributions of various phytoplankton size classes to carbon export are proportional to their contributions to total net primary production". However, their data is consistent with $f_c$ having different values for different classes. Figure 2 herein shows the local fractional uptake plotted versus the corresponding fractional export of each phytoplankton class for our typical estimate. The small class collapses to a compact curve because $f_{\mathrm{dia}} = f_{\mathrm{lrg}}$. If we chose different "detrital" fractions for the Diatoms and Large class, then the "small" curve would become a cloud of points whose boundaries are determined by the ratios $f_{\mathrm{lrg}}/f_{\mathrm{sml}}$ and $f_{\mathrm{dia}}/f_{\mathrm{sml}}$. Specifically, if one of these ratios were reduced to 4 or 3, the "small" points would spread out somewhat toward the 1:1 line. *Richardson and Jackson* [2007] show that the export through mezozooplankton dominates the purely detrital export by about an order of magnitude so that our "detrital" fraction pertains to the export through the mezozooplankton pathway. Therefore, Figure 2 should be compared to Fig. 1B of *Richardson and Jackson* [2007] for mezozooplankton-driven export. Note that our Diatom and Large points in Figure 2 are broadly consistent with the measurements plotted by *Richardson and Jackson* [2007], while our compact curve for the Small class is very close to two of the Arabian-Sea picoplankton measurements. Given the extreme sparseness of the measurements and the fact that they are presented without error bars, we think the *Richardson and Jackson* [2007] paper provides no strong evidence that our $f_c$ values should be drastically different from those of *Dunne et al.* [2005]. We acknowledge however that there is significant uncertainty in the precise values of the $f_c$ parameters and that the small-to-large ratio may well be smaller than 5, although the *Richardson and Jackson* [2007] data certainly does not suggest that it should be unity.

In response, we have relegated Figure 7 of the manuscript, which shows the export production of each functional class, to an appendix because the partition of the export among functional classes depends strongly on our choice of the

prescribed, somewhat uncertain $f_c$ parameters. (Former Figure 7 is now Figure F1 on page 45.) In addition, we have added a brief discussion on the uncertainty in the $f_c$ values where they are defined (page 6).

**3. What is the justification for not prescribing a minimum Fe:P quota in equation 14? It is impossible to sustain phytoplankton growth with no Fe, so if the model is optimizing towards zero it means that model is straying into unrealistic parameter space, not that this parameter can be neglected. The authors should set a reasonable lower limit during the optimization (e.g. low end of the range shown in Moore et al 2013), rather than allowing the Fe quota to approach zero at low [Fe].**

The justification for not prescribing a minimum Fe:P quota in Equation (14) is that it turns out that none is required to obtain realistic state estimates for the following reasons: For very low dFe concentrations, our Fe:P cellular ratio would fall below a realistic cell quota, but this has no mechanistic consequence because for such low dFe concentrations there is essentially no uptake in our formulation. This is because our Fe:P ratio is proportional to a dFe Monod term, while phosphate uptake is proportional to the square of a dFe Monod term. Thus, as dFe becomes small, the uptake goes to zero faster than the Fe:P quota itself. Simply put, this means that when Fe:P is unrealistically small, it does not matter because there is no P or Fe uptake.

One should therefore regard our formulation to have a minimum Fe:P quota of zero, and we should not have stated on page 8 that we "ignored it for simplicity". The fact that optimizing a minimum quota (constrained to be non-negative) resulted in an optimized minimum quota of zero means that a simple Monod factor suffices to capture the dFe dependence of the Fe:P ratio where there is significant uptake. In response, we have reworded these passages to make these points explicit (page 9).

We thank Reviewer 2 for making us revisit our formulation of the Fe:P ratio, which made us realize that there was an issue not with the minimum Fe:P ratio, but with the parameter $R_0^{\mathrm{Fe:P}}$. $R_0^{\mathrm{Fe:P}}$ multiplies the dFe Monod term in the Fe:P ratio and is thus the maximum attainable Fe:P ratio at high dFe. For a small fraction of our state estimates, the optimization pushed $R_0^{\mathrm{Fe:P}}$ to near-zero values. This is very unrealistic as it means significant P uptake and export are maintained without Fe uptake. In response, we have now corrected

this by excluding cases for which the optimized $R_0^{\text{Fe:P}} < 0.5\,\text{mmol}\,\text{Fe}\,(\text{mol}\,\text{P})^{-1}$ from our family of state estimates. We have updated all our figures accordingly and note that removing these unphysical outliers makes no visual difference, although it narrows the range of Fe export across our family of estimates to $0.87\text{–}5.6\,\text{Gmol}\,\text{Fe}\,\text{yr}^{-1}$. Where we discuss our family of solutions (Sec. 3.4), we have added a brief discussion on the fact that we excluded state estimates with $R_0^{\text{Fe:P}} < 0.5\,\text{mmol}\,\text{Fe}\,(\text{mol}\,\text{P})^{-1}$ because they are unrealistic (page 18).

[revised manuscript text omitted]

$$R^{\text{Fe:P}} = R_0^{\text{Fe:P}} \frac{\chi_{\text{Fe}}}{\chi_{\text{Fe}} + k_{\text{Fe:P}}} \quad , \tag{14}$$

where $R_0^{\text{Fe:P}}$ is the maximal Fe : P uptake ratio. In principle, $R_0^{\text{Fe:P}}$ and $k_{\text{Fe:P}}$ could be different for different functional classes. However,  constraining class-dependent Fe quotas is beyond the scope of what is possible with our inverse model: different values of $R_0^{\text{Fe:P}}$ and $k_{\text{Fe:P}}$ for  each class would directly compensate for one another in the global Fe

5   export. As noted by Galbraith et al. (2010), this formulation ignores the effects of light limitation suggested by several studies (e.g., Sunda and Huntsman, 1997; Strzepek et al., 2012).

Equation (14)  effectively encodes a minimum iron requirement  of zero. Thus, for very low dFe concentrations, our Fe : P ratio $R^{\text{Fe:P}}$ would fall below a realistic cell quota, but this has no mechanistic consequence because for such low dFe concentrations there is essentially no uptake in our formulation. This is because $R^{\text{Fe:P}}$ is proportional to a

10   dFe Monod term, while P uptake is proportional to the square of a dFe Monod term. Thus, as dFe becomes small, the uptake goes to zero faster than $R^{\text{Fe:P}}$ itself. Simply put, this means that when the Fe : P ratio is unrealistically small, it does not matter because there is no P or Fe uptake. When we introduced  a nonzero minimum for $R^{\text{Fe:P}}$, it tended to be optimized to zero, which means that a simple Monod factor suffices to capture the dFe dependence of the Fe : P ratio where there is significant uptake.

15   The Monod formulation (14) does capture luxury iron uptake (e.g., Marchetti et al., 2009a) when the half-saturation constant of (14) exceeds the half-saturation constant of the iron limitation in (11), as made explicit by Galbraith et al. (2010). This is the case for our optimized value of $k_{\text{Fe:P}}$ so that phytoplankton has the luxury to increase its iron uptake with increasing dFe concentration even when iron is not limiting.

Our representation of the $R^{\text{Si:P}}$ uptake ratio takes into consideration field studies and iron enrichment experiments, which

20   have indicated that in HNLC regions and upwelling regions iron limitation leads to increased diatom silicification, i.e., increased cellular Si : N and Si : P ratios (e.g., Takeda, 1998; Hutchins and Bruland, 1998; Franck et al., 2000; Brzezinski et al., 2003). However, there is no literature consensus on a mechanistic formulation of the iron dependence of silicic-acid uptake. For example, Matsumoto et al. (2013) assume a Si : N uptake ratio inversely proportional to the dFe concentration (capped at a minimum), while Jin et al. (2006) assume the Si : N ratio to depend only on the $\text{Si(OH)}_4$ concentration. Others suggest that

25   the dFe concentration only impacts the diatom growth rate and not the cellular Si : C ratio, while the $\text{Si(OH)}_4$ concentration impacts the cellular Si : C ratio and not growth rate (e.g., Marchetti et al., 2009b; Brzezinski et al., 2011a). Here, we chose to retain the effects of increased silicification due to iron limitation and the impact of high $\text{Si(OH)}_4$ concentration on silicification (Brzezinski 2016, personal communication). We model these effects with the formulation

$$R^{\text{Si:P}} = R_0^{\text{Si}} + \left(R_\text{m}^{\text{Si}} - R_0^{\text{Si}}\right) \frac{k_{\text{Si:P}}^{\text{Fe}}}{\chi_{\text{Fe}} + k_{\text{Si:P}}^{\text{Fe}}} \frac{\chi_{\text{Si}}}{\chi_{\text{Si}} + k_{\text{Si:P}}^{\text{Si}}} \quad . \tag{15}$$

[revised manuscript text omitted]
| $k_{I,\mathrm{lrg}}$ | Large class irradiance half-saturation rate | 20. | 9.0 | – | $\mathrm{W\,m^{-2}}$ |
| $k_{I,\mathrm{sml}}$ | Small class irradiance half-saturation rate | 20. | 8.8 | – | $\mathrm{W\,m^{-2}}$ |
| $R_{\mathrm{m}}^{\mathrm{Si}}$ | Diatom maximum Si : P | 160. | 220. | – | $\mathrm{mol\,Si\,(mol\,P)^{-1}}$ |
| $R_{0}^{\mathrm{Si}}$ | Diatom minimum Si : P | 8.0 | 13. | – | $\mathrm{mol\,Si\,(mol\,P)^{-1}}$ |
| $k_{\mathrm{Si:P}}^{\mathrm{Si}}$ | Silicon half-saturation constant in Si : P | 30. | 4.0 | – | $\mathrm{mmol\,Si\,m^{-3}}$ |
| $k_{\mathrm{Si:P}}^{\mathrm{Fe}}$ | Iron hyperbolic constant in Si : P | 1.0 | 0.077 | – | nM Fe |
| $k_{\mathrm{Fe:P}}$ | Iron half-saturation constant in Fe : P | 0.74 | 0.74 | – | nM Fe |
| $k_{\mathrm{dia}}^{\mathrm{P}}$ | Diatom class $PO_4$ half-saturation constant | 0.39 | 0.72 | – | $\mathrm{mmol\,P\,m^{-3}}$ |
| $k_{\mathrm{lrg}}^{\mathrm{P}}$ | Large class $PO_4$ half-saturation constant | 0.39 | 0.72 | – | $\mathrm{mmol\,P\,m^{-3}}$ |
| $k_{\mathrm{sml}}^{\mathrm{P}}$ | Small class $PO_4$ half-saturation constant | 0.030 | 0.13 | – | $\mathrm{mmol\,P\,m^{-3}}$ |
| $k_{\mathrm{dia}}^{\mathrm{Fe}}$ | Diatom class dFe half-saturation constant | 0.10 | 0.30 | – | nM Fe |
| $k_{\mathrm{lrg}}^{\mathrm{Fe}}$ | Large class dFe half-saturation constant | 0.10 | 0.29 | – | nM Fe |
| $k_{\mathrm{sml}}^{\mathrm{Fe}}$ | Small class dFe half-saturation constant | 0.010 | 0.11 | – | nM Fe |
| $p_{\mathrm{dia}}^{\mathrm{max}}$ | Diatom class maximum concentration | 23. | 42. | – | $\mathrm{mg\,C\,m^{-3}}$ |
| $p_{\mathrm{lrg}}^{\mathrm{max}}$ | Large class maximum concentration | 23. | 61. | – | $\mathrm{mg\,C\,m^{-3}}$ |
| $p_{\mathrm{sml}}^{\mathrm{max}}$ | Small class maximum concentration | 23. | 21. | – | $\mathrm{mg\,C\,m^{-3}}$ |
| $\tau_{\mathrm{dia}}$ | Maximal diatom growth timescale | 6.0 | 0.65 | – | d |
| $\tau_{\mathrm{lrg}}$ | Maximal large growth timescale | 6.0 | 1.5 | – | d |
| $\tau_{\mathrm{sml}}$ | Maximal small growth timescale | 6.0 | 7.4 | – | d |
| $R_{0}^{\mathrm{Fe:P}}$ | Maximum Fe : P uptake ratio | 5.0 | 2.0 | 0.52–3.0 | $\mathrm{mmol\,Fe\,(mol\,P)^{-1}}$ |
| $\kappa_{\mathrm{scv}}^{\mathrm{POP}}$ | POP scavenging rate constant | 0.13 | 1.0 | 0.015–7.9 | $\mathrm{(mmol\,POP\,m^{-3})^{-1}\,d^{-1}}$ |
| $\kappa_{\mathrm{scv}}^{\mathrm{bSi}}$ | Opal scavenging rate constant | 3.1 | 1.3 | 0.85–13. | $\mathrm{(mol\,bSi\,m^{-3})^{-1}\,d^{-1}}$ |
| $\kappa_{\mathrm{scv}}^{\mathrm{dst}}$ | Dust scavenging rate constant | 10000. | 9.4 | 8.5–10. | $\mathrm{(g\,dust\,m^{-3})^{-1}\,d^{-1}}$ |
| $L_{\mathrm{b}}$ | Background ligand concentration | 1.0 | 0.51 | 0.40–0.72 | nM Lig |
| $L_{\mathrm{v}}$ | Maximal hydrothermal vent ligand conc. | 3.0 | 1.2 | 0.68–1.4 | nM Lig |
| $\tau_{\mathrm{b}}$ | Hydrothermal vent plume restoring timescale | 10. | 5.7 | 3.0–7.5 | yr |
| $L_{\mathrm{max}}$ | Maximal age-enhanced ligand conc. | 2.3 | 0.97 | 0.82–1.3 | nM Lig |
| $\sigma_{\mathrm{A}}$ | Aeolian source strength | 1.9 | 5.3 | 0.63–22. | $\mathrm{Gmol\,Fe\,yr^{-1}}$ |
| $\sigma_{\mathrm{S}}$ | Sedimentary source strength | 4.2 | 1.7 | 0.11–22. | $\mathrm{Gmol\,Fe\,yr^{-1}}$ |
| $\sigma_{\mathrm{H,ATL}}$ | Hydrothermal source strength, Atlantic | 0.098 | 0.19 | 0.00013–0.50 | $\mathrm{Gmol\,Fe\,yr^{-1}}$ |
| $\sigma_{\mathrm{H,PAC}}$ | Hydrothermal source strength, Pacific | 0.21 | 0.42 | 0.035–2.9 | $\mathrm{Gmol\,Fe\,yr^{-1}}$ |
| $\sigma_{\mathrm{H,IND}}$ | Hydrothermal source strength, Indian O. | 0.066 | 0.13 | 0.011–0.80 | $\mathrm{Gmol\,Fe\,yr^{-1}}$ |
| $\sigma_{\mathrm{H,SO}}$ | Hydrothermal source strength, Southern O. | 0.066 | 0.13 | 0.011–1.2 | $\mathrm{Gmol\,Fe\,yr^{-1}}$ |

[revised manuscript text omitted]
_{\text{m}}^{\text{Si}}$, $R_0^{\text{Si}}$, $k_{\text{Si:P}}^{\text{Si}}$, and $k_{\text{Si:P}}^{\text{Fe}}$

10 together with the three growth timescales $\tau_c$ to match the fractions of total uptake by each phytoplankton class as estimated by Uitz et al. (2010).

(e) Because of compensation with the maximum $\text{Fe} : \text{P}$ uptake ratio, the associated half-saturation rate, $k_{\text{Fe:P}}$, was optimized on its own.

(f) We then jointly re-optimized the 13 parameters $k_c^{\text{P}}$, $k_c^{\text{Fe}}$, $k_{\text{dia}}^{\text{Si}}$, $p_c^{\text{max}}$, and $\tau_c$.

15 (g) Only the parameters of the iron cycle remain to be optimized. Iron source and sink parameters cannot jointly be optimized because of strong local compensation. (Although the aeolian source injects $\text{dFe}$ at the surface, uptake and scavenging export iron to depth, thus creating an effective interior source.) To generate our family of estimates, we therefore first assigned the aeolian, sedimentary, and hydrothermal source-strength parameters (keeping the same ratio of basin hydrothermal source strengths to global hydrothermal source strength), and held these fixed while jointly optimizing the parameters determining the

20 iron scavenging, namely $R_0^{\text{Fe:P}}$, $\kappa_{\text{scv}}^{\text{POP}}$, $\kappa_{\text{scv}}^{\text{bSi}}$, $\kappa_{\text{scv}}^{\text{dst}}$, $L_{\text{v}}$, $L_{\text{max}}$, $L_{\text{b}}$, and $\tau_{\text{
[revised manuscript text omitted]